# Learn the Ropes, Then Trust the Wins: Self-imitation with Progressive Exploration for Agentic Reinforcement Learning

**Yulei Qin**[1][*][✉]  **Xiaoyu Tan**[1][*][✉]  **Zhengbao He**[2][*]  **Gang Li**[1]  **Haojia Lin**[1]  **Zongyi Li**[1]
**Zihan Xu**[1]  **Yuchen Shi**[1]  **Siqi Cai**[1]  **Renting Rui**[1,2]  **Shaofei Cai**[3]  **Yuzheng Cai**[1,4]
**Xuan Zhang**[1,4]  **Sheng Ye**[1,5]  **Ke Li**[1]  **Xing Sun**[1]

[1]Tencent Youtu Lab  [2]Shanghai Jiao Tong University  [3]Peking University
[4]Fudan University  [5]Xiamen University  [*]Equal Contribution
{yuleiqin, arthurtan}@tencent.com
 Codes    Data & Checkpoints

## Abstract

Reinforcement learning (RL) is the dominant paradigm for sharpening strategic tool use capabilities of LLMs on long-horizon, sparsely-rewarded agent tasks, yet it faces a fundamental challenge of exploration-exploitation trade-off. Existing studies stimulate exploration through the lens of policy entropy, but such mechanical entropy maximization is prone to RL instability due to the multi-turn distribution shifting. In this paper, we target the progressive exploration-exploitation balance under the guidance of the agent's own experiences without succumbing to either entropy collapsing or runaway divergence. We propose SPEAR 🗡, a self-imitation learning (SIL) recipe for training agentic LLMs. It extends the vanilla SIL, where a replay buffer stores good experience for off-policy update, by gradually steering the policy entropy across stages. Specifically, the proposed curriculum scheduling harmonizes intrinsic reward shaping and self-imitation to 1) expedite exploration via frequent tool interactions at the beginning, and 2) strengthen exploitation of successful tactics upon convergence towards familiarity with the environment. We also combine bag-of-tricks of industrial RL optimizations for a strong baseline Dr.BoT to demonstrate our effectiveness. In ALFWorld and WebShop, SPEAR increases the success rates of GRPO/GiGPO/Dr.BoT by up to 16.1%/5.1%/8.6% and 20.7%/11.8%/13.9%, respectively. In AIME24 and AIME25, SPEAR boosts Dr.BoT by up to 3.8% and 6.1%, respectively. Such gains incur only 10%–25% extra theoretical complexity and negligible runtime overhead in practice, demonstrating the plug-and-play scalability of SPEAR.

## 1 Introduction

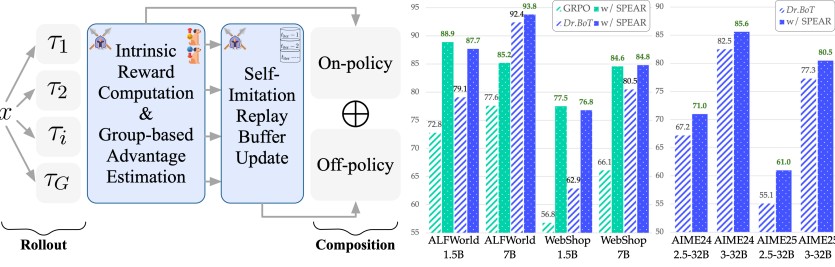

Figure 1: Our SPEAR harmonizes the curriculum-scheduled self-imitation learning with intrinsic reward shaping for progressive exploration, improving policy performance across agentic tasks.

Reinforcement Learning (RL) (Lambert et al., 2024; Guo et al., 2025; Qin et al., 2025b) has driven the development of reasoning capabilities of Large Language Models (LLMs). Built upon the reason-and-act (ReAct) paradigm (Yao et al., 2023), LLMs have powered various agentic applications such as simulated robot navigation (Shridhar et al., 2020; Li et al., 2024), mobile assistant (Wang et al., 2024; Li et al., 2025a), web navigator (Furuta et al., 2023; He et al., 2024), deep searcher (Jin et al., 2025b; Li et al., 2025c; Tao et al., 2025), and GUI master (Qin et al., 2025a; Hong et al., 2024). A fundamental challenge in applying RL to LLM agents is to manage the balance between exploration and exploitation. The LLM agent needs to *exploit* both pretrained knowledge and past interactions to formalize experience that maximize rewards. At the same time, it must *explore* novel behaviors through tool-integrated reasoning and reflection. The interweaving between exploration and exploitation determines the emerging agent's competence upon convergence.

Existing studies often quantify the exploration potential through entropy (Sutton, 1988; Williams & Peng, 1991; Cui et al., 2025b; Xue et al., 2025), where the decline of policy entropy indicates overconfidence with insufficient exploration. In this case, a series of regularization techniques (Ziebart et al., 2008; Schulman et al., 2017b; Haarnoja et al., 2018) have been proposed to maximize entropy (Haarnoja et al., 2017; Zhao et al., 2019; Xin et al., 2020; Zhang et al., 2021; Seo et al., 2021; Mehr et al., 2023; Kim & Sung, 2023; Hao et al., 2023). However, when it comes to LLM-driven agents, entropy-based control is fragile: the accumulation of low-probability tokens from the environment feedback induces severe distribution shifting, often leading to mode collapse (Xue et al., 2025; Dong et al., 2025b). Agent models may experience sustained entropy growth due to uncertainty about multi-turn interactions and training instability becomes frequent (Mai et al., 2025; Yao et al., 2025; Wang et al., 2025b). Recent approaches attempt to mitigate this issue by cold-start supervised fine-tuning (SFT) (Tao et al., 2025; Qin et al., 2025a; Feng et al., 2025a; Qin et al., 2025c) or hybrid schemes that combine RL with SFT (Zhang et al., 2025a). Although these methods improve stability, they compromise policy's discovery of strategies beyond those present in the SFT corpus. This limitation highlights the need for adaptive training frameworks that can *dynamically schedule LLM-driven agents to decide when to explore and when to exploit*.

In this paper, we are trying to answer the following core research question: **Can we schedule a smooth transition between *exploration* and *exploitation* guided by the policy's own experience without going to extremes of either entropy collapsing or runaway divergence?** We hypothesize that the agent should maintain its policy entropy within a dynamic but controlled range that evolves over time: 1) **At the early stages, *increasing entropy* is beneficial for broad *skill-level* exploration**. The agent is expected to rapidly develop tool-use capabilities, encounter unfamiliar observations, and engage in trial-and-errors. 2) **As training advances, however, a shift toward *converging entropy* is required**. This enables the agent to consolidate problem-solving heuristics and emphasize *action-level* exploration. The agent exploits reward signals to choose comparatively more effective actions and adapts to changing distributions for stabilizing its evolutionary path.

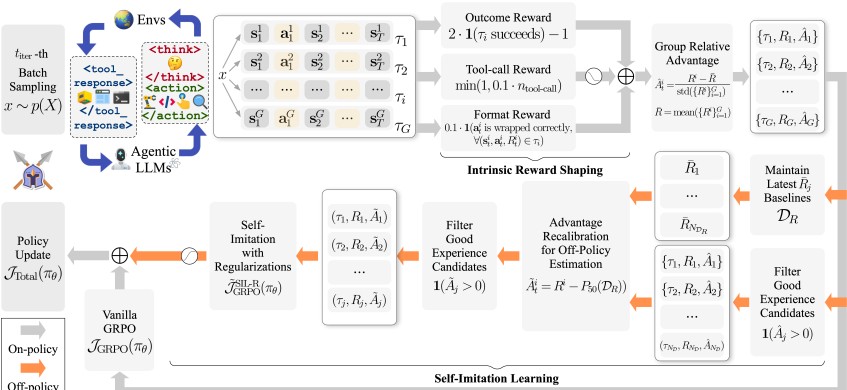

Figure 2: Overview of SPEAR. First, the agent interacts with the environment for a set of trajectories, which flow through intrinsic reward shaping and advantage estimation with on-policy updates. Second, they are selected and stored in a replay buffer, enabling off-policy updates via the proposed self-imitation scheme. This dual integration allows the maximal utility of past experiences, thereby expanding the effective exploration space, while simultaneously mitigating persistent uncertainty.

To address this, we propose the **S**elf-imitation with **P**rogressive **E**xploration for **A**gentic **R**einforcement learning (**SPEAR**⚔), a curriculum-based RL recipe for improving the *exploration-exploitation* balance with *self-imitation* and *intrinsic reward*. As shown in Figure 1, the core principle follows the vanilla Self-Imitation Learning (SIL) (Oh et al., 2018; Ferret et al., 2020) where an independent replay buffer is prepared to store the state-action pairs only when their returns in the past episodes exceed the baselines. Such a replay buffer is exploited to encourage actions with good returns and improve hard exploration based on these successful trajectories under the sparse-reward, long-horizon agent tasks. Specifically, we introduce three modifications to SIL tailored to the dynamics of policy entropy in agentic tasks. First, we incorporate a curriculum to integrate both *skill-level* and *action-level* exploration by adjusting reward shaping and self-imitation across stages. Second, we tackle the off-policy nature of the update with experiences in the buffer and avoid advantage recomputation by advantage recalibration. Third, we regularize policy updates to stabilize entropy and mitigate reward hacking. Finally, inspired by existing industrial bag-of-tricks, we present a strong baseline *Dr.BoT* for agentic RL training. Our SPEAR brings considerable performance gains to GRPO/GiGPO (Feng et al., 2025b)/*Dr.BoT* respectively up to 16.1%/5.1%/8.6% on ALFWorld (Shridhar et al., 2020) and 20.7%/11.8%/13.9% on WebShop (Yao et al., 2022). It boosts our *Dr.BoT* respectively up to 3.8% on AIME24 and 6.1% on AIME25 (AIME, 2025). These gains come with around $10\% \sim 25\%$ computation overhead in theoretical complexity, but end up with quite comparable runtime per iteration in practice. Such compatibility and scalability enable SPEAR a plug-and-play algorithm for training versatile agents. In summary, our contributions are:

1) We propose SPEAR, a generalization of the SIL for training LLM agents. It bypasses the costly expert imitation and allows exploration under the guidance of one's own rewarded experience.

2) We bring in curriculum scheduling to harmonize SIL with intrinsic reward shaping for policy entropy management and progressive transition from *skill-based* to *action-based* exploration.

3) We propose a strong baseline, *Dr.BoT*, which combines established RL techniques validated in industrial practice, confirming its effectiveness and superiority over existing baselines.

## 2  RELATED WORK

### 2.1  REINFORCEMENT LEARNING ALGORITHMS FOR LLMS

With the advent of large-scale reasoning models (Jaech et al., 2024), Reinforcement Learning (RL) (Ouyang et al., 2022) has been adopted more broadly. Proximal Policy Optimization (PPO) (Schulman et al., 2017b) leverages an actor–critic architecture together with the clipped surrogate objective and a Kullback–Leibler (KL) divergence penalty to constrain policy update. Group Relative Policy Optimization (GRPO) (Guo et al., 2025; Shao et al., 2024) simplifies this setup by replacing the critic with a group-wise baseline. Building on GRPO, DAPO (Yu et al., 2025) uses dynamic sampling and "clip higher" to encourage exploration and stabilize training. Dr.GRPO (Liu et al., 2025b) addresses length bias and the difficulty bias. Existing methods have greatly advanced RL for LLMs. However, naively combining them can lead to conflicts or tight couplings among techniques. To this end, we harmonize the strengths of DAPO, Dr.GRPO, and other agent studies from research and industrial practice to establish a strong baseline, *Dr.BoT*, as detailed in Section 4.4.

### 2.2  OPTIMIZATION OF LLM AGENTS

Recent researches investigate how to endow models with better tool-use capabilities (Feng et al., 2025a; Li et al., 2025d; Xue et al., 2025). LLMs are optimized to strengthen information seeking from open web (Jin et al., 2025b; Tao et al., 2025; Gao et al., 2025). RAGEN (Wang et al., 2025b) improves the stability of multi-turn RL through instance filtering and gradient shaping. GiGPO (Feng et al., 2025b) augments group-level advantages with additional step-level advantage estimates. ARPO (Dong et al., 2025b) monitors entropy dynamics during rollouts to branch trajectories adaptively. In this work, we address the exploration–exploitation dilemma under multi-turn tool-use settings. We introduce a curriculum–regulated RL regime that gradually shifts skill-based exploration towards action-based exploration. We integrate self-imitation and intrinsic reward to consolidate successful behaviors (Section 4.2). Our SPEAR can work with existing algorithms in a plug-and-play manner, exhibiting a high level of compatibility and generalization.

## 2.3 EXPLORATION IN REINFORCEMENT LEARNING

Curiosity-driven methods (Pathak et al., 2017; Houthooft et al., 2016) grant intrinsic rewards for prediction error or novelty to actively seek unfamiliar states. Count-based algorithms (Bellemare et al., 2016; Tang et al., 2017) introduce pseudo-counts derived from a density model to assign count-based bonuses. Skill acquisition methods (Gregor et al., 2016; Eysenbach et al., 2018) discover distinct options by maximizing the mutual information. Entropy-regularization methods (Haarnoja et al., 2018; Cui et al., 2025b) maximize the expected reward and entropy of the policy. However, traditional exploration techniques can lead to divergence of agent LLMs as the multi-turn interactions already result in the increased uncertainty on unfamiliar observations. Under such circumstance, we propose the curriculum-guided self-imitation to leverage the agent's own experience for balancing exploration and exploitation. It avoids handcrafted heuristic techniques in previous studies and instead fully relies on the agent itself to reinforce successful and valid patterns.

## 2.4 EXPERIENCE REPLAY IN REINFORCEMENT LEARNING

Self-Imitation Learning (SIL) (Oh et al., 2018) takes advantage of past successful experience to drive its future learning (Schaul et al., 2015; Horgan et al., 2018; Gangwani et al., 2018; Pan et al., 2022; Saglam et al., 2023). SAIL (Ferret et al., 2020) extends SIL to off-policy, action value-based RL methods. Tang (2020) proves that SIL's return-based update provides a bias–variance trade-off that speeds up learning. SILfD (Pshikhachev et al., 2022) extends SIL to leverage both external demonstrations and the agent's own experience. GSIL (Xiao et al., 2024) proposes an offline alignment framework that uses self-imitation on demonstration data. While SIL benefits long-horizon problems, its induces entropy collapsing to agent RL. To mitigate this, we harmonize both self-imitation and intrinsic reward with curriculum scheduling for progressive exploration.

## 3 PRELIMINARIES

### 3.1 PROBLEM DEFINITION

Given a task $x \sim p(X)$ where $p(X)$ represents data distribution, an LLM agent parameterized by $\theta$ interacts with the environment $E$ until it completes the task or exceeds the max number of turns $T$. It can be modeled by Markov Decision Process (MDP) where $\mathbf{s}_t$, $\mathbf{a}_t$, and $R_t$ respectively denote the state, action, and reward at time $t$. Given a full episode $\tau = \{(\mathbf{s}_1, \mathbf{a}_1, R_1), (\mathbf{s}_2, \mathbf{a}_2, R_2), ...\}$, we aim to optimize the agent policy $\pi_\theta$. Following previous studies (Dong et al., 2025a; Feng et al., 2025b;a; Dong et al., 2025b), we define three distinct types of actions (see Appendix A.2).

### 3.2 POLICY OPTIMIZATION

We adopt the GRPO (Shao et al., 2024) which stems from PPO (Schulman et al., 2017a;b) but replaces the model-based $A$ (Schulman et al., 2015) with the group-based $\hat{A}$ (Appendix A.3).

## 4 TRAINING AGENTIC LLMS WITH SPEAR 🗡

### 4.1 PRELIMINARY FINDINGS

The extension of SIL to LLM-driven agents faces entropy collapse. Figure 3 illustrates that the overfitting of the few available successful experience causes irreversible stagnation of exploration. In addition, we demonstrate that the inclusion of the tool-call reward is a double-edged sword (Figure 4), where the competition between reward terms causes the oscillations to converge. To address these challenges, we introduce SPEAR for progressive exploration with self-imitation (Algorithm 1).

### 4.2 SELF-IMITATION LEARNING

We resort to self-imitation to unearth past successful experience for effective *action-level* exploration, where the agent learns novel strategies along the promising decision path instead of random walk and bifurcation. We prevent policy entropy divergence by replaying rewarded trajectories.

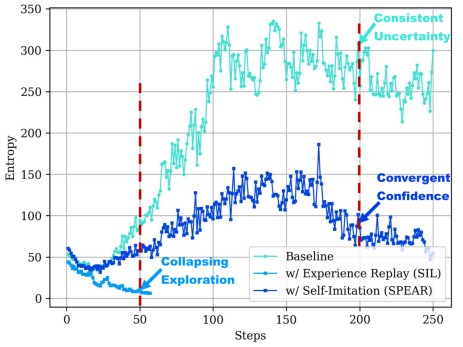 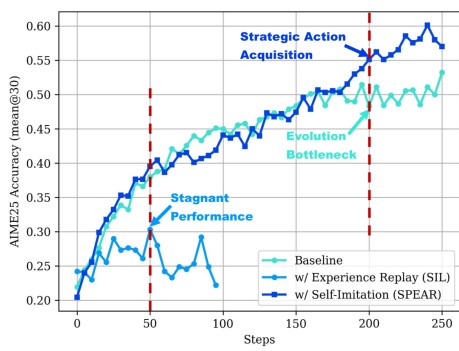

(a) Entropy (`seq-mean-token-sum-norm`).

(b) Accuracy on AIME 2025.

Figure 3: Effect of our self-imitation on action-level strategy exploration (Qwen2.5-32B with code interpreter). The vanilla experience replay technique (Oh et al., 2018) that enforces early overfitting of the few available trajectories in the buffer causes entropy collapsing and exploration shrinkage. At the beginning, the LLM agent struggles at tool-calling skills and fails to cultivate the transition of distribution towards frequent tool utilization and tool-integrated reasoning. The naive replay limits the transformation of reasoning paradigm. In contrast, our SPEAR introduces both curriculum- and covariance- based regularization into self-imitation. Its curriculum schedule with an increasing emphasis on the replay data allows easy acquisition of tool-use skills at first, and stimulates strategic action plans later. The covariance clipping removes over-confident tokens, whose log probabilities are highly associated with their advantage gains, out of optimization. Our self-imitation gives promises to exploring novel strategies and achieves steady growth on AIME 2025.

**Prioritized Experience Replay in Self-Imitation.**     A replay buffer is maintained to store previous trajectories, their rewards and advantages $\mathcal{D} = \{(\tau_j, R_j, \hat{A}_j)\}, j = 1, 2, ..., N_\mathcal{D}$ where $N_\mathcal{D}$ denotes the buffer size. To exploit only good trajectories, we keep those with positive advantages:

$$\mathcal{J}_{\text{GRPO}}^{\text{SIL}}(\pi_\theta) = \mathbb{E}_{\{\tau_j\}_{j=1}^{N_\mathcal{D}} \sim \{\pi_{\theta_{\text{old}}}(\cdot|x), \; x \sim p(X)\}} \sum_{j=1}^{N_\mathcal{D}} \mathcal{J}_{\text{GRPO}}^j \cdot \mathbf{1}(\hat{A}_j > 0), \tag{1}$$

where the indicator $\mathbf{1}(\cdot)$ equals to 1 when the condition satisfied and 0 otherwise. The past trajectories not only come from the last policy $\pi_{\theta_{old}}$ but also the policies $\{\pi_{\theta_{\text{old}}}\}$ of few steps earlier.

**Advantage Recalibration for Off-Policy Estimation.**     We propose to recalibrate the advantage of trajectories in the buffer to address the underlying off-policy challenge. That is to say, the observed return of a trajectory from the past policy becomes increasingly different from the current one, under the assumption that the policy keeps improving during iterations (Ferret et al., 2020; Luo et al., 2021). Under this assumption, vanilla SIL computes the advantage with a pointwise max with the per-state empirical return as a baseline, which can be seen as a proxy for the upper-envelope projection of the value function onto empirical returns. GRPO removes the learned value baseline by estimating the state-dependent baseline performance through its reliance on intra-group reward averaging, but this still depends on the target policy and requires extra computation resources for sampling. Dynamic adjustment on the baseline performance is performed to calibrate relative gains without introducing additional computing. Specifically, we maintain a First-In-First-Out (FIFO) buffer of intra-group baselines for the latest $N_{\mathcal{D}_R}$ trajectories $\mathcal{D}_R = \{\bar{R}_j\}_{j=1}^{\mathcal{D}_R}$ where $N_{\mathcal{D}_R}$ denotes the size of the baseline buffer. As training progresses, due to the high variance nature of agentic RL, we utilize the 50-th percentile $P_{50}(\mathcal{D}_R)$ as a conservative but robust estimation of the policy baseline with either upward or downward trends. To bypass the inaccurate estimation of intra-group standard deviation, we follow (Liu et al., 2025b) to simply remove such a term in advantage computation:

$$\tilde{A}_t^i = R^i - P_{50}(\mathcal{D}_R). \tag{2}$$

Such recalibrated advantage enjoys three benefits: 1) the baseline performance correlates with the policy change; 2) the outdated experiences can be filtered out with both $\hat{A}_j > 0$ and $\tilde{A}_j > 0$; 3) the

difficulty bias by group normalization can be mitigated. The updated off-policy SIL objective is:

$$\tilde{\mathcal{J}}_{\text{GRPO}}^{\text{SIL}}(\pi_\theta) = \mathbb{E}_{\{\tau_j\}_{j=1}^{N_\mathcal{D}} \sim \{\pi_{\theta_{\text{old}}}(\cdot|x), \, x \sim p(X)\}} \sum_{j=1}^{N_\mathcal{D}} \tilde{\mathcal{J}}_{\text{GRPO}}^{j} \cdot \mathbf{1}(\hat{A}_j > 0 \, \& \, \tilde{A}_j > 0), \qquad (3)$$

$$\tilde{\mathcal{J}}_{\text{GRPO}}^{i} = \left[ \frac{1}{T} \sum_{t=1}^{T} (\min(r_t^i(\theta) \tilde{A}_t^i, \text{clip}(r_t^i(\theta), 1 - \epsilon, 1 + \epsilon) \tilde{A}_t^i) - \beta D_{\text{KL}}^i(\pi_\theta || \pi_{\text{ref}}) \right]. \qquad (4)$$

**Progressive Experience Utilization with Curriculum Schedule.** We perform scheduling to 1) restrict mechanical imitation of probable-yet-immature experience at an early stage, and 2) prevent consistent uncertainty about the environment states and policy actions at later stage. We apply a warm-up $\gamma$ on the SIL term under the assumption that initially the transition of distribution towards diverse actions outweighs the imitation of limited solution patterns (see Equation 13 and Figure 6a).

$$\mathcal{J}_{\text{Total}}(\pi_\theta) = \mathcal{J}_{\text{GRPO}}(\pi_\theta) + \gamma \cdot \tilde{\mathcal{J}}_{\text{GRPO}}^{\text{SIL}}(\pi_\theta). \qquad (5)$$

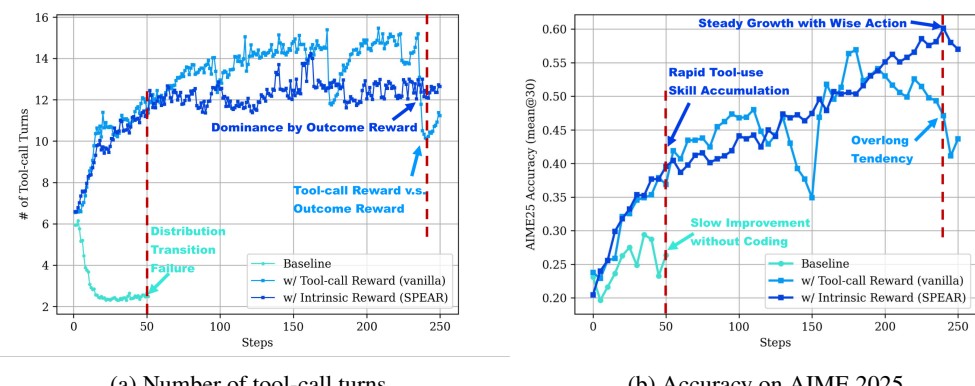

(a) Number of tool-call turns.        (b) Accuracy on AIME 2025.

Figure 4: Effect of our intrinsic reward on skill-level strategy exploration (Qwen2.5-32B with code interpreter). The baseline does not consider tool-calling as a rewarded behavior and its number of interaction with the environment drops quickly due to the negative feedback of bad codes. In this case, the LLM gives up coding and degrades to text-based reasoning. The vanilla tool-call reward, despite being effective in learning tool-call skills at first, causes competition with the outcome reward later. Due to the limited context length, the excessive tool-call turns prevents submission of the final answer and thereafter the accuracy declines immediately. We propose the curriculum schedule as an intrinsic reward design where its strength decays over step to allow the agent to merely focus on the accuracy with wiser actions. It prevents reward hacking for unnecessarily long interactions.

### 4.3 INTRINSIC REWARD SHAPING

We resort to intrinsic reward for *skill-level* exploration where the agent is guided by a tool-call reward to broadly investigate tool usage. Such design not only benefits tool learning but more importantly stimulates interactions that familiarize the agent with the environment for experience accumulation.

**Reward Composition.** A compound reward $R^i$ of each trajectory $\tau_i$ not only considers the final outcome but also the behaviors that are promising to achieve the goal: an outcome accuracy reward $R_{\text{outcome}}^i$, a continuous tool-call reward $R_{\text{tool-call}}^i$, and a format reward $R_{\text{format}}^i$ (see Appendix A.7).

**Progressive Reward Modulation with Curriculum Schedule.** We regulate the contribution of tool call reward to: 1) accelerate the mastering of tool usage for quick distribution shifting towards new task settings at an early stage, and 2) prevent optimization oscillation and competition at a later

stage. Although previous studies (Qian et al., 2025; Li et al., 2023; Da et al., 2025; Xia et al., 2025; Singh et al., 2025; Wei et al., 2025; Gou et al., 2023; Lin & Xu, 2025) experimented with various auxiliary rewards, we show that the addition of tool-call reward is a double-edged sword. The agent trained without the tool-call reward fails to develop tool-integrated reasoning (Figure 4) due to negative tool response: 1) missing `import` of modules; 2) reference to `undefined` variables; 3) unexpected `indentation` error; and 4) forgetting to `print` results. The agent quickly gives up coding to run away from errors and turns to pure textual reasoning. On the other hand, the enforcement of tool-call reward stimulates an increasing number of interaction turns, leading to over-long responses that cause oscillation to outcome accuracy. We alleviate the competition between reward terms by scheduling the tool-call reward with $\mu$ (Equation 14 and Figure 6b):

$$R^i = R^i_{\text{outcome}} + \mu \cdot R^i_{\text{tool-call}} + R^i_{\text{format}}. \tag{6}$$

### 4.4 *Dr.BoT* AS A STRONG BASELINE

To provide a strong baseline, we refer to the existing studies (Liu et al., 2025c; Sun et al., 2025; Bai et al., 2025a; Cui et al., 2025b) for diverse exploration, stable convergence, and effective training. Our baseline, *Dr.BoT*, consists of bag-of-tricks modifications to the GRPO (see Appendix A.8).

Table 1: Results on ALFWorld & WebShop (%). PT & FW stand for prompting & framework.

| Type | Method | ALFWorld | | | | | | | WebShop | |
|------|--------|------|------|-------|------|------|-------|-----|-------|-----|
| | | Pick | Look | Clean | Heat | Cool | Pick2 | All | Score | SR |
| | | | | | *Qwen2.5-1.5B-Instruct* | | | | | |
| PT | I/O | 5.9 | 5.5 | 3.3 | 9.7 | 4.2 | 0.0 | 4.1 | 23.1 | 5.2 |
| FW | ReAct | 17.4 | 20.5 | 15.7 | 6.2 | 7.7 | 2.0 | 12.8 | 40.1 | 11.3 |
| FW | Reflexion | 35.3 | 22.2 | 21.7 | 13.6 | 19.4 | 3.7 | 21.8 | 55.8 | 21.9 |
| RL | PPO | 64.8 | 40.5 | 57.1 | 60.6 | 46.4 | 47.4 | 54.4 | 73.8 | 51.5 |
| RL | RLOO | 88.3 | 52.8 | 71.0 | 62.8 | 66.4 | 56.9 | 69.7 | 73.9 | 52.1 |
| RL | GRPO | 85.3 | 53.7 | 84.5 | 78.2 | 59.7 | 53.5 | 72.8 | 75.8 | 56.8 |
| RL | + SPEAR (ours) | 93.9 | 80.9 | 96.4 | 87.4 | 88.3 | 79.1 | 88.9(+16.1%) | 90.0 | 77.5(+20.7%) |
| RL | *Dr.BoT* (GRPO) | 92.2 | 75.8 | 81.0 | 81.8 | 72.8 | 61.9 | 79.1 | 78.7 | 62.9 |
| RL | + SPEAR (ours) | 91.2 | 72.2 | 94.1 | 95.1 | 88.3 | 74.4 | 87.7(+8.6%) | 88.4 | 76.8(+13.9%) |
| RL | GiGPO w/std | 94.4 | 67.5 | 94.8 | 94.4 | 79.8 | 76.4 | 86.7 | 83.1 | 65.0 |
| RL | GiGPO w/o std | 96.0 | 76.5 | 91.8 | 91.3 | 71.7 | 79.5 | 86.1 | 83.5 | 67.4 |
| RL | + SPEAR (ours) | 95.2 | 79.2 | 89.1 | 94.0 | 88.8 | 95.5 | 91.2(+5.1%) | 90.7 | 79.3(+11.8%) |
| RL | *Dr.BoT* (GiGPO) | 98.6 | 91.4 | 93.7 | 93.8 | 85.4 | 78.4 | 90.6 | 84.1 | 68.8 |
| RL | + SPEAR (ours) | 96.4 | 86.5 | 96.1 | 99.0 | 87.6 | 91.6 | 93.2(+2.6%) | 90.9 | 81.1(+12.2%) |
| | | | | | *Qwen2.5-7B-Instruct* | | | | | |
| PT | I/O | 33.4 | 21.6 | 19.3 | 6.9 | 2.8 | 3.2 | 14.8 | 26.4 | 7.8 |
| FW | ReAct | 48.5 | 35.4 | 34.3 | 13.2 | 18.2 | 17.6 | 31.2 | 46.2 | 19.5 |
| FW | Reflexion | 62.0 | 41.6 | 44.9 | 30.9 | 36.3 | 23.8 | 42.7 | 58.1 | 28.8 |
| RL | PPO | 92.3 | 64.0 | 92.5 | 89.5 | 80.3 | 68.8 | 80.4 | 81.4 | 68.7 |
| RL | RLOO | 87.6 | 78.2 | 87.3 | 81.3 | 71.9 | 48.9 | 75.5 | 80.3 | 65.7 |
| RL | GRPO | 90.8 | 66.1 | 89.3 | 74.7 | 72.5 | 64.7 | 77.6 | 79.3 | 66.1 |
| RL | + SPEAR (ours) | 93.7 | 62.4 | 97.2 | 78.0 | 83.1 | 75.5 | 85.2(+7.6%) | 92.4 | 84.6(+18.5%) |
| RL | *Dr.BoT* (GRPO) | 99.9 | 95.8 | 93.8 | 92.8 | 90.4 | 80.6 | 92.4 | 90.4 | 80.5 |
| RL | + SPEAR (ours) | 98.8 | 97.9 | 97.1 | 88.5 | 89.2 | 87.2 | 93.8(+1.4%) | 91.4 | 84.8(+4.3%) |
| RL | GiGPO w/std | 97.7 | 82.7 | 98.8 | 83.7 | 89.3 | 79.2 | 90.8 | 84.4 | 72.8 |
| RL | GiGPO w/o std | 91.8 | 88.6 | 95.9 | 90.2 | 86.5 | 85.2 | 90.2 | 86.2 | 75.2 |
| RL | + SPEAR (ours) | 99.9 | 82.4 | 98.0 | 92.8 | 92.6 | 86.6 | 94.1(+3.9%) | 92.7 | 83.8(+8.6%) |
| RL | *Dr.BoT* (GiGPO) | 98.3 | 99.9 | 96.9 | 92.8 | 91.8 | 88.3 | 94.0 | 90.7 | 81.8 |
| RL | + SPEAR (ours) | 99.9 | 85.1 | 95.6 | 96.4 | 89.9 | 95.1 | **94.7**(+0.7%) | 92.5 | **85.7**(+3.9%) |

## 5 EXPERIMENTS

### 5.1 EXPERIMENTAL SETUP

Three benchmarks are used: ALFWorld (Shridhar et al., 2020), WebShop (Yao et al., 2022), and DAPO-MATH-17K (Yu et al., 2025) (Appendix A.10). According to these benchmarks, we respectively follow (Feng et al., 2025b) and (Feng et al., 2025a) to report a range of competitive baselines (Appendix A.11). All the training settings and hyper-parameters are detailed in Appendix A.12.

### 5.2 PERFORMANCE

Table 1 demonstrates our effectiveness on ALFWorld and WebShop. It is compatible with GRPO (Shao et al., 2024), GiGPO (Feng et al., 2025b), and our *Dr.BoT*. SPEAR brings consistent gains across 1.5B and 7B models up to 20%. Such generalization benefits from the collection of

successful trajectories, which acts as a walkthrough guide to the agent. Especially for tasks where the success rate is fairly low at the beginning, the agent has to figure out the underlying interaction logics and summarize action plans tailored specific to each task. The experience replay expedites the accumulation of tactics and thereafter reduces blind trials and errors. Furthermore, our *Dr.BoT* boosts GRPO and GiGPO up to 15%, showcasing the validity of mixture of tricks.

Table 2: Results (mean@30) on AIME 2024 & 2025 (%). [†]: Official implementation already utilizes DAPO tricks. [‡]: Official results reported by Qwen (Yang et al., 2025). PT stands for prompting.

| Type | Method | Model | Tool | Context Train | Context Test | AIME24 | AIME25 |
|------|--------|-------|------|-------|------|--------|--------|
| PT | I/O | *Qwen2.5-32B-Instruct* | – | – | 16K | 13.4 | 12.9 |
| PT | I/O | *Qwen2.5-32B-Instruct* | CI | – | 16K | 29.6 | 23.1 |
| RL | PPO[†] | *Qwen2.5-32B-Instruct* | CI | 16K | 16K | – | 55.0 |
| RL | GRPO[†] | *Qwen2.5-32B-Instruct* | CI | 16K | 16K | – | 60.0 |
| RL | ReTool | *Qwen2.5-32B-Instruct* | CI | 16K | 16K | 67.0 | 49.3 |
| RL | SimpleTIR | *Qwen2.5-32B-Instruct* | CI | 12K | 12K | 59.9 | 49.2 |
| RL | ZeroTIR | *Qwen2.5-32B-Instruct* | CI | 8K | 8K | 56.7 | 33.3 |
| RL | AFM | *Qwen2.5-32B-Instruct* | CI | 32K | 32K | 66.7 | 59.8 |
| RL | *Dr.BoT* (GRPO) | *Qwen2.5-32B-Instruct* | CI | 16K | 16K | 64.7 | 54.0 |
| RL | + SPEAR (ours) | *Qwen2.5-32B-Instruct* | CI | 16K | 16K | 66.3(+1.6%) | 60.1(+6.1%) |
| RL | *Dr.BoT* (GRPO) | *Qwen2.5-32B-Instruct* | CI | 32K | 32K | 67.2 | 55.1 |
| RL | + SPEAR (ours) | *Qwen2.5-32B-Instruct* | CI | 32K | 32K | 71.0(+3.8%) | 61.0(+5.9%) |
| PT | I/O | *Qwen3-32B-Instruct* | – | – | 16K | 68.5 | 53.5 |
| PT | I/O[‡] | *Qwen3-32B-Instruct* | – | – | 38K | 81.4 | 72.9 |
| PT | I/O | *Qwen3-32B-Instruct* | CI | – | 16K | 31.1 | 24.4 |
| RL | *Dr.BoT* (GRPO) | *Qwen3-32B-Instruct* | CI | 16K | 16K | 81.3 | 74.1 |
| RL | + SPEAR (ours) | *Qwen3-32B-Instruct* | CI | 16K | 16K | 81.8(+0.5%) | 78.8(+4.7%) |
| RL | *Dr.BoT* (GRPO) | *Qwen3-32B-Instruct* | CI | 32K | 32K | 82.5 | 77.3 |
| RL | + SPEAR (ours) | *Qwen3-32B-Instruct* | CI | 32K | 32K | 85.6(+3.1%) | 80.5(+3.2%) |

Table 2 reports the performance of CI-integrated reasoning on AIME24 and AIME25. *Dr.BoT* indeed outperforms recent RL baselines. The reduced context length of Qwen3 impedes complete reasoning and answer parsing. The agent learns to exploit the CI feedback for double-check and self-reflection. SPEAR achieves comparable performance with Qwen3 but using a much smaller token budget. When the context is relaxed to 32K, an improvement is observed on both Qwen2.5 and Qwen3, confirming our generalization with more interactions turns and reasoning tokens.

## 5.3 ABLATION STUDY

Table 3: Ablation on ALFWorld & WebShop. SI & IR stand for Self-Imitation & Intrinsic Reward.

| Type | Method | ALFWorld Pick | Look | Clean | Heat | Cool | Pick2 | All | WebShop Score | SR |
|------|--------|------|------|-------|------|------|-------|-----|-------|-----|
| | | *Qwen2.5-1.5B-Instruct* | | | | | | | | |
| RL | GRPO | 85.3 | 53.7 | 84.5 | 78.2 | 59.7 | 53.5 | 72.8 | 75.8 | 56.8 |
| RL | + SI | 86.8 | 61.0 | 87.4 | 87.7 | 71.1 | 56.6 | 77.3(+4.5%) | 85.1 | 74.2(+17.4%) |
| RL | + SI + IR (SPEAR) | 93.9 | 80.9 | 96.4 | 87.4 | 88.3 | 79.1 | 88.9(+16.1%) | 90.0 | 77.5(+20.7%) |
| RL | GiGPO w/o std | 96.0 | 76.5 | 91.8 | 91.3 | 71.7 | 79.5 | 86.1 | 83.5 | 67.4 |
| RL | + SI | 93.2 | 82.5 | 96.3 | 87.4 | 92.7 | 87.5 | 90.6(+4.5%) | 89.4 | 79.0(+11.6%) |
| RL | + SI + IR (SPEAR) | 95.2 | 79.2 | 89.1 | 94.0 | 88.8 | 95.5 | 91.2(+5.1%) | 90.7 | 79.3(+11.8%) |
| | | *Qwen2.5-7B-Instruct* | | | | | | | | |
| RL | GRPO | 90.8 | 66.1 | 89.3 | 74.7 | 72.5 | 64.7 | 77.6 | 79.3 | 66.1 |
| RL | + SI | 93.2 | 82.5 | 96.3 | 87.4 | 92.7 | 87.5 | 90.6(+13.0%) | 90.4 | 83.4(+17.3%) |
| RL | + SI + IR (SPEAR) | 93.7 | 62.4 | 97.2 | 78.0 | 83.1 | 75.5 | 85.2(+7.6%) | 92.4 | 84.6(+18.5%) |
| RL | GiGPO w/o std | 91.8 | 88.6 | 95.9 | 90.2 | 86.5 | 85.2 | 90.2 | 86.2 | 75.2 |
| RL | + SI | 96.1 | 81.9 | 98.4 | 95.3 | 94.5 | 83.9 | 93.6(+3.4%) | 94.6 | 87.5(+12.3%) |
| RL | + SI + IR (SPEAR) | 99.9 | 82.4 | 98.0 | 92.8 | 92.6 | 86.6 | 94.1(+3.9%) | 92.7 | 83.8(+8.6%) |

**Self-Imitation.** The SIL improves baselines consistently across model scales (Table 3). Since either 1.5B or 7B models perform poorly at the early stage (i.e., success rate $< 15\%$), past experiences are quite beneficial to explore promising strategies. The re-use of trajectories facilitates convergence and prevents mechanical trials especially for small agents. Table 4 shows that AIME24 dropped a bit by self-imitation but AIME25 still gets improved. Such fluctuation is related to the phenomenon (Figure 4) where the imitation of samples with multiple tool calls leads to rapid increase of interaction turns and thereafter causes training instability. The competition between different reward terms affects the robust selection of good experience, ultimately degrading the effectiveness of SIL.

Table 4: Ablation on AIME 2024 & 2025 (%). SI & IR stand for Self-Imitation & Intrinsic Reward.

| Type | Method | Model | Tool | Context Train | Context Test | AIME24 | AIME25 |
|------|--------|-------|------|-------|-------|--------|--------|
| RL | *Dr.BoT* (GRPO) | *Qwen2.5-32B-Instruct* | CI | 16K | 16K | 64.7 | 54.0 |
| RL | + SI | *Qwen2.5-32B-Instruct* | CI | 16K | 16K | 63.8(-0.9%) | 56.9(+2.9%) |
| RL | + SI + IR (SPEAR) | *Qwen2.5-32B-Instruct* | CI | 16K | 16K | 66.3(+1.6%) | 60.1(+6.1%) |
| RL | *Dr.BoT* (GRPO) | *Qwen2.5-32B-Instruct* | CI | 32K | 32K | 67.2 | 55.1 |
| RL | + SI | *Qwen2.5-32B-Instruct* | CI | 32K | 32K | 66.0(-1.2%) | 60.5(+5.4%) |
| RL | + SI + IR (SPEAR) | *Qwen2.5-32B-Instruct* | CI | 32K | 32K | 71.0(+3.8%) | 61.0(+5.9%) |
| RL | *Dr.BoT* (GRPO) | *Qwen3-32B-Instruct* | CI | 16K | 16K | 81.3 | 74.1 |
| RL | + SI | *Qwen3-32B-Instruct* | CI | 16K | 16K | 81.2(-0.1%) | 75.8(+1.70%) |
| RL | + SI + IR (SPEAR) | *Qwen3-32B-Instruct* | CI | 16K | 16K | 81.8(+0.5%) | 78.8(+4.70%) |
| RL | *Dr.BoT* (GRPO) | *Qwen3-32B-Instruct* | CI | 32K | 32K | 82.5 | 77.3 |
| RL | + SI | *Qwen3-32B-Instruct* | CI | 32K | 32K | 81.8(-0.7%) | 78.2(+0.9%) |
| RL | + SI + IR (SPEAR) | *Qwen3-32B-Instruct* | CI | 32K | 32K | **85.6**(+3.1%) | **80.5**(+3.2%) |

**Intrinsic Reward.** The rewarding of interaction turns benefit 1.5B models consistently (Table 3). Two 7B outliers are found where the self-imitation alone brings the most performance gains. Such exception might be related to both the task definition and the RL algorithm. One should experiment with different combinations in practice. Table 4 shows that the intrinsic reward is indispensable for both Qwen2.5 and 3 because it encourages transformation from text-based reasoning to tool-integrated reasoning. It promotes frequent tool calling and such rich observation signals motivate the agent to correct coding errors, check the validity of the answer, and reflect on alternative solutions.

## 5.4 GENERALIZATION ON VISION-LANGUAGE AGENTS

Table 5: Success rate (%) of the visual agent for playing Sokoban.

| Type | Method | Sokoban |
|------|--------|---------|
| | *Qwen2.5-VL-3B-Instruct* | |
| PT | I/O | 11.7 |
| RL | GRPO | 67.1 |
| RL | + SPEAR (ours) | 86.7(+19.6%) |
| RL | *Dr.BoT* (GRPO) | 76.0 |
| RL | + SPEAR (ours) | 85.4(+9.4%) |
| RL | GiGPO w/ std | 76.9 |
| RL | GiGPO w/o std | 81.0 |
| RL | + SPEAR (ours) | 87.7(+6.7%) |
| RL | *Dr.BoT* (GiGPO) | 81.3 |
| RL | + SPEAR (ours) | 87.9(+6.6%) |

 

(a) Before (step 15).     (b) After (step 125).

Figure 5: The agent learns to push the box.

To test whether the proposed SPEAR is still complimentary to existing GRPO-like algorithms on training visual agents, we follow (Feng et al., 2025b) to conduct experiments on the popular visual game Sokoban (Schrader, 2018). In this setting, the Qwen2.5-VL-3B-Instruct (Bai et al., 2025b) is adopted as the agentic LLM to solve the puzzle game where the player must push the boxes along the grid towards target positions without hitting the walls. It challenges the agent on spatial comprehension and long-term planning capabilities. The grid size is of $6 \times 6$ and the visual agent receives both the visual (RGB arrays) and textual inputs as states. As shown in Table 5, the proposed method generally improves the performance on Sokoban with either GRPO, GiGPO, and the proposed *Dr.BoT* baselines. At first, the visual agent is unaware of the winning logic behind the game and wanders around for "aimlessly" exploration (see Figure 5). After optimization, it not only comprehends the spatial relationship to control the box but also learns to stop moving when the task is completed.

## 5.5 GENERALIZATION ON SEARCH-AUGMENTED QA TASKS

To evaluate the performance of SPEAR on knowledge-intensive reasoning tasks, we conduct experiments on search-augmented QA tasks, including the single-hop QA datasets (NQ (Kwiatkowski et al., 2019), TriviaQA (Joshi et al., 2017), and PopQA (Mallen et al., 2023)) and multi-hop QA datasets (HotpotQA (Yang et al., 2018), 2Wiki (Ho et al., 2020), MuSiQue (Trivedi et al., 2022), and Bamboogle (Press et al., 2023)). We follow the experimental settings of SearchR1 (Jin et al.,

Table 6: Results on search-augmented QA Tasks.

| Type | Method | Single-Hop QA | | | | Multi-Hop QA | | | Avg. |
|------|--------|-----|----------|-------|----------|-------|---------|-----------|------|
| | | NQ | TriviaQA | PopQA | HotpotQA | 2Wiki | MuSiQue | Bamboogle | |
| | | | | *Qwen2.5-7B-Instruct* | | | | | |
| RL | Search-R1 | 39.3 | 61.0 | 39.7 | 37.0 | 40.1 | 14.6 | 36.8 | 38.5 |
| RL | SPEAR | 35.7 | 62.7 | 34.5 | 46.9 | 43.4 | 17.2 | 44.8 | 40.7 |
| | | | | *Qwen2.5-14B-Instruct* | | | | | |
| RL | Search-R1 | 48.8 | 67.7 | 48.2 | 45.5 | 47.0 | 21.1 | 51.6 | 49.1 |
| RL | SPEAR | 47.6 | 69.3 | 47.8 | 48.5 | 48.8 | 26.7 | 56.6 | 49.3 |

2025b; Gao et al., 2025) to launch the local wiki-18 retrieval service. We adopt the Hierarchical Navigable Small World (HNSW) CPU indexing as approximation of nearest neighbor retrieval. Our SPEAR with GRPO improves over the Search-R1 baseline on average, especially on the multi-hop QA benchmarks. Such multi-hop QA datasets require reasoning for problem decomposition and several turns of information seeking. In this case, our intrinsic reward that encourages multiple tool uses for broad exploration prevents arbitrary conclusions with only one or two searches. Our SPEAR respectively requires $\sim 14.48$ and $\sim 14.42$ calls for 7B and 14B models, respectively. Such behavior is expected due to the stimulation of exploration at the beginning. Despite the QA tasks are relatively short-horizon, the agent still benefits from the detailed decomposition of the complex queries with cross-validation via step-by-step searching. Note that our retrieval service adopts the HNSW E5 embedding for efficient training, which slightly impedes performance (Jin et al., 2025a).

## 5.6 MORE DISCUSSIONS

Due to the page limit, discussions on theoretical analysis on convergence A.9, hyper-parameters A.13, qualitative analysis A.14, training cost and complexity A.15, and future research directions A.16 are presented in the appendix. One could easily adapt SPEAR to training any (M)LLM-driven agents robustly without binding to a specific optimization algorithm.

## 6 CONCLUSIONS AND LIMITATIONS

In this paper, we target the pivotal challenge of balancing exploration and exploitation in RL training of LLM agents. Our proposed solution, SPEAR, extends the vanilla SIL by advantage recalibration, scheduled entropy control, and intrinsic rewards. These components work in a curriculum manner to prevent policy collapse and excessive uncertainty, progressively guiding the policy through a smooth transition between exploration and exploitation. In addition, we propose a strong baseline *Dr.BoT* tailored for agentic RL with existing bag-of-tricks verified from numerical industrial practices. Empirical results across tasks and models showcase SPEAR's superiority over existing methods, with performance boosts and acceptable computational overhead. The effectiveness of our SPEAR underscores the value of learning from past experiences while managing policy entropy, offering a robust framework for training LLMs with strong reasoning and tool integration skills.

There exist two potential limitations: 1) The vague definition of good experiences under highly complex, stochastic environments with unreliable tools. In such cases, observations can be noisy and severely degrade the feasibility of the task. The sparse outcome reward cannot distinguish between good and bad experiences and thereafter the relative advantages might be simply attributed to randomness instead of the agent's behavior. We suggest a possible solution that more fine-grained, stepwise supervision should be enforced. For example, a step-wise process reward that evaluates the logical consistency (Zhang et al., 2025b) of the agent's thought and action might be helpful. 2) The rigidity of entropy control which relies on prior-based scheduling and covariance-based clipping. In the present study, the proposed scheduling and clipping designs might not be optimal for all kinds of agentic tasks. A more adaptive solution lies in the policy's self-confidence on decisions under each observation. One might use the token-level dynamic reweighting for SIL (Wu et al., 2025) which avoids over-concentration on certain low-probability reference tokens in the replay buffer. Similarly, the clipping could depend on token probability instead of the bounded random sampling. We leave the exploration mentioned above as a promising direction for improvement in the future.

**Ethics Statement.** The present study conforms with the ICLR Code of Ethics. The paper does not involve crowdsourcing nor research with human subjects.

**Reproducibility Statement.** All datasets used in the paper are publicly accessible (see Section 5.1). All the codes are available at `https://github.com/TencentYoutuResearch/SPEAR` for reproduction. In addition, we provide all the details of implementation in Section A.12.

**Acknowledgment.** Zhengbao He acknowledges support from the AI for Science Program, Shanghai Municipal Commission of Economy and Informatization (No. 2025-GZL-RGZN-BTBX-02026).

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

# A APPENDIX

## A.1 SUMMARY OF THE APPENDIX

In the appendix, we provide detailed explanations on the following.

- Descriptions about the Action Space
- Brief Introduction to the PPO and GRPO
- PseudoCode of the SPEAR
- Visualization of the Curriculum Schedule
- Definition of the Reward Function
- Descriptions about RL Bag-of-Tricks
- Theoretical Analysis on Convergence
- Descriptions of the Data and Environment
- Choice of Baselines
- Implementation Details
- Discussions and Guidelines on Hyper-parameters
- Qualitative Analysis
- Training Cost and Complexity
- Future Research Directions

## A.2 DETAILED ACTION SPACE

The following contents correspond to Section 3.1 in the main text.

**TextWorld Embodied Tool.** The embodied actions follows ALFWorld (Shridhar et al., 2020) where a language-driven agent interacts with the TextWorld (Côté et al., 2018). It allows the agent to take one of the following high-level actions: `goto {recep}`, `take {obj} from {recep}`, `put {obj} in/on {recep}`, `open {recep}`, `close {recep}`, `toggle {obj}{recep}`, `clean {obj} with {recep}`, `heat {obj} with {recep}`, and `cool {obj} with {recep}`, where `{obj}` and `{recep}` denote `objects` and `receptacles`, respectively.

**Web Browsing Tool.** The definition of web browsing follows WebShop (Yao et al., 2022) where only two actions are allowed: `search[query]` and `choose[button]` where `query` and `button` respectively stand for searching `query` and clickable elements such as `back to search`, `prev/next page`, `{product title}`, `{option}`, `{desc/overview}`, `previous`, and `buy`.

**Code Interpreter Tool.** The code interpreter executes the code generated by the language model and return both the `stdout` and `stderr`. If the code runs correctly, the `stdout` contains the output. On the other hand, the compiler error messages are provided for the next-round correction. We follow (Feng et al., 2025a) to deploy a SandBox (Bytedance-Seed-Foundation-Code-Team et al., 2025) service that receives execution requests from the interpreter tool. In addition, we add a reminder in the `stdout` for empty output when the LLM forgets to `print` computation results: *Empty stdout! You might forget to print the answer.* For non-empty `stderr`, we also add an instruction as hint: *Errors occurred! Check your code*.

## A.3 DETAILED POLICY OPTIMIZATION ALGORITHMS

The following contents correspond to Section 3.2 in the main text.

**Proximal Policy Optimization (PPO).** Typically, PPO optimizes the following:

$$\mathcal{J}(\pi_\theta) = \mathbb{E}_{x \sim p(X), \mathbf{a} \sim \pi_\theta(\cdot|x,\mathbf{s})}\left[\mathcal{R}(x, \mathbf{s}, \mathbf{a}) - \beta D_{\text{KL}}[\pi_\theta(\cdot|x, \mathbf{s})||\pi_{\text{ref}}(\cdot|x, \mathbf{s})]\right], \quad (7)$$

where $\mathcal{R}(x, \mathbf{s}, \mathbf{a}) = \sum_{t=1}^{T} r_t(x, \mathbf{s}_t, \mathbf{a}_t)$ is the return (Sutton et al., 1998) for the trajectory and $\pi_{\text{ref}}$ is the reference policy model. The KL divergence proposed (Christiano et al., 2017) to prevent the policy $\pi_\theta$ from deviating greatly from the reference $\pi_{\text{ref}}$ ($\beta > 0$). In consideration of the simplicity, we follow TULU 3 (Lambert et al., 2024) to adopt RL with the verifiable reward where the rule-based verifiers are designed to provide the outcome reward signal $r$ instead of the reward model $r_\theta$. In addition, we follow (Liu et al., 2025b) to drop the KL term by setting $\beta = 0$, which not only emphasizes agent performance but also saves memory and computation during training.

**Group Relative Policy Optimization (GRPO).** Specifically, the policy model $\pi_{\theta_{\text{old}}}$ from the previous iteration generates a group of $G$ individual trajectories $\{\tau_i\}_{i=1}^{G}$. GRPO updates the policy $\pi_\theta$ by maximizing the objective below.

$$\mathcal{J}_{\text{GRPO}}(\pi_\theta) = \mathbb{E}_{x \sim p(X), \{\tau_i\}_{i=1}^{G} \sim \pi_{\theta_{old}}(\cdot|x)} \frac{1}{G} \sum_{i=1}^{G} \mathcal{J}_{\text{GRPO}}^{i}, \quad (8)$$

$$\tau_i = \{(\mathbf{s}_1^i, \mathbf{a}_1^i, R_1^i), (\mathbf{s}_2^i, \mathbf{a}_2^i, R_2^i), ..., (\mathbf{s}_T^i, \mathbf{a}_T^i, R_T^i)\},$$

$$\mathcal{J}_{\text{GRPO}}^{i} = \frac{1}{T} \sum_{t=1}^{T} \min\left[r_t^i(\theta)\hat{A}_t^i, \text{clip}[r_t^i(\theta), 1 - \epsilon, 1 + \epsilon]\hat{A}_t^i\right] - \beta D_{\text{KL}}^i(\pi_\theta||\pi_{\text{ref}}), \quad (9)$$

$$r_t^i = \frac{\pi_\theta(\mathbf{a}_t^i|x, \mathbf{s}_t^i)}{\pi_{\theta_{\text{old}}}(\mathbf{a}_t^i|x, \mathbf{s}_t^i)}, \hat{A}_t^i = \frac{R^i - \bar{R}}{\text{std}(\{R^i\}_{i=1}^{G})}, \bar{R} = \text{mean}(\{R^i\}_{i=1}^{G}), \quad (10)$$

$$D_{\text{KL}}^i(\pi_\theta||\pi_{\text{ref}}) = \frac{\pi_{\text{ref}}(\mathbf{a}_t^i|x, \mathbf{s}_t^i))}{\pi_\theta(\mathbf{a}_t^i|x, \mathbf{s}_t^i)} - \log\frac{\pi_{\text{ref}}(\mathbf{a}_t^i|x, \mathbf{s}_t^i))}{\pi_\theta(\mathbf{a}_t^i|x, \mathbf{s}_t^i)} - 1. \quad (11)$$

### A.4 PSEUDO CODE

The following contents correspond to Section 4 in the main text.

Algorithm 1 summarizes the full training procedure of the proposed SPEAR. It is noted that our SPEAR is compatible with various baselines such as GRPO (Shao et al., 2024) and GiGPO (Feng et al., 2025b), enjoying a high-level of generalization. Specifically, the algorithm is featured by: 1) Maintenance of a replay buffer and a baseline buffer that respectively stores the trajectories for good experience replay and estimates the current policy's average performance; 2) Recalibration of the previous advantages for off-policy update; 3) Regularization against the pre-mature entropy collapsing; 4) Shaping of the composite intrisic rewards for dominance of the outcome reward.

Compared with the vanilla GRPO-like training, the proposed method only introduced: 1) Additional policy update iterations positively associated with the number of $N_\mathcal{D}$ in terms of computational complexity; 2) A replay buffer of the size $N_\mathcal{D}$ and a baseline performance buffer of the size $N_{\mathcal{D}_R}$ in terms of space complexity.

Since we re-utilize previous trajectories without completely re-computing the rollout generation, log-probability estimation, and the advantages, such operations are light-weight and incur minimal computation overhead. In the present study, we empirically set $N_\mathcal{D} = 2048$ without meticulous hyper-parameter tuning. For both ALFWorld, WebShop, and Sokoban, the number of trajectories per data batch is the product of train_batch_size×n_samples_per_prompt=256 and there exist around 4K turn-level training samples under the VeRL-agent (Feng et al., 2025b) framework. For the DAPO-MATH-17K, the number of trajectories per data batch is 2048 and there exist exactly 2048 trajectory-level training samples under the VeRL (Sheng et al., 2024) framework. In this case, our replay buffer reaches its full capacity around every two or three training steps on average for all experiments. For each policy update by self-imitation, the number of iterations is comparable to that of the vanilla policy update by GRPO under the present settings. The detailed analyses on the training cost and complexity can also be found in Section A.15.

---

**Algorithm 1** Training Agentic LLMs with SPEAR

---

**Require:** Initial policy $\pi_{\theta_{old}}$, data distribution $p(X)$, clipping bounds $\epsilon_{lb}$, $\epsilon_{ub}$, KL penalty $\beta$ ($\beta = 0$), replay buffer $\mathcal{D}$ with buffer size $N_{\mathcal{D}}$, intra-group baseline buffer $\mathcal{D}_R$ with buffer size $N_{\mathcal{D}_R}$, the warm-up factor $\gamma$ with the number of warm-up steps $T_{warm-up}$, covariance clipping bounds $\omega_{lb}$, $\omega_{ub}$, the covariance-based clipping ratio $\lambda$ ($\lambda = 0.02$), the decay factor $\mu$ with the number of decay steps $T_{decay}$, the group size $G$, the maximum allowed interaction turns $T$.
**Ensure:** Updated policy $\pi_\theta$

1: Initialze $\mathcal{D} = \emptyset$ and $\mathcal{D}_R = \emptyset$
2: **for** each training step $t_{iter}$ **do**
3:    Update the old policy model: $\theta_{old} \leftarrow \theta$
4:    # Repeat batch sampling and rollout generation for trajectories
5:    Sample data batch with each unique sample $x \sim p(X)$
6:    # Sample $G$ trajectories $\{\tau_i\}_{i=1}^G$ for each $x$
7:    **for** $i = 1$ to $G$ **do**
8:       Initialize environment states $\mathbf{s}_1^i$
9:       # Sample at most $T$ actions
10:       **for** $t = 1$ to $T$ **do**
11:          Sample action $\mathbf{a}_t^i \sim \pi_\theta(\cdot|x, \mathbf{s}_t^i)$
12:          Execute actions, receive rewards $R_t^i$, observe the new states $\mathbf{s}_{t+1}^i$
13:       **end for**
14:       Organize the trajectory $\tau_i = \{(\mathbf{s}_1^i, \mathbf{a}_1^i, R_1^i), (\mathbf{s}_2^i, \mathbf{a}_2^i, R_2^i), ..., (\mathbf{s}_T^i, \mathbf{a}_T^i, R_T^i)\}$
15:    **end for**
16:    # Apply intrinsic reward shaping for advantage estimation
17:    Compute the vanilla objective $\mathcal{J}_{GRPO}(\pi_\theta)$ via Equation 8 with the decay-scheduled $R^i$ via Equation 6
18:    # Maintain the replay buffer and the baseline buffer
19:    $\mathcal{D}_R \leftarrow \mathcal{D}_R \cup \{\bar{R}\}, \bar{R} = \text{mean}(\{R^i\}_{i=1}^G)$
20:    **while** $|\mathcal{D}_R| > N_{\mathcal{D}_R}$ **do**
21:       Pop the oldest baseline $\mathcal{D}_R \leftarrow \mathcal{D}_R \setminus \{\bar{R}_0\}$
22:    **end while**
23:    **if** $|\mathcal{D}| < N_{\mathcal{D}}$ **then**
24:       # Add trajectories into the buffer only when their advantages are positive
25:       $\mathcal{D} \leftarrow \mathcal{D} \cup \{\tau_i | \hat{A}_i > 0\}$
26:       # Apply on-Policy update with the vanilla GRPO
27:       Update policy by maximizing objective $\mathcal{J}_{GRPO}(\pi_\theta)$
28:    **else**
29:       # Recalibrate the advantage
30:       Compute the newly estimated advantage $\tilde{A}_j$ for all $\tau_j \in \mathcal{D}$ via Equation 2
31:       Only keep $\tau_j$ with positive $\tilde{A}_j$ as $\mathcal{D} \leftarrow \{\tau_j | \tilde{A}_j > 0, \forall \tau_j \in \mathcal{D}\}$
32:       # Apply regularization on self-imitation learning
33:       Compute the self-imitation objective $\tilde{\mathcal{J}}_{GRPO}^{SIL-R}(\pi_\theta)$ via Equation 18 with covariance-based clipping via Equation 19
34:       Apply the warm-up schedule for the total objective $\mathcal{J}_{Total}(\pi_\theta)$ via Equation 5
35:       # Apply both the on-policy and the off-policy update for self-imitation
36:       Update policy by maximizing objective $\mathcal{J}_{Total}(\pi_\theta)$
37:       Reset the replay buffer $\mathcal{D} \leftarrow \emptyset$
38:    **end if**
39: **end for**
40: **return** $\pi_\theta$

---

## A.5 POLICY ENTROPY

The following contents are mentioned in Section 4.2 in the main text.

The policy entropy quantifies the confidence inherent in the actions triggered off by the LLM. Under the context of agent tasks, we measure the average entropy of the entire trajectory $\tau$ for the policy model via sequence-mean-token-sum in accordance with the Dr.GRPO technique (Liu et al., 2025b). Given the training data batch $\mathcal{D}_B$, the entropy is defined as:

$$\mathcal{H}(\pi_\theta, \mathcal{D}_B) = -\mathbb{E}_{\mathcal{D}_B, \pi_\theta}[\log \pi_\theta(\tau|x)] = -\frac{1}{|\mathcal{D}_B|} \sum_{x \in \mathcal{D}_B, x \sim p(X)} \sum_{(\mathbf{s}_t, \mathbf{a}_t) \in \tau} \mathbb{E}_{\mathbf{a}_t \sim \pi_\theta}[\log \pi_\theta(\mathbf{a}_t|x, \mathbf{s}_t)] \tag{12}$$

## A.6 CURRICULUM SCHEDULE

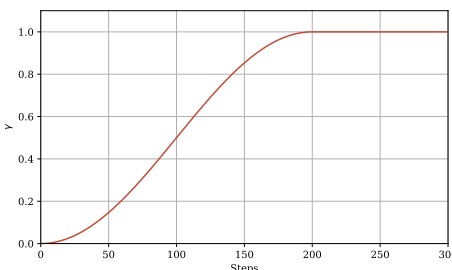 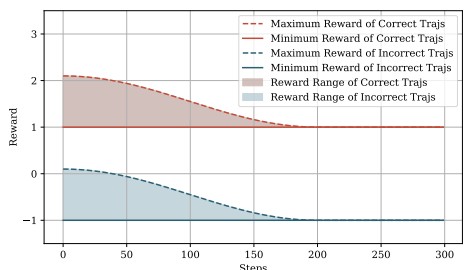

(a) Visualization of $\gamma$ for the SIL term with $T_{\text{warm-up}} = 200$. The weight of SIL loss gradually increases from 0 to 1 in the first $T_{\text{warm-up}}$ steps.

(b) Visualization of the composite intrinsic reward ($T_{\text{decay}} = 200$). The tool-call reward gradually decays from 1 to 0 in the first 200 training steps.

Figure 6: Visualization of the curriculum for progressive exploration.

**Self-Imitation.** The following contents are mentioned in Section 4.2 in the main text.

The schedule for strengthening SI is defined as below:

$$\gamma = \begin{cases} \frac{1}{2}(1 - \cos(\pi \frac{t_{\text{iter}}}{T_{\text{warm-up}}})), & t_{\text{iter}} \leq T_{\text{warm-up}}, \\ 1, & t_{\text{iter}} > T_{\text{warm-up}}, \end{cases} \tag{13}$$

where $t_{\text{iter}}$ and $T_{\text{warm-up}}$ respectively denote the training iteration step and the total warm-up steps.

**Intrinsic Reward.** The following contents are mentioned in Section 4.3 in the main text.

The schedule for decaying IR is defined as below:

$$\mu = \begin{cases} \frac{1}{2}(\cos(\pi \frac{t_{\text{iter}}}{T_{\text{decay}}}) + 1), & t_{\text{iter}} \leq T_{\text{decay}}, \\ 0, & t_{\text{iter}} > T_{\text{decay}}, \end{cases} \tag{14}$$

where $T_{\text{decay}}$ denotes the number of decaying steps.

## A.7 REWARD DEFINITION

The following contents correspond to Section 4.3 in the main text.

**Outcome Reward** A binary signal is assigned at the end of a episode according to the pre-defined verification rules.

$$R_{\text{outcome}}^i = \begin{cases} 1, & \tau_i \text{ succeeds}, \\ -1, & \text{otherwise}. \end{cases} \tag{15}$$

**Tool-call Reward.** To incentivize multi-turn interactions, an action-based reward that is proportional to the number of tool call turns is added. To avoid reward hacking where the LLM repeats meaningless tool calling, the action reward is confined smaller than the outcome reward.

$$R_{\text{tool-call}}^i = \min(1, 0.1 \cdot n_{\text{tool-call}}), n_{\text{tool-call}} \geq 0, \tag{16}$$

where $n_{\text{tool-call}}$ denotes the number of valid tool call turns in the trajectory $\tau_i$.

**Format Reward.** A negligible reward is assigned to the trajectory if the model's output contains valid wrapping format given the task descriptions (e.g., `<think>...</think><action>...</action>`).

$$R_{\text{format}}^i = \begin{cases} 0.1, & \text{if } \mathbf{a}_t^i \text{ is wrapped correctly}, \forall (\mathbf{s}_t^i, \mathbf{a}_t^i, R_t^i) \in \tau_i \\ 0, & \text{otherwise.} \end{cases} \tag{17}$$

### A.8 BAG-OF-TRICKS FOR *Dr.BoT*

The following contents correspond to Section 4.4 in the main text.

**Removal of KL Divergence.** We follow (Yu et al., 2025; Liu et al., 2025b) to simply remove the KL divergence by setting $\beta = 0$. This allows the distribution of the LLM to diverge from the initial policy $\pi_0$ for adaptation to tool-integrated reasoning under the agent tasks.

**Clip-Higher.** We follow (Yu et al., 2025) to raise the upper clip bound $\epsilon_{\text{ub}} = 0.28$ and keep the lower bound $\epsilon_{\text{lb}} = 0.2$ as default. The decoupled lower and higher clipping range leaves more space for the increase of low-probability tokens. It relaxes the exploration of the policy which benefits premature entropy collapsing.

**Removal of Intra-group Normalization.** We follow (Liu et al., 2025b) to drop the advantage normalization term where the standard deviations lead to a difficulty bias in optimization. It has two benefits: 1) The samples with smaller intra-group standard deviations contribute more to the policy update and the removal of normalization allows balancing between samples of various difficulty; 2) The estimation of standard deviations are inaccurate for the off-policy advantage recalibration of replay samples. It is challenging to measure the sampling diversity of a specific group.

**Removal of Length Normalization.** We follow (Liu et al., 2025b) to drop the length normalization terms. We choose the token-level sum and sequence-level normalization as the aggregation approach for both loss computation and the entropy monitoring.

**Filtering of Over-long and Void-turn Samples.** We follow (Zhuang et al., 2025; Yu et al., 2025) to mask out the loss for rollout samples that exceed the predefined maximum response length. The improper reward shaping for overlong samples introduces noise into training, which causes instability of training. Besides, it prevents from test-time scaling when the context length of evaluation is longer than that of training. In addition, we mask out all the trajectories with void turns (Xue et al., 2025), where the LLM fails to call any tools in the response. Such void turns are often accompanied with the occurrence of repetitive reasoning contents, wrong chat-template formatting, and nonsensical tokens. The filtering of these void-turn samples prevents mode-collapsing where their distribution deviate severely from the initial policy.

**Filtering of Low-variance Groups.** We follow (Wang et al., 2025b) to only keep groups with high intra-group variance for each batch of training samples. The bottom $25\%$ samples with small intra-group reward standard deviations are removed to keep the policy update informative. High intra-group variance indicates diverse agent behaviors and the contrast between different actions is beneficial to exploitation.

**Regularization with Covariance-based Clipping** We introduce the covariance-based clipping (Cui et al., 2025b) to the trajectory-level entropy control. The changes of output logits that are highly associated with advantage gains greatly decrease the entropy. We remove tokens with

high covariances (Cui et al., 2025b; Wang et al., 2025a) out of loss contribution for $\tilde{\mathcal{J}}_{\text{GRPO}}^{\text{SIL-R}}(\pi_\theta)$, preventing aggressive changes of log probability for advantage acquisition.

$$\tilde{\mathcal{J}}_{\text{GRPO}}^{\text{SIL-R}}(\pi_\theta) = \mathbb{E}_{\{\tau_j\}_{j=1}^{N_\mathcal{D}} \sim \{\pi_{\theta_{\text{old}}}(\cdot|x),\, x \sim p(X)\}} \sum_{j=1}^{N_\mathcal{D}} \tilde{\mathcal{J}}_{\text{GRPO}}^j \cdot \mathbf{1}(\hat{A}_j > 0 \;\&\; \tilde{A}_j > 0) \cdot M^j, \quad (18)$$

$$M_t^j = \begin{cases} 0, & t \in I_{\text{clip}}^j, \\ 1, & t \notin I_{\text{clip}}^j, \end{cases} \quad (19)$$

$$I_{\text{clip}}^i = Ind \sim \text{Uniform}(t|\omega_{\text{lb}} \leq \text{Cov}(\log \pi_\theta(\mathbf{a}_t^i|x, \mathbf{s}_t^i), \tilde{A}_t^i) \leq \omega_{\text{ub}}, N_{\text{clip}}^i), \quad (20)$$

$$\text{Cov}(\log \pi_\theta(\mathbf{a}_t^i|x, \mathbf{s}_t^i), \tilde{A}_t^i) = (\log \pi_\theta(\mathbf{a}_t^i|x, \mathbf{s}_t^i) - \frac{1}{G}\sum_{j=1}^G \log \pi_\theta(\mathbf{a}_t^j|x, \mathbf{s}_t^j)) \cdot (\tilde{A}_t^i - \frac{1}{G}\sum_{j=1}^G \tilde{A}_t^j), \quad (21)$$

where the lower bound and upper bound for determining the range of high-covariance tokens are respectively represented as $\omega_{\text{lb}}$ and $\omega_{\text{ub}}$. The operation $\text{Uniform}(t|\cdot, N_{\text{clip}})$ refers to the uniform sampling of tokens $t$ with high covariance until a budget of $N_{\text{clip}}$ tokens. The indices of the selected tokens for loss masking are represented as $Ind$. It is noted that such masking introduces randomness which benefits the convergence of RL. The detailed settings of $\omega_{\text{lb}}$, $\omega_{\text{ub}}$, and $N_{\text{clip}}$ are subject to both the LLM and the task. We empirically set the rounded integers of the mean covariance in the range of top $20\%$ and top $0.02\%$ respectively for $\omega_{\text{lb}}$ and $\omega_{\text{ub}}$, and set $N_{\text{clip}}^i = \lambda N^i$ with $N^i$ being the total number of learnable tokens of $\tau_i$ and $\lambda$ denoting the clipping ratio.

## A.9 THEORETICAL JUSTIFICATION

**Claim 1.** *The self-imitation, with a warm-up schedule coefficient $\gamma(t_{iter})$ that increases from 0 to 1 (Eq. 5), implements a constrained projection onto the distribution of good responses, ensuring monotonic improvement of the surrogate objective.*

**Theorem 1 (Surrogate Objective Improvement Bound).** Let $\pi_{\theta_{t_{\text{iter}}}}$ be the policy at iteration $t_{\text{iter}}$, $\gamma(t_{\text{iter}}) \in [0, 1]$ the warm-up coefficient, and $r(\mathbf{a}) = \frac{\pi_{\theta_{t_{\text{iter}}+1}}(\mathbf{a})}{\pi_{\theta_{t_{\text{iter}}}}(\mathbf{a})}$ the importance weight ratio with its clipped surrogate $\tilde{r}(\mathbf{a}) = \text{clip}(r(\mathbf{a}), 1 - \epsilon, 1 + \epsilon)$. We define the good experiences for group sample $j$ as $I_j = \mathbf{1}(\hat{A}_j > 0 \;\&\; \tilde{A}_j > 0)$, where $\hat{A}_j$ and $\tilde{A}_j$ are the estimated and baseline-corrected advantages. Under the assumptions that: (1) the policy change is bounded by the clipping range, and (2) the advantage estimates are unbiased, the surrogate objective improvement satisfies:

$$\mathcal{J}(\pi_{\theta_{t_{\text{iter}}+1}}) - \mathcal{J}(\pi_{\theta_{t_{\text{iter}}}}) \geq \underbrace{\mathbb{E}_{\mathbf{a} \sim \pi_{\theta_{t_{\text{iter}}}}}\left[\tilde{r}(\mathbf{a}) \cdot A_{\pi_{\theta_{t_{\text{iter}}}}}(\mathbf{a})\right]}_{\text{GRPO improvement}} + \gamma(t_{\text{iter}}) \cdot \underbrace{\mathbb{E}_{j \sim \mathcal{D}}\left[I_j \cdot \log r(\mathbf{a}_j)\right]}_{\text{SIL improvement}} - \epsilon R_{\max},$$

$$(22)$$

where $R_{\max}$ is the maximum per-token reward, and $\mathcal{J}$ denotes the surrogate objective function.

**Proof 1.** Consider the combined objective (Eq. 5), we can decompose the total improvement by linearity:
$$\Delta\mathcal{J}_{\text{total}} = \mathcal{J}(\pi_{\theta_{t_{\text{iter}}+1}}) - \mathcal{J}(\pi_{\theta_{t_{\text{iter}}}}) = \Delta\mathcal{J}_{\text{GRPO}} + \gamma(t_{\text{iter}}) \cdot \Delta\tilde{\mathcal{J}}_{\text{GRPO}}^{\text{SIL-R}}. \quad (23)$$

The GRPO component has a lower bound from the clipped surrogate theorem (Schulman et al., 2017b):
$$\Delta\mathcal{J}_{\text{GRPO}} \geq \mathbb{E}_{\mathbf{a} \sim \pi_{\theta_{t_{\text{iter}}}}}\left[\tilde{r}(\mathbf{a}) \cdot A_{\pi_{\theta_{t_{\text{iter}}}}}(\mathbf{a})\right] - \epsilon R_{\max}. \quad (24)$$

For the self-imitation term, under the assumption of small policy changes ($\|\theta_{t+1} - \theta_t\|$ bounded), we approximate the finite difference via gradient integration:
$$\nabla_\theta \tilde{\mathcal{J}}_{\text{SIL}}^{\text{GRPO}} = \mathbb{E}_{j \sim \mathcal{D}}\left[I_j \cdot \frac{\nabla_\theta \pi_\theta(\mathbf{a}_j)}{\pi_\theta(\mathbf{a}_j)}\right]. \quad (25)$$

Using the mean value theorem and assuming smoothness of the objective, we integrate from $\theta_{t_{\text{iter}}}$ to $\theta_{t_{\text{iter}}+1}$:

$$\Delta \tilde{\mathcal{J}}_{\text{SIL}}^{\text{GRPO}} \approx \mathbb{E}_{j \sim \mathcal{D}} \left[ I_j \cdot \log \frac{\pi_{\theta_{t_{\text{iter}}+1}}(\mathbf{a}_j)}{\pi_{\theta_{t_{\text{iter}}}}(\mathbf{a}_j)} \right] = \mathbb{E}_{j \sim \mathcal{D}} \left[ I_j \cdot \log r(\mathbf{a}_j) \right]. \tag{26}$$

The coefficient $\gamma(t_{\text{iter}})$ scales the SIL contribution gradually. Combining terms yields the final bound:

$$\Delta \mathcal{J}_{\text{total}} \geq \mathbb{E}_{\mathbf{a} \sim \pi_{\theta_{t_{\text{iter}}}}} \left[ \tilde{r}(\mathbf{a}) \cdot A_{\pi_{\theta_{t_{\text{iter}}}}}(\mathbf{a}) \right] + \gamma(t_{\text{iter}}) \cdot \mathbb{E}_{j \sim \mathcal{D}} \left[ I_j \cdot \log r(\mathbf{a}_j) \right] - \epsilon R_{\max}. \tag{27}$$

Under trust region constraints, improvements in the surrogate objective $\mathcal{J}$ translate to improvements in expected return $J$ (Schulman et al., 2017b).

**Claim 2.** *The choice of median ($P_{50}$) as the baseline estimator is grounded in robust statistics and variance minimization in agentic RL with heavy-tailed return distributions.*

**Theorem 2 (Robustness to Outliers).** Let $\mathcal{R} = \{R_1, R_2, ..., R_n\}$ be a set of returns in baseline buffer $\mathcal{D}_R$. The median $P_{50}$ minimizes the expected absolute deviation and has a bounded influence function, making it robust to outliers compared to the mean.

**Proof 2.** For any estimator $b$, the loss minimization objectives are:

- Mean: $\underset{b}{\operatorname{argmin}} \sum_{i=1}^{n} (R_i - b)^2 \implies b = \frac{1}{n} \sum_i R_i$

- Median: $\underset{b}{\operatorname{argmin}} \sum_{i=1}^{n} |R_i - b| \implies b = P_{50}(\mathcal{R})$

The influence functions characterize robustness (Huber, 2011):

- Mean: $\text{IF}(R; \text{mean}) = R - \mathbb{E}[R]$ (unbounded)

- Median: $\text{IF}(R; \text{median}) = \frac{\text{sgn}(R - P_{50})}{2f(P_{50})}$ (bounded when $f(P_{50}) > 0$)

Thus, the median is robust to outliers while the mean is sensitive. This property extends to advantage estimation since advantages are linear functions of returns.

**Claim 3.** *The $P_{50}$ achieves a balance between robustness and informativeness. Comparatively, the $P_{25}$ and $P_{75}$ percentiles are either overly conservative or aggressive during advantage-based replay filtering.*

**Theorem 3 (Minimax Risk).** For the class $\mathcal{P}$ of symmetric unimodal distributions, the median minimizes the minimax risk for absolute error loss among translation-equivariant estimators:

$$\inf_{\hat{b}} \sup_{p \in \mathcal{P}} \mathbb{E}[|\hat{b} - \mu(p)|] = \sup_{p \in \mathcal{P}} \mathbb{E}[|P_{50}(X) - \mu(p)|] \tag{28}$$

**Proof 3.** This is a standard result in robust statistics (Law, 1986; Huber, 2011). For symmetric unimodal distributions, the median is minimax for absolute deviation loss among translation-equivariant estimators.

**Claim 4.** *The dual filtering mechanism using both historical advantage $\hat{A}_j$ and recalibrated advantage $\tilde{A}_j$ ensures robust policy updates and leads to better convergence properties.*

**Theorem 4 (Dual Filtering).** The combined condition $\hat{A}_j > 0$ and $\tilde{A}_j > 0$ in the SIL objective reduces the variance of gradient estimates and promotes stable policy improvement.

**Proof 4.** The dual filtering mechanism provides two benefits:

1. **Variance Reduction:** By filtering trajectories that were both historically good ($\hat{A}_j > 0$) and remain valuable under the current policy ($\tilde{A}_j > 0$), we focus on a higher-quality subset of experiences. This reduces the effective sample size but increases the signal-to-noise ratio, potentially lowering gradient variance.

2. **Stability:** The exponential decay in the probability of reusing old trajectories (Eq. 39) prevents over-reliance on outdated experiences. Under appropriate importance weighting and assuming the advantages are estimated correctly, the policy improvement follows the standard off-policy policy gradient theorem (Degris et al., 2012).

The combined filtering ensures that policy updates are based on relevant, high-quality experiences, promoting monotonic improvement under trust region constraints.

### A.10 DETAILED DATASETS AND ENVIRONMENTS

The following contents correspond to Section 5.1 in the main text.

*ALFWorld* is an interactive environment created to evaluate how well LLM agents can handle multi-step decision-making tasks. In each scenario, the agent is given a textual goal and must achieve it by engaging in multiple rounds of interaction with the environment. The platform offers 4,639 task examples spanning six typical household activity categories: Pick & Place (Pick), Examine in Light (Look), Clean & Place (Clean), Heat & Place (Heat), Cool & Place (Cool), and Pick Two & Place (Pick2).

*WebShop*, on the other hand, is a sophisticated web-based platform aimed at assessing LLM agents in authentic online shopping situations. Agents are required to interact with a simulated HTML shopping site to search for products, browse items, and purchase an appropriate product. WebShop supports a broad and varied action space, featuring more than 1.1 million products and 12K user instructions.

*DAPO-MATH-17K* is a rigorously engineered, competition-grade benchmark designed to stress-test large-scale RL on LLM agents. The agent must develop multi-step mathematical reasoning, perform strategic tool-calling for code verification, and reflect on feedback from the sandbox before submitting its final answer. It contains 17K manually-curated prompts sourced from olympiad-level problems, each transformed so that every ground-truth label is an integer—eliminating symbolic-parsing noise and yielding a clean, deterministic reward signal.

For ALFWorld, we report the average success rate for each subtask as well as the overall results. For WebShop, we report the average score and the success rate (SR).

### A.11 DETAILED BASELINES

The following contents correspond to Section 5.1 in the main text.

**ALFWorld and WebShop.** We compare with baselines such as prompting-based method (i.e., direct I/O) for the proprietary models GPT-4o (Achiam et al., 2023) and Gemini (Team et al., 2023), framework-based method such ReAct (Yao et al., 2023) and Reflexion (Shinn et al., 2023), RL methods including PPO (Schulman et al., 2017b), RLOO (Kool et al., 2019; Ahmadian et al., 2024), GRPO (Shao et al., 2024; Guo et al., 2025), GiGPO (Feng et al., 2025b), and our proposed strong baseline Dr.BoT.

**DAPO-MATH-17K.** We compare with baselines including domain-specific experts (e.g., Qwen2.5-Math (Yang et al., 2024)), existing reasoning models (e.g., Sky-T1 (Team, 2025a), o1 (Jaech et al., 2024), DeepSeek-distilled Qwen 32B (Guo et al., 2025), QwQ (Team, 2025b), and s1 (Muennighoff et al., 2025)), and the tool-integrated RL counterparts (e.g., ReTool (Feng et al., 2025a), SimpleTIR (Xue et al., 2025), ZeroTIR (Mai et al., 2025), and AFM (Li et al., 2025b)).

## A.12 IMPLEMENTATION DETAILS

The following contents correspond to Section 5.1 in the main text.

For ALFWorld and WebShop, we follow (Feng et al., 2025b) to use Qwen2.5-1.5B-Instruct and Qwen2.5-7B-Instruct (Yang et al., 2024) as our base models. For DAPO-MATH-17K, we follow (Feng et al., 2025a) to use Qwen2.5-32B-Instruct (Yang et al., 2024) for fair comparison. In addition, we use the latest Qwen3-32B-Instruct (Yang et al., 2025) for generalization studies.

The implementation of the present study is based on VeRL (Sheng et al., 2024) and its extension VeRL-Agent (Feng et al., 2025b). We use the vLLM (Kwon et al., 2023) as the inference engine during online rollout generation.

Table 7: Descriptions of the hyper-parameters for training and inference.

| Config | Explanation |
|---|---|
| train_batch_size | The batch size for training |
| val_data_size | The batch size for validation |
| ppo_mini_batch_size | The mini batch size for actor update iterations |
| ppo_max_token_len_per_gpu | The maximum number of tokens on each GPU for training |
| ppo_micro_batch_size_per_gpu | The micro batch size on each GPU for training |
| log_prob_max_token_len_per_gpu | The maximum number of tokens on each GPU for log-probability |
| log_prob_micro_batch_size_per_gpu | The micro batch size on each GPU for log-probability |
| use_dynamic_bsz | Whether to use dynamic batch size for load balance |
| ulysses_sequence_parallel_size | The sequence parallel size for training efficiency |
| tensor_model_parallel_size | The tensor parallel size of model deployment for rollout generation |
| temperature | The temperature for decoding in LLM generation |
| top_p | The top-p for decoding in LLM generation |
| n_samples_per_prompt | The number of generated samples per prompt |
| actor_learning_rate | The learning rate of the actor |
| max_epochs | The maximum number of epochs |
| num_steps | The number of steps |
| $T_{\text{warm-up}}$ | The number of steps |
| $T_{\text{decay}}$ | The number of steps |
| use_kl_in_reward | Whether to use the KL term in reward |
| kl_coef | The coefficient for the KL divergence term |
| use_kl_loss | Whether to use the KL loss |
| $\beta$ | The coefficient of the KL loss (i.e., kl_loss_coef) |
| max_prompt_length | The maximum length of input prompt |
| max_response_length | The maximum length of output generation |
| multi_turn_max_turns | The maximum number of tool-call turns |
| $\epsilon_{\text{lb}}$ | The lower bound of the policy ratio clipping (i.e., clip_ratio_low) |
| $\epsilon_{\text{ub}}$ | The upper bound of the policy ratio clipping (i.e., clip_ratio_high) |
| $N_{\mathcal{D}}$ | The replay buffer size for self-imitation learning |
| $N_{\mathcal{D}_R}$ | The baseline buffer size for storing the intra-group average performance |
| $C$ | The lower bound of the value for dual-clip PPO/GRPO (i.e., clip_ratio_c) |
| $\omega_{\text{lb}}$ | The lower bound of the covariance-based clipping |
| $\omega_{\text{ub}}$ | The upper bound of the covariance-based clipping |
| $\lambda$ | The ratio of the covariance-based clipping |
| rollout_filter_type | The type of filtering based on intra-group variance |
| rollout_filter_ratio | The ratio of filtered group |
| loss_agg_mode | The aggregation technique for loss |
| norm_adv_by_std_in_grpo | Whether to drop the advantage normalization |
| training strategy | The strategy of training (e.g., FSDP, megatron) |

### A.12.1 HYPER-PARAMETERS

We present the details of the hyper-parameter settings in the present study. Table 7 provides the definitions of the hyper-parameters used in the present study. We follow (Sheng et al., 2024) to keep most of the default empirical settings unchanged for comparability. For the covariance-based clipping, we follow (Cui et al., 2025b) to set the clipping bounds $\omega_{\text{lb}}, \omega_{\text{ub}}$ respectively as the mean value of the top 0.02% and top 2% covariance. It is noted that the token-level covariance differs from

Table 8: Hyper-parameters of ALFWorld, WebShop, DAPO-MATH, Sokoban, and SearchR1.

| Config | ALFWorld | WebShop | DAPO-MATH | Sokoban | SearchR1 |
|---|---|---|---|---|---|
| train_batch_size | 32 | 32 | 128 | 32 | 128 |
| val_data_size | | | 128 | | |
| ppo_mini_batch_size | 1024 | 256 | 32 | 64 | 32 |
| ppo_max_token_len_per_gpu | – | – | 18432 | – | – |
| ppo_micro_batch_size_per_gpu | 8 | 4 | – | 8 | – |
| log_prob_max_token_len_per_gpu | – | – | 73728 | – | 73728 |
| log_prob_micro_batch_size_per_gpu | 8 | 4 | – | 8 | – |
| use_dynamic_bsz | False | False | True | False | True |
| ulysses_sequence_parallel_size | – | – | 8 | – | 8 |
| tensor_model_parallel_size | 2 | 2 | 4 | 2 | 4 |
| temperature | 0.4 | 0.4 | 1.0 | 0.4 | 1.0 |
| top_p | 1 | 1 | 0.6 | 1 | 0.6 |
| n_samples_per_prompt | 8 | 8 | 16 | 8 | 16 |
| actor_learning_rate | | | 1e-6 | | |
| max_epochs | 200 | 350 | 1 | 200 | 20 |
| num_steps | – | – | 300 | – | 300 |
| $T_{\text{warm-up}}$ | 100 | 200 | 300 | 100 | 300 |
| $T_{\text{decay}}$ | | | 200 | | |
| use_kl_in_reward | | | False | | |
| kl_coef | | | 0 | | |
| use_kl_loss | | | False | | |
| $\beta$ | | | 0 | | |
| max_prompt_length | 2048 | 4096 | 2048 | 1024 | 2048 |
| max_response_length | 512 | 1024 | 16384/30000 | 1024 | 30000 |
| multi_turn_max_turns | 50 | 15 | 8/15 | 15 | 32 |
| $\epsilon_{\text{lb}}$ | | | 0.2 | | |
| $\epsilon_{\text{ub}}$ | | | 0.28 | | |
| $N_{\mathcal{D}}$ | | | 2048 | | |
| $N_{\mathcal{D}_R}$ | | | 10240 | | |
| $C$ | | | 10 | | |
| $\omega_{\text{lb}}$ | 2 | 2 | 1 | 2 | 1 |
| $\omega_{\text{ub}}$ | 60 | 60 | 40 | 60 | 40 |
| $\lambda$ | | | 0.02 | | |
| rollout_filter_type | | | std. | | |
| rollout_filter_ratio | | | 0.75 | | |
| loss_agg_mode | | | seq-mean-token-sum-norm | | |
| norm_adv_by_std_in_grpo | | | False | | |
| training strategy | | | FSDP | | |

task to task. Therefore, we perform statistics analysis on the covariance between action probability and the advantage with the initial model at the first training step to determine the clipping bounds.

All the settings of their values can be found in Table 8. Without loss of generalizability, we do not perform meticulous fine-tuning of the hyper-parameters. One would expect better performance with grid search for the optimal hyper-parameters.

### A.12.2 COMPUTING RESOURCES

All experiments are performed on workstations with 380 CPU cores, 2.2TB memory, and 8 GPUs of 96GB memory. For both 1.5B/7B LLMs and 3B VLMs, the training is performed on four workstations with 32 GPUs in total. For the 32B models, the training is performed on sixteen workstations with 128 GPUs in total.

For ALFWorld, Webshop, and Sokoban, it takes less than 60 hours for optimization of 1.5B and 7B models. While for the DAPO-MATH-17K, it takes around a week for training the 32B models.

### A.13 DISCUSSIONS ON HYPER-PARAMETERS

The following contents are mentioned in Section 5.6 in the main text.

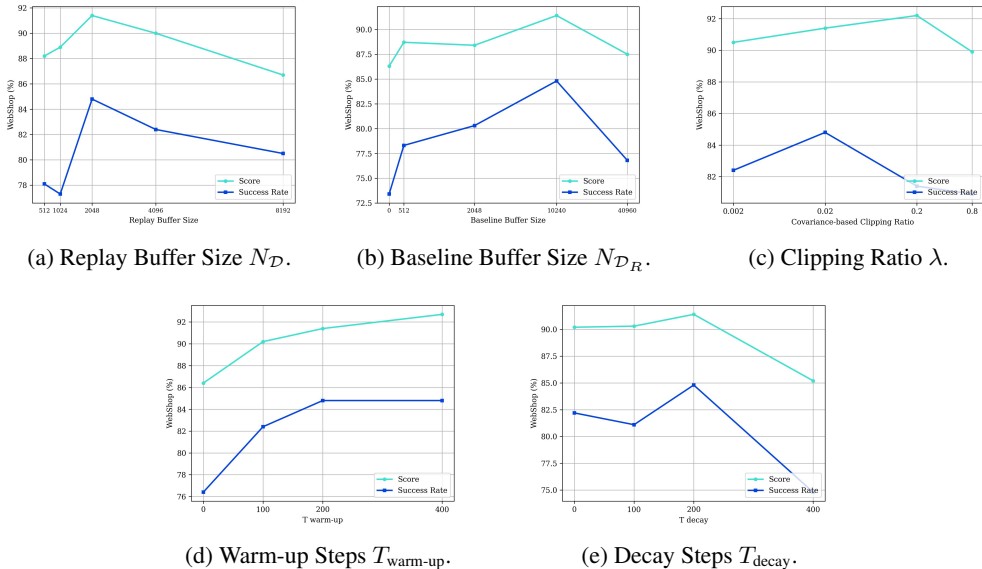

(a) Replay Buffer Size $N_{\mathcal{D}}$.    (b) Baseline Buffer Size $N_{\mathcal{D}_R}$.    (c) Clipping Ratio $\lambda$.

(d) Warm-up Steps $T_{\text{warm-up}}$.    (e) Decay Steps $T_{\text{decay}}$.

Figure 7: Effect of hyper-parameters of *Dr.BoT* (GRPO) with SPEAR on WebShop (Qwen2.5-7B-Instruct).

### A.13.1 EFFECT OF HYPER-PARAMETERS

We investigate the following key hyper-parameters (see Figure 7) of *Dr.BoT* (GRPO) with SPEAR on WebShop (Qwen2.5-7B-Instruct) while keeping the value of others fixed (see Table 8).

**Replay Buffer Size $N_{\mathcal{D}}$.**    As the buffer size increases, the performance first improves due to the improved diversity and impact of the collected trajectories in the buffer. However, when the buffer continues to expand, trajectories in the buffer might come from earlier training batches and thereafter causes more severe degree of off-policy. The self-imitation of excessively outdated experiences becomes detrimental to the update of current policy. In addition, the large replay buffer takes more iterations to refill and thereafter the policy update frequency from self-imitation is lower than that of a smaller buffer, further diminishing its intervention in agent exploration.

**Baseline Buffer Size** $N_{\mathcal{D}_R}$**.** When $N_{\mathcal{D}_R} = 0$, the original advantages are used without recalibration and filtering (see Equation 3). It shows that the direct imitation of these experiences can be suboptimal where certain trajectories are outdated for the current policy. By timely adjusting the advantages and removing inappropriate experiences ($\tilde{A}_j \leq 0$), we reduce the inaccurate estimation for off-policy update. It is noted that using advantage rather than reward in the baseline buffer helps mitigate learning bias, as it allows for contributions from samples with negative rewards as long as there is variance within a group. The removal of the standard deviation of outcome rewards is crucial for reducing difficulty bias. Furthermore, our double-positive advantage gate for replay filtering is essential for off-policy learning. We also find that $N_{\mathcal{D}_R}$ should not be set too large as such 50-th percentile reward deviates from the latest ones, decreasing the effectiveness of recalibration.

**Covariance-based Clipping Ratio** $\lambda$**.** The clipping ratio can be viewed as the degree of regularization for policy entropy, where a larger ratio causes more tokens to be ignored during policy update. In this case, the contribution of self-imitation gets weakened. A modest range of clipping ratio (e.g., $0.0002 \sim 0.02$) not only suffices the entropy management but also allows proper exploitation of the collected experiences.

**Warm-up Step** $T_{\text{warm-up}}$**.** A smaller warm-up step implies earlier self-imitation of the premature, suboptimal experiences during RL. Especially when the distribution of the task and environment differs greatly from the pre-trained knowledge, the overfitting of the initial trajectories hinders exploration of low-probable solutions and leads to action-level local optima. Intuitively, $T_{\text{warm-up}}$ can be first set the same as the total number of training steps and then adjusted according to the task and the model for the improved performance.

**Decay Step** $T_{\text{decay}}$**.** A smaller decay step reduces the stimulation from the intrinsic reward for acquisition of tool-use skills. If the LLM already excels at interacting with the environment (e.g., use of tools and comprehension of observations), $T_{\text{decay}}$ can be set close to 0. A large $T_{\text{decay}}$ is not encouraged as the interference with the outcome reward causes inconsistent policy optimization for convergence.

### A.13.2 GUIDELINES ON HYPER-PARAMETERS TUNING

In this section, we provide guidelines on the choice of these hyper-parameters for practical usage. It is noted that most of the hyper-parameters share the same value settings across benchmarks of various domains and tasks. One would expect performance gains without meticulous fine-tuning.

**Replay Buffer Size** $N_{\mathcal{D}}$**.** It should not be set too large to avoid severe off-policy deviation. A modest size of $2K \sim 4K$ proportional to the training batch size of 128 and group size of 16 ($128 \times 16$) is expected to work well for frequent refilling and policy update. In other word, $N_{\mathcal{D}}$ cam be set as 2x/4x of train_batch_size $\times$ n_samples_per_prompt.

**Baseline Buffer Size** $N_{\mathcal{D}_R}$**.** An appropriate setting between 2K and 10K prevents outdated and untimely estimation of current policy baseline performance. In other word, $N_{\mathcal{D}_R}$ can be set as 1x/4x of $N_{\mathcal{D}}$.

**Covariance-based Clipping Ratio** $\lambda$**.** The percentage of clipped tokens should be controlled between 0.02% and 2%. A smaller percentage would reduce the effect of anti-overfitting while a larger percentage slows down the policy exploitation of experiences.

**Warm-up Step** $T_{\text{warm-up}}$**.** The self-imitation should be scheduled to reach its maximum after 200 steps. For difficult tasks, it should be increased to allow exploration of diverse trajectories without convergence to local sub-optimum. One could first try $T_{\text{warm-up}} = \text{num\_steps}$.

**Decay Step** $T_{\text{decay}}$**.** A decay step between 100 and 200 would be sufficient. If the tool is hard to master (e.g., complex slot-filling), the decay step should be increased to allow more stimulation of tool-calling behaviors. One could first try $T_{\text{decay}} = \text{num\_steps}$.

## A.14 QUALITATIVE ANALYSIS

The following contents are mentioned in Section 5.6 in the main text.

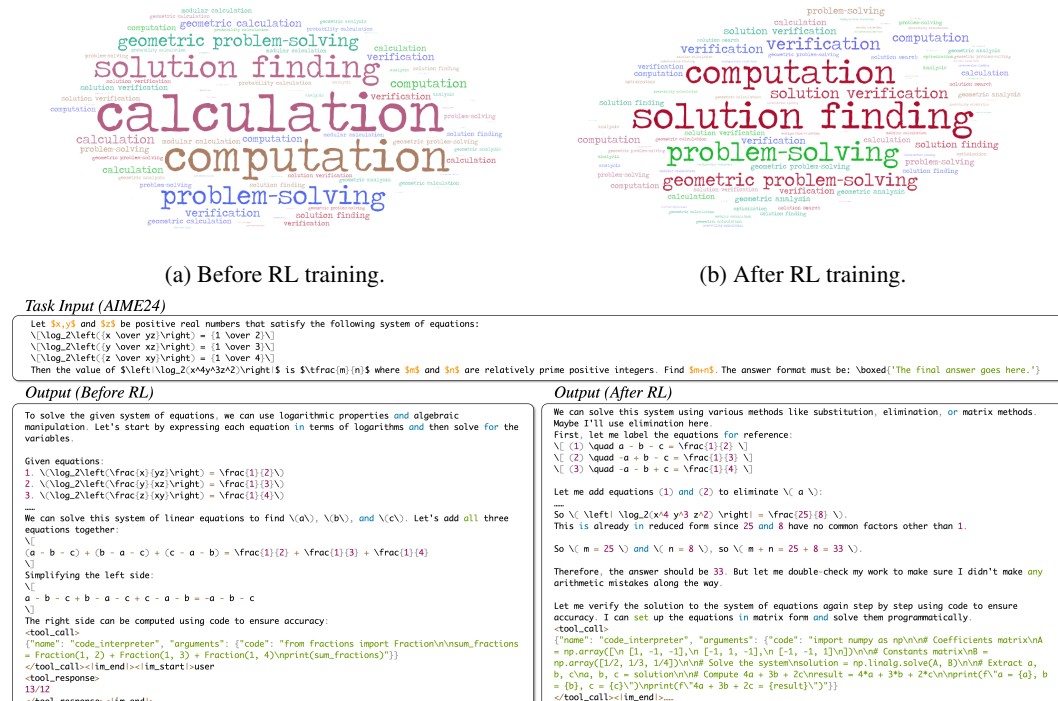

(a) Before RL training.        (b) After RL training.

(c) The evolution of efficient coding from the purpose of computation to verification (best viewed magnified).

Figure 8: Development of the agent's coding skills.

### A.14.1 TOOL-INTEGRATED REASONING

**Skill Development.** We follow (Feng et al., 2025a) to analyze the coding capabilities of the agent before and after RL by classifying the purpose of the code snippets. Specifically, we employ Hunyuan-Large (Sun et al., 2024) to interpret reasoning contexts before each tool-calling and judge the intention of the codes passed into the code interpreter on DAPO-MATH-17K dataset. The external LLM first performs intent classification with open-ended categories in a free-form manner. Then, we manually deduplicate these categories and only keep the top 20 frequent ones: *calculation, computation, solution finding, problem-solving, geometric problem-solving, verification, geometric calculation, solution verification, modular calculation, probability calculation, geometric analysis, analysis, optimization, intersection finding, solution search, function analysis, configuration validation, data computation, game analysis, data processing, game strategy analysis, solution exploration, data analysis, list validation*. We further use the LLM to classify each code snippet into at most three categories. The agreement between the LLM and manual classification is above 90% on 50 randomly chosen samples. The code purposes with their frequency over twice are kept and illustrated in Figure 8. We find that the after RL, the agent becomes proficient in writing codes for solution finding, problem-solving, and solution verification, which is quite advanced compared with the basic calculation and computation before RL. Our case study shows that after training, the agent learns to master matrix computation with `numpy` for higher efficiency, confirming the improvement of coding skills.

**Action Refinement.** As shown in Figure 9, the agent initially aims at finding the target product that satisfies all the constraints simply by searching. However, such continuous choice of the action `search` is trapped by the unqualified retrieval results. The attributes of product such as color and size should be determined only at the product page. After RL, the agent jumps out of the

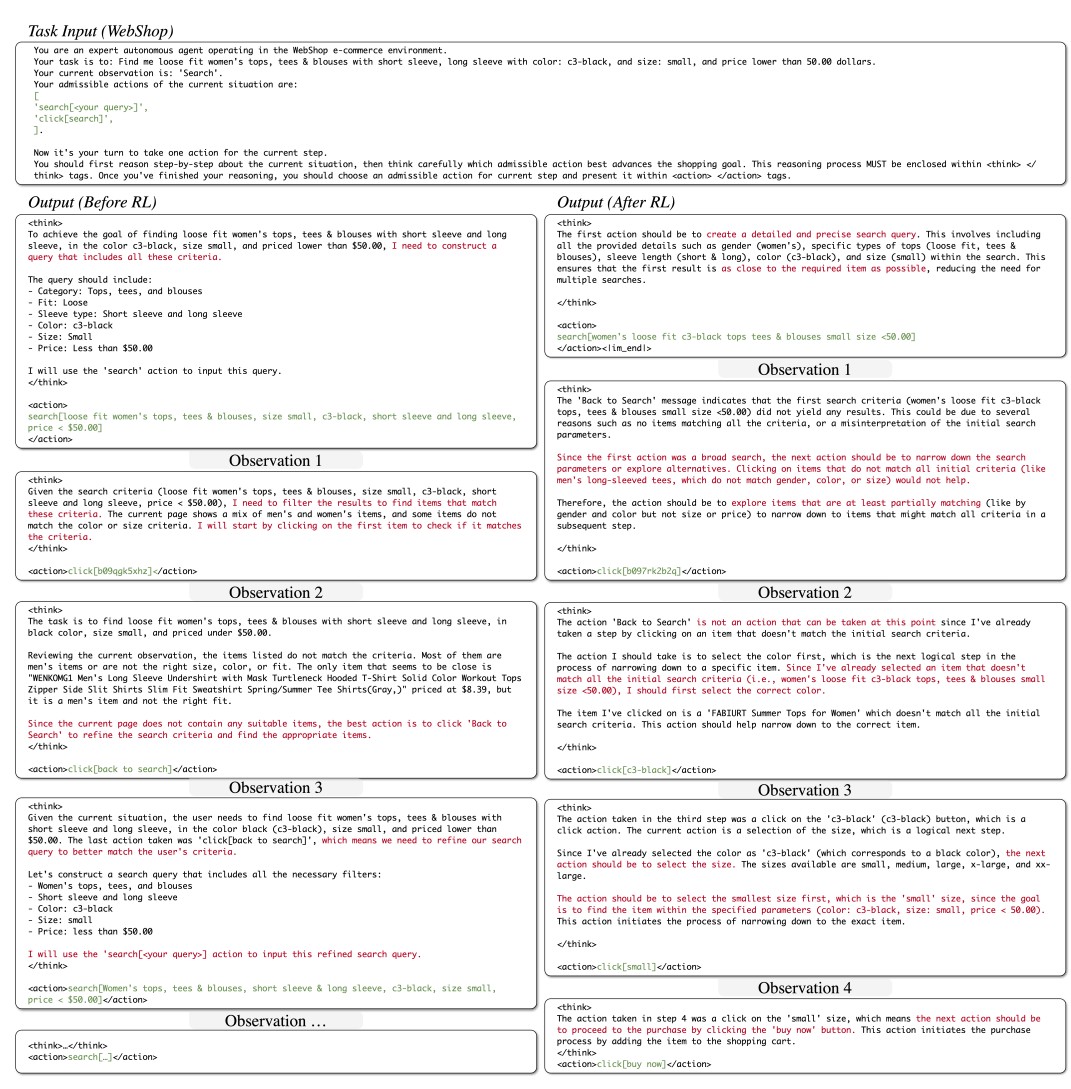

Figure 9: The advancement of strategy from the search query perfectionism to goal-oriented active progression (best viewed magnified).

perfectionism for the search queries and tries to break the task step by step. It learns to choose the action wisely for persistent focus on the task.

### A.14.2 CONTEXT BUDGET

Table 9: The number of tool call turns and response length of SPEAR on Qwen2.5-32B and Qwen3-32B under 16K and 32K context budgets, respectively.

| Model | # Turns@16K | Responses Len@16K | # Turns@32K | Responses Len@32K |
|---|---|---|---|---|
| Qwen2.5-32B | 7.18 | 4855.48 | 7.13 | 7502.59 |
| Qwen3-32B | 3.23 | 10522.38 | 4.43 | 12371.95 |

In this section, we provide more analysis on the differences of reasoning behaviors between 16K and 32K token contexts. Table 9 shows that for Qwen2.5 models, the number of tool call turns does not increase abruptly from 16K to 32K. Two reasons are possible: 1) The tool call reward (Eq. 16) allows a maximum of 1 score which corresponds to 10 tool calls. More tool calls (¿10) will not be rewarded. 2) The intrinsic reward design is targeted at stimulating exploration at the beginning and the dominance of outcome reward is guaranteed via scheduling. The mechanical increase of tool use for reward hacking will be penalized to promote reasoning for accuracy. For Qwen3-32B, the number of tool calls increases a bit but still falls behind that of Qwen2.5-32B. This is because the Qwen3 series are reasoning models and tend to develop sophisticated solution patterns via pure text. In this case, the agent mainly uses the tool to double-check its previous textual reasoning and computation. The context budget from 16K to 32K allows 2K more response tokens and accordingly follows one or two more rounds of tool calls for verification.

Examples on the reasoning patterns of Qwen2.5 and Qwen3 under 16K and 32K contexts are respectively provided in Figures 10 11 12. We randomly choose one sample from the AIME 24 benchmark. It shows that for both Qwen2.5 and Qwen3 models, the number of tool calls does not increase drastically, which is consistent with the Table 9. We believe the AIME benchmarks are of reasoning-heavy tasks which challenge the agent the most its complex reasoning capabilities. In this case, our SPEAR balances the tool call frequency and the final outcome by: 1) stimulating exploration at an early stage with a bounded tool call reward (maximum of 1), and 2) guaranteeing dominance of the outcome reward via scheduled adjustment. Such design prevents the agent from hacking reward simply by frequent tool calling. Instead, the agent learns to reason deeply in text, and uses the tool observation as feedback to cross-validate its previous deduction and computation. The increased context budget allows longer thinking and reflection process, leading to performance gains.

### A.14.3 ADDITIONAL ENTROPY MEASUREMENTS

Figure 13 illustrates the variance of entropy of the proposed *Dr.BoT* with and without SPEAR. We can observe that:

1) For most tasks and model scales, the policy entropy of the vanilla *Dr.BoT* does not converge. This is in line with our findings in Figure 3 where the consistent uncertainty about the environments and actions causes policy entropy divergence.

2) Due to the curriculum scheduling of self-imitation, the policy entropy maintains a steady trend across stages. The SPEAR allows sufficient exploration at the beginning and gradually strengthens imitation of self-generated promising experience. Therefore, the entropy varies smoothly during training.

3) Due to the curriculum scheduling of tool call reward, the interaction with the environments is encouraged and therefore the policy entropy of SPEAR can even surpass the baseline (e.g., ALFWorld 1.5B and WebShop 1.5B). However, such exploration about the environment does not necessarily correlate with entropy variation. We believe the distributional gap between task domains and the pretrained knowledge of LLMs plays a critical role. For larger models (7B), its internal parameterized knowledge is richer to handle the observation states properly.

*Task Input (AIME24)*

Let $x,y$ and $z$ be positive real numbers that satisfy the following system of equations:
$$\log_2\left(\frac{x}{yz}\right) = \frac{1}{2}$$
$$\log_2\left(\frac{y}{xz}\right) = \frac{1}{3}$$
$$\log_2\left(\frac{z}{xy}\right) = \frac{1}{4}$$
Then the value of $\left|\log_2(x^4y^3z^2)\right|$ is $\tfrac{m}{n}$ where $m$ and $n$ are relatively prime positive integers. Find $m+n$. The answer format must be: \boxed{'The final answer goes here.'}

*Output (Qwen2.5 @16K)* `1333 Tokens; 2 Tool-calls`

To solve the given system of logarithmic equations, we can start by converting each logarithmic equation into an exponential form. The given equations are:

1 $(\log_2\left(\frac{x}{yz}\right) = \frac{1}{2})$
2 $(\log_2\left(\frac{y}{xz}\right) = \frac{1}{3})$
3 $(\log_2\left(\frac{z}{xy}\right) = \frac{1}{4})$

We can rewrite each equation in terms of exponents:

1 $(\frac{x}{yz} = 2^{1/2})$
2 $(\frac{y}{xz} = 2^{1/3})$
3 $(\frac{z}{xy} = 2^{1/4})$

Let's express each equation in a more manageable form:

1 $(x = 2^{1/2} \cdot yz)$
2 $(y = 2^{1/3} \cdot xz)$
3 $(z = 2^{1/4} \cdot xy)$

We can substitute these expressions into each other to solve for $(x)$, $(y)$, and $(z)$. However, this might get complicated algebraically. Alternatively, we can take the logarithm base 2 of each variable and solve the resulting system of linear equations.

Let's set $(a = \log_2 x)$, $(b = \log_2 y)$, and $(c = \log_2 z)$. Then we can rewrite the original equations in terms of $(a)$, $(b)$, and $(c)$:

1 $(\log_2 x - \log_2 y - \log_2 z = \frac{1}{2})$ which becomes $(a - b - c = \frac{1}{2})$
2 $(\log_2 y - \log_2 x - \log_2 z = \frac{1}{3})$ which becomes $(b - a - c = \frac{1}{3})$
3 $(\log_2 z - \log_2 x - \log_2 y = \frac{1}{4})$ which becomes $(c - a - b = \frac{1}{4})$

So we have the following system of linear equations:

1 $(a - b - c = \frac{1}{2})$
2 $(-a + b - c = \frac{1}{3})$
3 $(-a - b + c = \frac{1}{4})$

We can solve this system of equations using code to find the values of $(a)$, $(b)$, and $(c)$. `<tool_call> {"name": "code_interpreter", "arguments": {"code": "from sympy import symbols, Eq, solve\n\na, b, c = symbols('a b c')\n\neq1 = Eq(a - b - c, 1/2)\neq2 = Eq(-a + b - c, 1/3)\neq3 = Eq(-a - b + c, 1/4)\n\nsolution = solve((eq1, eq2, eq3), (a, b, c))\nprint(solution)"}} </tool_call><|im_end|><|im_start|>user <tool_response> {a: -0.291666666666667, b: 0.375000000000000, c: -0.416666666666667} </tool_response><|im_end|> <|im_start|>assistant` The solution to the system of equations is:

$[ a = -\frac{7}{24}, \quad b = -\frac{3}{8}, \quad c = -\frac{5}{12} ]$

Now, we need to find the value of $(\left|\log_2(x^4 y^3 z^2)\right|)$. Using the properties of logarithms:

$[ \log_2(x^4 y^3 z^2) = 4a + 3b + 2c ]$

Let's substitute the values of $(a)$, $(b)$, and $(c)$ into this expression and compute the result. `<tool_call> {"name": "code_interpreter", "arguments": {"code": "a_val = -7/24\nb_val = -3/8\nc_val = -5/12\nresult = 4 * a_val + 3 * b_val + 2 * c_val\nprint(result)\n\n# Convert to fraction to find m and n\nfrom fractions import Fraction\nfrac_result = Fraction(result).limit_denominator()\nprint(frac_result)"}} </tool_call><|im_end|><|im_start|>user <tool_response> -3.1250000000000004 -25/8 </tool_response><|im_end|> <|im_start|>assistant` The value of $(4a + 3b + 2c)$ is $(-\frac{25}{8})$. Therefore, the absolute value is $(\left| -\frac{25}{8} \right| = \frac{25}{8})$. The fraction $(\frac{25}{8})$ is already in its simplest form since 25 and 8 are relatively prime. Thus, $(m = 25)$ and $(n = 8)$, so $(m + n = 25 + 8 = 33)$.

Therefore, the final answer is 33.

\boxed{33}<|im_end|>

*Output (Qwen2.5 @32K)* `1987 Tokens; 3 Tool-calls`

To solve the given system of equations involving logarithms, we can start by converting the logarithmic equations into exponential form. Let's first rewrite each equation:

1 $(\log_2\left(\frac{x}{yz}\right) = \frac{1}{2})$
2 $(\log_2\left(\frac{y}{xz}\right) = \frac{1}{3})$
3 $(\log_2\left(\frac{z}{xy}\right) = \frac{1}{4})$

We can convert each logarithmic equation into its exponential form:

1 $(\frac{x}{yz} = 2^{1/2})$
2 $(\frac{y}{xz} = 2^{1/3})$
3 $(\frac{z}{xy} = 2^{1/4})$

Now, let's express each equation in terms of $(x, y,)$ and $(z)$:

1 $(x = yz \cdot 2^{1/2})$
2 $(y = xz \cdot 2^{1/3})$
3 $(z = xy \cdot 2^{1/4})$

Next, we can substitute these equations into each other to solve for $(x, y,)$ and $(z)$. Alternatively, we can use the logarithmic properties to solve for $(x, y,)$ and $(z)$ more directly. Let's take the original logarithmic equations and express them in terms of $(\log_2 x)$, $(\log_2 y)$, and $(\log_2 z)$.

Given:

1 $(\log_2\left(\frac{x}{yz}\right) = \frac{1}{2})$
2 $(\log_2\left(\frac{y}{xz}\right) = \frac{1}{3})$
3 $(\log_2\left(\frac{z}{xy}\right) = \frac{1}{4})$

Using logarithm properties, we can rewrite each equation as follows:

1 $(\log_2 x - \log_2 y - \log_2 z = \frac{1}{2})$
2 $(\log_2 y - \log_2 x - \log_2 z = \frac{1}{3})$
3 $(\log_2 z - \log_2 x - \log_2 y = \frac{1}{4})$

Let's denote $(a = \log_2 x)$, $(b = \log_2 y)$, and $(c = \log_2 z)$. Then, we can rewrite the system of equations in terms of $(a)$, $(b)$, and $(c)$:

1 $(a - b - c = \frac{1}{2})$
2 $(-a + b - c = \frac{1}{3})$
3 $(-a - b + c = \frac{1}{4})$

Now, we have a system of linear equations:

1 $(a - b - c = \frac{1}{2})$
2 $(-a + b - c = \frac{1}{3})$
3 $(-a - b + c = \frac{1}{4})$

We can solve this system of equations to find $(a)$, $(b)$, and $(c)$. Then, we can find $(\left|\log_2(x^4 y^3 z^2)\right|)$:

$[ \log_2(x^4 y^3 z^2) = 4a + 3b + 2c ]$

Let's use Python to solve the system of equations accurately. `<tool_call> {"name": "code_interpreter", "arguments": {"code": "from sympy import symbols, Eq, solve, Rational\n\na, b, c = symbols('a b c')\n\neq1 = Eq(a - b - c, Rational(1, 2))\neq2 = Eq(-a + b - c, Rational(1, 3))\neq3 = Eq(-a - b + c, Rational(1, 4))\n\nsolution = solve((eq1, eq2, eq3), (a, b, c))\n\na_val = solution[a]\nb_val = solution[b]\nc_val = solution[c]\n\nresult = 4*a_val + 3*b_val + 2*c_val\nabs_result = abs(result)\n\nprint(abs_result)"}} </tool_call><|im_end|><|im_start|>user <tool_response> 25/8 </tool_response><|im_end|><|im_start|>assistant` So, the value of $(\left|\log_2(x^4 y^3 z^2)\right|)$ is $(\frac{25}{8})$. The problem states that this value is $(\frac{m}{n})$ where $(m)$ and $(n)$ are relatively prime positive integers. Here, $(m = 25)$ and $(n = 8)$, and since 25 and 8 are relatively prime, we can find $(m + n)$:

$[ m + n = 25 + 8 = 33 ]$

Let's verify this result using Python to ensure accuracy. `<tool_call> {"name": "code_interpreter", "arguments": {"code": "from sympy import symbols, log, Rational, simplify, Abs\n\nx, y, z = symbols('x y z', positive=True, real=True)\n\neq1 = log(x / (y * z), 2) - Rational(1, 2)\neq2 = log(y / (x * z), 2) - Rational(1, 3)\neq3 = log(z / (x * y), 2) - Rational(1, 4)\n\nsolution = solve(eq1, eq2, eq3), (x, y, z))\n\nx_val = solution[x]\ny_val = solution[y]\nz_val = solution[z]\n\nexpression = log(x4 * y3 * z**2, 2)\nresult = Abs(expression.subs({x: x_val, y: y_val, z: z_val}))\n\nprint(result)"}} </tool_call><|im_end|><|im_start|>user <tool_response> solution = solve(eq1, eq2, eq3), (x, y, z)) NameError: name 'solve' is not defined Errors occurred! Check your code. </tool_response><|im_end|> <|im_start|>assistant` It seems there was an error because the `solve` function wasn't recognized in that context. Let me correct that by properly using SymPy's solving capabilities. Let's re-solve the equations step by step again directly in Python:

1 Convert the logarithmic equations to their exponential forms.
2 Solve for $(x)$, $(y)$, and $(z)$.
3 Substitute into the expression $(\left|\log_2(x^4 y^3 z^2)\right|)$. `<tool_call> {"name": "code_interpreter", "arguments": {"code": "import sympy as sp\n\nx, y, z = sp.symbols('x y z', positive=True, real=True)\n\neq1 = sp.log(x / (y * z), 2) - sp.Rational(1, 2)\neq2 = sp.log(y / (x * z), 2) - sp.Rational(1, 3)\neq3 = sp.log(z / (x * y), 2) - sp.Rational(1, 4)\n\nsolutions = sp.solve([eq1, eq2, eq3], (x, y, z), dict=True)\n\nsolution = solutions[0]\n\nexpression = sp.log(x4 * y3 * z**2, 2)\nresult = sp.Abs(expression.subs(solution))\n\nresult_simplified = sp.simplify(result)\n\nprint(result_simplified)"}} </tool_call><|im_end|><|im_start|>user <tool_response> 25/8 </tool_response><|im_end|> <|im_start|>assistant` The result is confirmed to be $(\frac{25}{8})$, so $(m = 25)$ and $(n = 8)$, and $(m + n = 25 + 8 = 33)$.

Thus, the final answer is $(\boxed{33})$.

$(\boxed{33})$<|im_end|>

Figure 10: The qualitative results of Qwen2.5 reasoning between 16K and 32K contexts.

*Task Input (AIME24)*

Let $x,y$ and $z$ be positive real numbers that satisfy the following system of equations:
$$\log_2\left({x \over yz}\right) = {1 \over 2}$$
$$\log_2\left({y \over xz}\right) = {1 \over 3}$$
$$\log_2\left({z \over xy}\right) = {1 \over 4}$$
Then the value of $\left|\log_2(x^4y^3z^2)\right|$ is $\tfrac{m}{n}$ where $m$ and $n$ are relatively prime positive integers. Find $m+n$. The answer format must be: \boxed{'The final answer goes here.'}

*Output (Qwen3 @16K)* **4140 Tokens; 3 Tool-calls**

"<think>\nOkay, let's try to solve this problem step by step using multiple python code calls. \n\nSo, we have three equations involving logarithms with base 2. The variables x, y, z are positive real numbers. The equations are:\n\nlog₂(x/(y z)) = 1/2,\nlog₂(y/(x z)) = 1/3,\nlog₂(z/(x y)) = 1/4.\n\nWe need to find |log₂(x⁴ y³ z²)| and express it as m/n where m and n are coprime, then find m + n.\n\nFirst, I remember that logarithms can be converted into exponents. For example, log₂(A) = B means A = 2^B. Maybe converting each equation into exponential form would help.\n\nLet me rewrite each equation:\n\n1. x/(y z) = 2^{1/2}\n2. y/(x z) = 2^{1/3}\n3. z/(x y) = 2^{1/4}\n\nHmm. So these equations relate x, y, z. Maybe we can express each variable in terms of others and substitute?\n\nAlternatively, since all equations involve x, y, z in fractions, maybe taking the product of all three equations could simplify things. Let me try that.\n\nMultiplying left sides and right sides:\n\n(x/(y z)) * (y/(x z)) * (z/(x y)) = 2^{1/2} * 2^{1/3} * 2^{1/4}\n\nSimplify left side:\n\nLet me compute numerator and denominator.\n\nNumerator: x * y * z.\n\nDenominator: (y z) * (x z) * (x y) = y z x z x y = x² y² z².\n\nSo overall, (x y z) / (x² y² z²) = 1/(x y z).\n\nTherefore left side is 1/(x y z) equals right side which is 2^{(1/2 + 1/3 + 1/4)}.\n\nLet me compute the exponent sum:\n\n1/2 + 1/3 + 1/4. Common denominator is 12: 6/12 +4/12 +3/12=13/12.\n\nWhich implies x y z = 2^{-13/12}\n\nHmm. Interesting. So the product xyz is known. Now, perhaps we can find individual variables?\n\nAlternatively, maybe express each equation as equations involving log base 2 of x, y, z. Let me denote:\n\nLet a = log₂ x, b = log₂ y, c = log₂ z. Since x, y, z are positive reals, this substitution is valid.\n\nOur target is to find |log₂(x⁴ y³ z²)| = |4 log₂ x + 3 log₂ y + 2 log₂ z| = |4a + 3b + 2c|.\n\nSo if we can find a, b, c, then compute this expression.\n\nNow let's rewrite the original equations using a, b, c.\n\nOriginal equations:\n\nFirst equation: log₂(x/(y z)) = 1/2.\n\nlog₂ x - log₂(y z) = 1/2.\n\nWhich is a - [log₂ y + log₂ z] = 1/2\n\na + (b + c) = 1/2.\n\nSimilarly for other equations:\n\nSecond equation: log₂(y/(xz)) = 1/3 → b - (a + c) = 1/3.\n\nThird equation: log₂(z/(xy)) + c - (a + b) = 1/4.\n\nSo now we have three equations:\n\n1. a - b - c = 1/2\n\n2. -a + b - c = 1/3\n\nWait, wait: second equation is b - (a + c) = 1/3 → -a + b - c = 1/3?\n\nWait.\n\nOriginal equation: b - a - c = 1/3 → -a + b - c = 1/3. Yes.\n\nThus similarly third equation:\n\nc - a - b = 1/4 → -a - b + c = 1/4.\n\nSo our system is:\n\nEquation 1: a - b - c = 1/2 \nEquation 2: -a + b - c = 1/3 \nEquation 3: -a - b + c = 1/4 \n\nNow we can write this system in terms of variables a, b, c.\n\nLet me write them:\n\n1. a - b - c = 1/2 \n2. -a + b - c = 1/3 \n3. -a - b + c = 1/4 \n\nHmm. Let me think how to solve this system.\n\nLet me denote:\n\nEquation (1): a - b - c = 1/2 \nEquation (2): -a + b - c = 1/3 \nEquation (3): -a - b + c = 1/4 \n\nWe can try adding equations to eliminate variables.\n\nFor example, adding equations (1) and (2):\n\n(a - b - c) + (-a + b - c) = 1/2 + 1/3 \nLeft side: (a -a) + (-b +b) + (-c -c) = -2c \nSo -2c = 5/(12)\n\nWait, 1/2 +1/3 is 5/6? 1/2 is 3/6 and 1/3 is 2/6. Sum is 5/6 yes.\n\nSo c = -5/(12)\n\n(-2) ?\n\nWait:\n\n-2c = 5/6 → c = -5/(12) ?\n\nWait.\n\nSolving for -2c = 5/6 → c = -(5/6)*2 = -5/12. Yes.\n\nWait.\n\nSo now -2c = 5/6 → c = -(5/6)*(1/2) → c = -5/12.\n\nHmm.\n\nWait:\n\nSo now -2c = 5/6 + c = -(5/6)*1/2?\n\nWait.\n\nLet me redo this. From equation (1)+(2): -2c = 5/6.\n\nSo c = -5/12.\n\nYes.\n\nNow let's do equation (1) and (3): a - b - c = 1/2 and -a - b + c = 1/4. Adding:\n\n-2b = 3/4 → b = -3/(8)\n\nWhich is\n\n1/2 +1/4 is 3/4.\n\nYes. So -2b = 3/4 → b = -3/(8). Correct.\n\nSimilarly, adding equations (2) and (3): -a + b - c = 1/3 and -a - b + c = 1/4\n\nAdding:\n\n-2a = 1/3 + 1/4\n\n7/12 → a = 7/(24)\n\nWait, no: -2a = 7/(24)? No, 1/3 + 1/4 is 4/12 +3/12 = 7/12. Sum is 7/12.\n\nTherefore:\n\na = -(7/12)*(1/2) = -7/24.\n\nWait, no: -2a = 7/12 → a = -(7/24).\n\nWait, so: na = -7/24, nb = -3/8 = -9/24\n\nWait, -3/8 = -9/24? Let me check:\n\nn-3/8 = (-9)/24?\n\nBecause multiplying numerator and denominator by 3: denominator becomes 24.\n\nYes. So -3/8 = -9/24.\n\nSimilarly c = -5/12 = -10/24.\n\nWait, that's interesting. So let me confirm.\n\nYes. So:\n\nna = -7/24, nb = -3/8 = -9/24, nc = -5/12 = -10/24.\n\nWait, because 5/12 is 10/24. So\n\nnow we can compute the target expression: 4a +3b +2c.\n\nLet me compute each term:\n\n4a = 4*(-7/24) = -28/24\n\n3b = 3*(-9/24)\n\n= -27/24\n\n2c = 2*(-10/24)\n\n= -20/24\n\nAdding them up: (-28 -27 -20)/24 = (-75)/24\n\nSimplify that fraction:\n\nDivide numerator and denominator by 3: -75 ÷ -25; 24 ÷3=8?\n\nNo, 24 divided by 3 is 8?\n\n3*8=24 yes. Wait no: 24 +3 is 8? Wait 3*8 is 24. Yes. Wait but\n\n75 +3 is 25. So 75 is -25/8? Wait.\n\nHmm.\n\nOriginal sum is -75/24 = -(75/24)\n\n= -(25/8) since 75 +3 is 25, 24 +3 is 8. Yes. \n\nSo absolute value is | -25/8 | = 25/8.\n\nWhich is already reduced since 25 and 8 are coprime. So m is 25, n is 8, m +n is 33.\n\nWait but let me check my calculations again. Because maybe I made an error in solving equations. Let me check the steps again.\n\nFirst solving the equations:\n\nEquation (1): a - b - c = 1/2 \nEquation (2): -a + b - c = 1/3 \nEquation (3): -a - b + c = 1/4 \n\nWe added equations 1 and 2 to get -2c = 5/6 c=-5/12.\n\nAdded equations 1 and 3 to get -2b = 3/4 → b=-3/8.\n\nAdded equations 2 and 3 to get -2a = 7/12.\n\nSo substituting into 4a +3b +2c\n\n+2c:\n\nCompute 4*(-7/24) = (-28)/24\n\n3*(-3/8) = (-9/8)\n\n(-27/24) \nin n2*(-5/12) = (-10)/12 = (-20)/24\n\nTotal: (-28 -27\n\n-20)/24 = (-75)/24 → -25/8 ✕ absolute value is 25/8. \n\nYes. 25 and 8 are coprime. So m +n is 25+8=33. \n\nBut to confirm, maybe I can use code to solve the system of equations. Let me use sympy in Python to check.\n\nLet me write code for solving the equations. \n\nFirst, define variables a, b, c. Then set up the equations. \n\nSo:\n\nEquation1: a - b - c - 1/2 =0 \nEquation2: -a + b - c - 1/3 =0

*Output (Qwen3 @32K)* **6462 Tokens; 3 Tool-calls**

"<think>\nOkay, let me try to solve this problem step by step using multiple python code calls. The problem gives me three logarithmic equations involving x, y, and z. My goal is to find the absolute value of log base 2 of (x⁴ y³ z²) and express it as a fraction m/n where m and n are coprime, then find m + n. \n\nFirst, let me write down the equations again to make sure I have them right:\n\n1. log₂(x/(y z)) = 1/2.\n2. log₂(y/(x z)) = 1/3.\n3. log₂(z/(x y)) = 1/4.\n\nSince all variables are positive real numbers, I don't have to worry about undefined logs or negative arguments. \n\nI remember that logarithms can be converted into exponents. For example, log₂(A) = B is equivalent to 2^B = A. Maybe converting these equations into exponential form will help me find relations between x, y, and z. Let me try that for each equation. \n\nStarting with the first equation: \n\nlog₂(x/(y z)) = 1/2 → x/(y z) = 2^{1/2} = √2. \n\nSimilarly for the second: \n\nlog₂(y/(x z)) = 2^{1/3}. \n\nlog₂(z/(x y)) = 2^{1/4}. \n\nNow, these three equations involve ratios of x, y, z. Maybe if I can express each variable in terms of others or find some system to solve for them. Alternatively, since there are products and ratios, taking logs again? Wait, but they already are logs. \n\nAlternatively, maybe taking all equations and multiplying them together. Let me try that. \n\nMultiply LHS and RHS:\n\n(x/(y z)) * (y/(x z)) * (z/(x y)) equals (x y z) / (x * y z z)? \n\nWait let me compute numerator and denominator.\n\nNumerator: x * y * z \n\nDenominator: (y z) * (x z) * (x y) = y z * x z * x y = x² y² z².\n\nTherefore overall product is (x y z) / (x² y² z²) = 1/(x y z). \n\nOn the RHS, multiplying 2^{1/2} * 2^{1/3} * 2^{1/4} equals 2^{(1/2 + 1/3 + 1/4)}. \n\nLet me calculate that exponent. \n\n1/2 + 1/3 + 1/4. Let me find common denominator. 12? \n\nSo 6/12 + 4/12 + 3/12 = 13/12. \n\nTherefore, the product of the three LHSs equals 1/(xyz) equals 2^{13/12}?\n\nWait no:\n\nWait product of RHSs: \n\nEach equation converted to exponential gives us:\n\nnx/(y z) = 2^{1/2},\n\nny/(x z) = 2^{1/3}, \n\nnz/(x y) = 2^{1/4}\n\nMultiply all three: [x/(y z)] * [y/(x z)] * [z/(x y)] = [x y z]/[x² y² z²] = 1/(x y z). \n\nAnd the product of RHSs is 2^{1/2} * 2^{1/3} * 2^{1/4} = 2^{(1/2 + 1/3 + 1/4)}\n\n= 2^{(6+4+3)/12} = 2^{13/12}\n\nTherefore: 1/(x y z) = 2^{13/12}\n\nWhich implies that xyz equals\n\n2^{-13/12}\n\nWait that would be taking reciprocal on both sides:\n\nxyz = 1 / [2^{13/12}] = \n\n2^{-13/12}\n\nOkay that gives me xyz. But I need log₂(x⁴ y³ z²). \n\nLet me now denote log₂(x⁴ y³ z²). \n\nBy log properties, that equals log₂(x⁴) + log₂(y³) + log₂(z²) \n\nWhich equals 4 log₂ x + 3 log₂ y + 2 log₂ z. \n\nAlternatively, if I let a = log₂ x, b = log₂ y, c = log₂ z, then I need to find 4a + 3b + 2c. \n\nAlso since log(xyz) = log x + log y + log z = a + b + c = -13/12, taking log base 2:\n\n\nlog₂(xyz) = log₂(2^{-13/12}) → a + b + c = -13/12. \n\nSo if I can find individual values for a, b, c or relations between them, I can find 4a + 3b + 2c. \n\nAlternatively, maybe converting original equations into equations involving a, b, c. \n\nOriginal first equation:\n\nlog₂(x/(y z)) = log₂ x - log₂(y z) = log₂ x - (log y + log z) \n\n= a - (b + c) \n\nSo if a = log₂ x etc. Then that equation would be: \n\nna - (b + c) = 1/2 \n\nSimilarly for the second:\n\nlog₂(y/(xz)) is log y - (log x + log z) → b - (a + c) = 1/3 \n\nThird:\n\nlog z - (log x + log y) → c - (a + b) = 1/4 \n\nSo now we have three equations:\n\n1. a - b - c = 1/2 \n2. -a + b - c = 1/3 \n3. -a - b + c = 1/4 \n\nAnd also I know from earlier:\n\nna + b + c = -13/12 \n\nWait perhaps solving this system of equations?\n\n\nYes. Since four equations but only three variables. But since multiplying three original equations already used to get xyz. But since these equations would need to be consistent.\n\nAlternatively, since I already have the sum a + b + c = -13/12. \n\nAlternatively solving equations 1-3 for a,b,c.\n\nAlternatively, adding up equations:\n\nLet me write the first three equations again:\n\n(1): a - b - c = 1/2 \n(2): -a + b - c = 1/3 \n(3): -a - b + c = 1/4 \n\nLet me write each equation. Also since there is another equation:\n\n(4): a + b + c = -13/12 \n\nSo maybe solving equations (1), (2), (3) first. But perhaps if adding equations?\n\nAlternatively adding equations 1, 2 and 3? \n\nLet me check adding (1)+(2)+(3):\n\nLeft side:\n\na - b - c + (-a + b - c) + (-a - b + c) \n\nCompute term by term: \n\n\na -a -a = -a \n\nn-b +b -b = -b \n\n-c -c +c = -c \n\nSo overall left side: -a -b -c \n\nRight side: 1/2 + 1/3 + 1/4 \n\nWhich equals the 13/12 \n\nTherefore:\n\nn-(a + b + c) = 13/12 → a + b + c = -13/12\n\nWhich gives:\n\nn/na + b + c = -13/12 \n\nWhich matches with equation (4). \n\nTherefore equations (1), (2), (3) are consistent with equation (4). \n\nTherefore to solve, perhaps adding equation (1) + equation (4). \n\nAlternatively solving equations (1), (2), and (4). Since (4) replaces adding up all three. \n\nAlternatively perhaps express each variable. \n\nLet me take equations:\n\n(1): a - b - c = 1/2 \n(4): a + b + c = -13/12 \n\nLet me add them (1) and (4):\n\nLeft side: (a -b -c) + (a + b + c) = 2a \n\nRight side: 1/2 -13/12 \n\nLet me calculate 1/2 - 13/12. \n\nConvert 1/2 = 6/12. \n\nn6/12 - 13/12 = -7/12 → a = -7/24 \n\nSimilarly perhaps? \n\nnso that gives me value of a. \n\nOnce I get a, I can plug back into equations. \n\nAlternatively similarly for others. \n\nLet me try that.\n\n\nnAdding equation (1) + (4):\n\nnAs done, gives me:\n\nn2a = 1/2 + (-13/12)? Wait original (1) had RHS 1/2,\n\nand equation (4) RHS -13/12. \n\nWait adding both equations:\n\nnLHS becomes 2a. \n\nnRHS: 1/2 + (-13/12) = ? \n\nnYes:\n\nnConvert to same denominator:\n\nn1/2 = 6/12 \n\nSo 6/12 -13/12 = -7/12 \n\nnSo 2a = -7/12 → a = -7/24 \n\nnSimilarly if I do same with equation (2) and (4). \n\nnEquation (2): -a + b - c = 1/3 \nnEquation (4): a + b + c = -13/12 \n\nnAdding:\n\nnLeft: (-a + b - c) + (a + b + c) = 2b \n\nnRight: 1/3 -13/12 \n\nnAgain converting 1/3 as 4/12 \n\nn4/12 - 13/12 = -9/12 → -3/4 → b = -3/8 \n\nnSimilarly for equation (3) and equation (4). \n\nnEquation (3): -a - b + c = 1/4 \n\nnEquation (4): a + b + c = -13/12 \n\nnAdding them:\n\nnLeft: (-a -b + c) + (a + b + c) = 2c \n\nnRight: 1/4 -13/12 \n\nnConvert to same denominator:\n\nn1/4 = 3/12 \n\nSo 3/12 -13/12 = -10/12 → -5/6 \n\nnSo 2c = -5/6 → c = -5/12 \n\nnSo I have:\n\nnna = -7/24 \nnnb = -3/8 \nnnc = -5/12 \n\nnLet me check if that works. \n\nnAlternatively, since adding (1)+(4) gives me a, similarly with others. \n\nnOnce I know these values\nnNow, I have to compute 4a + 3b + 2c. \nnLet me me first compute each term. \n\nnBut since fractions with different denominators. \n\nnAlternatively compute numerically:\nn\nn4a = -7/24 ≈ -0.291666...\nnnb = -3/8 → -0.375 \nnnc = -5/12 → -0.416666...\n\nnAlternatively\ncalculate exact fractions:\n\nnCompute 4a: n4 * (-7/24) = -28/24 = -7/6 \n\nnn3b = 3 * (-3/8) = -9/8 \nnn2c =\n2 * (-5/12) = -10/12 = -5/6 \n\nnAdding all together:\n\nn-7/6 -9/8 -5/6 \n\nnCombine denominators. Let me add fractions. \n\nnLet me combine denominators starting by step:\n\nnFirst add -7/6 -5/6 since same denominator. \n\nn-7/6 -5/6 = -12/6 = -2 \n\nnNow we have -2 -9/8 \n\nnWhich equals -(2 + 9/8) \n\nnBut 2 = 16/8 \n\nnSo -(16/8 + 9/8) = -25/8 \n\nnWait:\n\nnOriginal sum: -7/6 -9/8 -5/6 \n\nnGroup (-7/6 -5/6) -9/8 \n\nn= (-12/6) -9/8 \n\nn= -2 - 9/8 \n\nnConvert to fractions:\n\nn-2 = -16/8 \n\nSo -16/8 -9/8 = -25/8 \n\nnBut since the problem asks for absolute value of log₂(x⁴ y³ z²), that is | -25/8 | equals 25/8 \n\nnWhich is already in reduced form since 25 and 8 have no common divisors except 1. \n\nnTherefore m is 25, 8, m + n equals 33. \n\nnWait but let me verify my calculations again because\nif there was miscalculations here then answer would be wrong. \n\nnLet me recompute 4a + 3b + 2c. \n\n\nn-7/24 \nnSo 4a: n4 * (-7/24) = -28/24 = -7/6 \nnSimilarly nb = -3/8 \nnn3b: -9/8 \nnnc = -5/12 \nnn2c: -10/12 = -5/6 \nnTherefore:\nnTotal sum: -7/6 -9/8 -5/6 \nnCombine denominators. \nnLet me take all terms over common denominator. \nnThe denominators are 6, 8 and again 6. Let me use 24? \nnn-7/6 = -28/24 \nnn-9/8 = -27/24 \nnn-5/6 = -20/24 \nnnAdding: -55/6 = -204/24 all over 24 \nnThat would be -(28 +27 +20)/24 \nnWhich equals -(75)/24 \nnSimplify: n-75/24 reduces? Let me divide numerator and denominator by 3? \nn-25/8 \nnSame as earlier. 25/8, so m + n is 33. \n\nnAlternatively to cross-verify, I can compute x, y, z numerically? \nnAlternatively check with Python. \n\nnSo if we take a = log₂x = -7/24 \nnSo x equals 2^{-7/24} \nnSimilarly y = 2^{-3/8}, z = \n2^{-5/12}\nnThen let me compute x/(y z)? \nnWhich equals 2^{-7/24}/( 2^{-3/8} * 2^{-5/12}) \nnnDenominator:\nn2^{-3/8 -5/12} \nnnWhich equals exponents:\nn-3/8 -5/12 \nnnCommon denominator of 24? n-3/8 = -9/24 -5/12 = -10/24 \nnnTherefore denominator = 2^{-19/24} \nnnTherefore overall ratio:\nnx/(2^{-19/24}) = 2^{[(-7/24)-(-19/24)]} \nn+ 19)/24] = 2^{12/24} = 2^{1/2} \nnWhich matches with original equation 1. \nnSimilarly for equation 2:\nnny/(xz) = log₂(2y/(xz)) = ?\nnnCompute exponent for y/(xz) would be log y - log₂x - log₂z \nnWhich would be b - a -c \nnPlugging values:\nn-3/8 - (-7/24) - (-5/12)\nnnLet me compute:\nn-3/8 + 7/24 + 5/12 \nnnConvert to 24 denominator:\nn-9/24 + 7/24 + 10/24 \nnn= (-9 + 7 + 10)/24 = 8/24 = 1/3 \nnWhich matches equation 2 as required. \nnSimilarly third equation:\nnnlog₂(z/(xy)) \nnnWhich equals log₂ z - log x \n- log y \nnnc -a -b = -5/12 - (-7/24) - (-3/8) \nnnCompute:\nnn-5/12 +7/24 +3/8 \nnnAgain common denominator of 24:\nn-10/24 +7/24 +9/24 \nnn= 6/24 = 1/4 \nnWhich is correct. \n\n"

Figure 11: The qualitative results of Qwen3 reasoning between 16K and 32K context (first part).

*Task Input (AIME24)*

Let $x, y$ and $z$ be positive real numbers that satisfy the following system of equations:
$\log_2\left({x \over yz}\right) = {1 \over 2}$
$\log_2\left({y \over xz}\right) = {1 \over 3}$
$\log_2\left({z \over xy}\right) = {1 \over 4}$
Then the value of $\left|\log_2(x^4 y^3 z^2)\right|$ is $\tfrac{m}{n}$ where $m$ and $n$ are relatively prime positive integers. Find $m+n$. The answer format must be: \boxed{'The final answer goes here.'}

*Output (Qwen3 @16K)* [4140 Tokens; 3 Tool-calls]

*Output (Qwen3 @32K)* [6462 Tokens; 3 Tool-calls]

Figure 12: The qualitative results of Qwen3 reasoning between 16K and 32K context (second part).

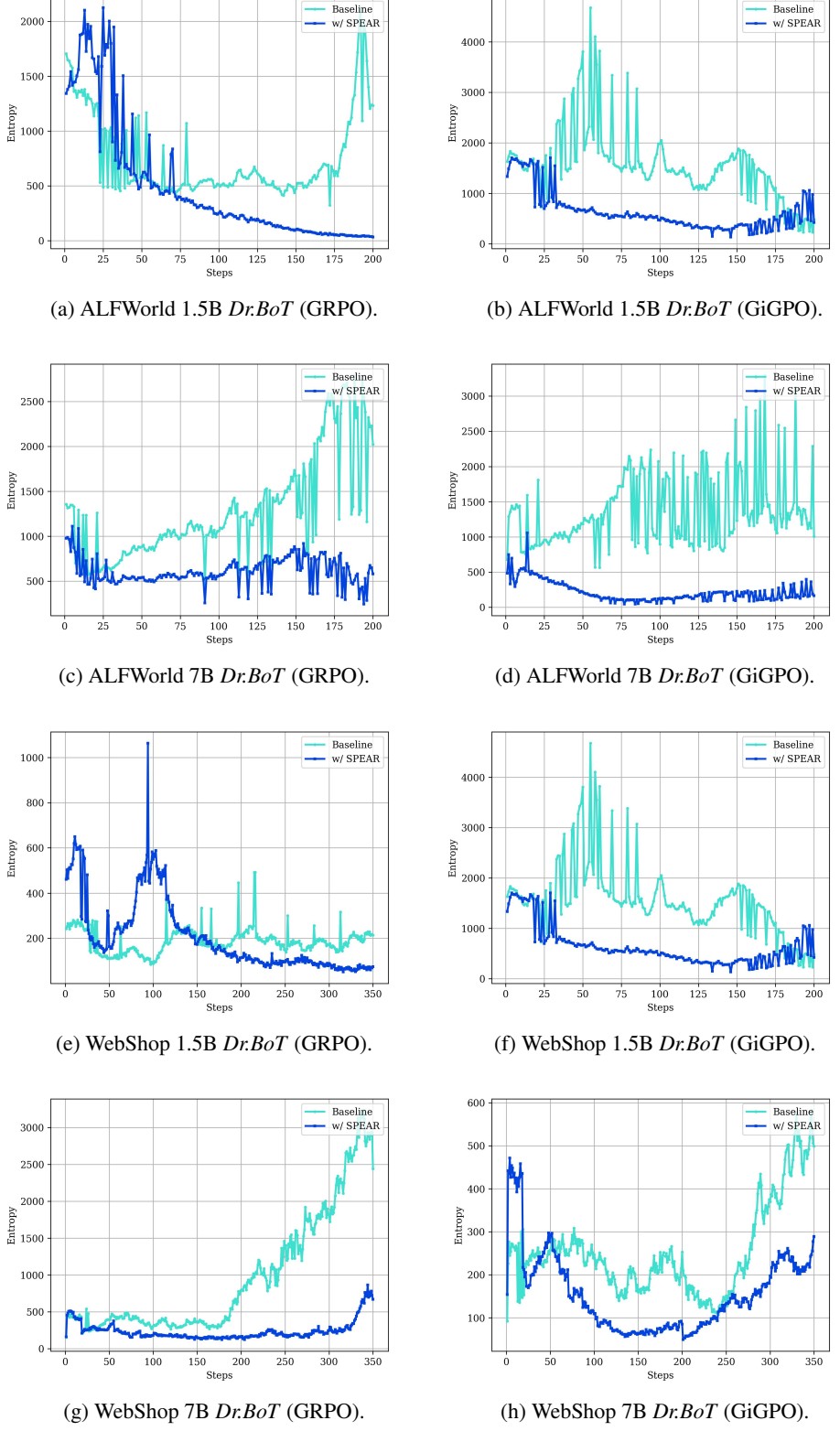

(a) ALFWorld 1.5B *Dr.BoT* (GRPO).

(b) ALFWorld 1.5B *Dr.BoT* (GiGPO).

(c) ALFWorld 7B *Dr.BoT* (GRPO).

(d) ALFWorld 7B *Dr.BoT* (GiGPO).

(e) WebShop 1.5B *Dr.BoT* (GRPO).

(f) WebShop 1.5B *Dr.BoT* (GiGPO).

(g) WebShop 7B *Dr.BoT* (GRPO).

(h) WebShop 7B *Dr.BoT* (GiGPO).

Figure 13: Entropy (`seq-mean-token-sum-norm`) across tasks, algorithms, and model scales.

A.15   TRAINING COST AND COMPLEXITY

The following contents are mentioned in Section 5.6 in the main text.

Table 10: Comparison on the complexity of the vanilla GRPO and the proposed SPEAR. PG, FW, BP, RB, and Adv respectively stand for the policy gradient loss computation, forward, back-propagation, replay buffer, and advantage. Out of simplicity, we use the $\mathcal{O}(M)$ to denote the forward FLOPs which is positively associated with the model size and the input length. $\mathcal{O}(P)$ denotes the BP operations proportional to the number of LLM parameters. We use $n_{\text{SIL}}$ to refer to the equivalent number of off-policy update (by SIL) per on-policy update. After filtering by $\hat{A}_j > 0$ & $\tilde{A}_j > 0$ (Equation 3), the number of samples in SIL is represented as $K, K \leq N_{\mathcal{D}}$.

| Training Stage | Computation of GRPO (vanilla) | Additional Computation by SPEAR | Description |
|---|---|---|---|
| On-policy Rollout | $2GT\mathcal{O}(M)$ | – | FW & sampling w/ $\pi_{\theta_{\text{old}}}$. |
| RB Update | – | $\mathcal{O}(GT)$ | Copy operation (negligible). |
| On-policy PG | $GT\mathcal{O}(M)$ | – | FW w/ $\pi_\theta$ (w/o KL $\pi_{\theta_{\text{ref}}}$). |
| On-policy BP | $\mathcal{O}(P)$ | – | BP w/ $\pi_\theta$. |
| RB Filtering | – | $\mathcal{O}(N_{\mathcal{D}})$ | Look-up operation (negligible). |
| Adv Recalibration | – | $\mathcal{O}(N_{\mathcal{D}})+\mathcal{O}(N_{\mathcal{D}_R})$ | Additive operation (negligible). |
| Replay PG | – | $n_{\text{SIL}}KT\mathcal{O}(M)$ $+ n_{\text{SIL}}\mathcal{O}(KT)$ | FW w/ $\pi_\theta$, token-wise clip& min (negligible). |
| Replay BP | – | $n_{\text{SIL}}\mathcal{O}(P)$ | BP w/ $\pi_\theta$. |
| In Total | $3GT\mathcal{O}(M) + \mathcal{O}(P)$ | $n_{\text{SIL}}(KT\mathcal{O}(M) + \mathcal{O}(P))$ | Dominance by FW & BP |

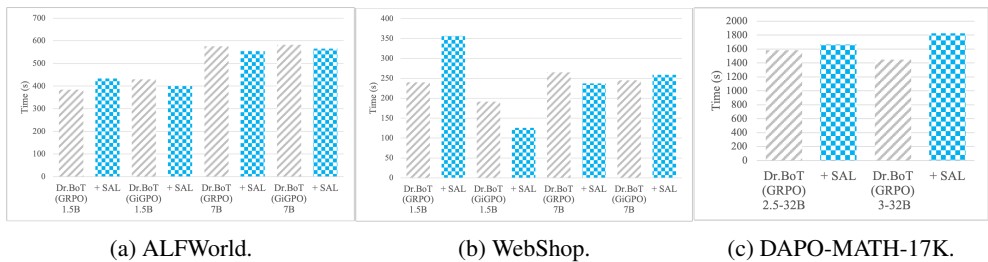

(a) ALFWorld.         (b) WebShop.         (c) DAPO-MATH-17K.

Figure 14: The averaged policy training time (s) per step with and without the proposed SPEAR.

We compare the computational complexity of our SPEAR with the vanilla GRPO algorithm in Table 10. Most of the computation comes from the forward and back-propagation of the filtered samples in the replay buffer. The memory operations such as the update and filtering of the buffer are light-weight and can be simply ignored. Given the current experimental settings (see Table 8), we observe that $n_{\text{SIL}} \approx 0.5$ for ALFWorld and WebShop, and $n_{\text{SIL}} \approx 0.33$ for DAPO-MATH-17K. In this case, our SPEAR additionally introduces around $10\% \sim 25\%$ computation overhead with $K \leq G$. Such computation complexity is acceptable in practice as the time of each training iteration is dominated by that of on-policy rollout generation.

Figure 14 shows the runtime per iteration step with and without the proposed SPEAR across different tasks and model scales. the total optimization procedure (including the rollout generation, advantage computation, log-probability inference, reward computation, and the actor update) is quite similar on average for ALFWorld, WebShop, and their SPEAR counterparts. For ALFWorld and WebShop, the 1.5B models exhibit larger variance than 7B models in training time. We believe such variance is associated with findings of the previous study (Havrilla et al., 2024) that the size of LLMs matters to the exploration diversity. Smaller LLMs are less diverse in exploring strategies due to their shallower reasoning nature, and are therefore prone to suboptimal policies with relatively increased stochasticity in training dynamics. For DAPO-MATH-17K, an increase around 5% and 26% is observed respectively on Qwen2.5 and Qwen3 models. Since the time per step is dominated by the rollout generation and actor update, we believe such increase in time is caused by the longer reasoning traces, more tool call interactions, and the additional iterations from the replay buffer. Such encouraged exploration by SPEAR is exaggerated on the reasoning model Qwen3 and leads to longer training time.

It is noted that the proposed SPEAR does not increase GPU memory usage. The introduction of the experience replay buffer is equivalent to increasing the training batch size per step. Due to the current sequential implementation that uses gradient accumulation with a fixed mini training batch size, we can achieve policy optimization on batches of any size without OOM issues.

## A.16 FUTURE WORK

### A.16.1 DYNAMIC SCHEDULING

In the future, one of the promising research direction is to model and adjust the scheduling parameters dynamically. It is noted that there exists no clear-cut line between exploitation and exploration during training (Snoek et al., 2012; Wang et al., 2018). The exploitation and exploration are intertwined and optimized together, which is often context-dependent (Bellemare et al., 2016) or guided by the policy itself (Pathak et al., 2017). Therefore, the scheduling should be progressive and smooth. We believe three kinds of techniques can be utilized for guiding the exploration:

**Entropy as the medium.** Following ARPO (Dong et al., 2025b), we could schedule the self-imitation and intrinsic reward with monitoring of the entropy itself. It is direct and intuitive, and it allows flexible and frequent adjustments. However, the modeling of the relationship between policy entropy and scheduling itself is often task-dependent and parameter-involved, introducing additional computation. In addition, the entropy is prone to noise where outliers of certain tokens might interfere with the scheduling negatively.

**Performance as the medium.** One could also adjust the scheduling by the performance-related metrics (Agrawal & Goyal, 2012) such as the task completion rate and the number of tool-calls. The association between exploration and success rate can be utilized. Furthermore, the number of tool-calls often indicates the degree of exploration with the environment. Nevertheless, the metrics might be deceptive as an early stop of exploration stimulation could lead to suboptimal convergence.

**Curiosity or Self-confidence as the medium.** One could intensify the exploration when the policy exhibits uncertainty (Pathak et al., 2017; Ladosz et al., 2022) about its actions or confusion about the transition of environment states. The policy's familiarity of the environment and its action reflects the exploration status. But it often requires parameterized learning of the curiosity or confidence via quantification of the inconsistency between the expected state transition and the real one.

### A.16.2 STEPWISE CREDIT ASSIGNMENT

In a extremely noisy tool ecosystem, the discrimination between good and bad experience is rather challenging merely with the outcome reward (Deng et al., 2025; Zeng et al., 2025). Under such circumstance, a process reward model (PRM) would be beneficial to provide fine-grained, stepwise supervision. However, it remains prohibitive to conduct manual evaluation and preference annotation for training online PRMs. Very recent studies highlight a few potential directions:

**The usage of meta-reward via LLM-as-a-Judge.** Instead of training a process reward from scratch, one could directly use an off-the-shelf LLM to assess each step not from the accuracy but from the aspect of meta-reasoning (Zhang et al., 2025b) behaviors (e.g., planning, exploration, and reflection).

**The employment of implicit PRMs.** One could derive an implicit PRM (Cui et al., 2025a) by reparameterization of the outcome reward as a sum of log-likelihood ratios of two LLMs over steps. Therefore, the step-wise reward can be approximated as the differences between two adjacent steps (agent actions) (Liu et al., 2025a)

**The introduction of world models.** The noise from real-world tool ecosystem might be inevitable and therefore it is reasonable to perform a model-based sim2real RL (Moerland et al., 2023). One could prepare an internal world model to deliver reliable state transition (Gu et al., 2024) for tool-based interaction, which help the agentic LLMs develop strategies via RL. Then, the trained LLM further adapts to real environment after a few more steps of training to gain robustness against noise.

### A.17 THE USE OF LARGE LANGUAGE MODELS

In the present study, we use the LLMs for the polishing of the manuscript writing and the discussions for analysis.

