# OpenReview forum: "Learn the Ropes, Then Trust the Wins: Self-imitation with Progressive Exploration for Agentic Reinforcement Learning"
_ICLR.cc/2026/Conference — ICLR 2026 Poster_

### Official Review · Reviewer_pTe3 · 2025-10-31

**Soundness:** 3
**Presentation:** 2
**Contribution:** 2
**Rating:** 2
**Confidence:** 2

**Summary:**

This paper aims to address the exploration-exploitation tradeoff for agentic
LLMs that use tools to complete certain tasks. The paper proposes a method
called SPEAR that enhances existing RL methods for finetuning LLMs. The main
function of SPEAR is to control the entropy of the policy to be within a narrow
range. It does this with methods such as intrinsic reward shaping and
self-imitation learning. Results on various benchmarks demonstrate that SPEAR
improves performance (success rates) when integrated with various contemporary
RL algorithms.

**Strengths:**

1. The paper seems to achieve high performance on the benchmarks presented.
1. My understanding is that the SPEAR method is achieving what it claims to,
   i.e., it is managing the entropy. This understanding is based on results like
   Figure 3.
1. The paper provides good low-level details of how exactly the method works.

**Weaknesses:**

As this paper is outside my field, I am unable to comment on the novelty of the
method or the validity of the low-level details, so my comments mainly focus on
the presentation of the paper:

1. While this paper excels at providing low-level details, I find that too much
   time is spent on these low-level details. As such, I overall had a hard time
   reading this paper and understanding the insight that it provides. For
   example, the abstract and introduction both dive quite deep into the
   low-level details of SPEAR, which causes them to be far too long. Meanwhile,
   in Section 4.1 and 4.2, it is unclear what the purpose is for each of these
   components, as the low-level details are just presented immediately.
1. Overall, the paper seems to be written for a highly technical audience who is
   already familiar with RL finetuning for LLMs. I think revising the paper to
   be written for a slightly broader audience would make it much easier to
   understand. For example, the abstract and introduction should seek to explain
   the high-level insights of the paper, rather than directly describing the
   modifications introduced by SPEAR.
1. Writing the paper for a high-level audience may help in shortening it. One
   aspect that seriously hurts this paper's readability is that important
   sections like Related Work and Discussion are left to the appendix. If all
   the sections truly need to be as long as they are, this paper may be better
   fit for a journal. That being said, I think this paper can be shortened a
   considerable amount by optimizing the usage of space. For example, there are
   many bulleted lists in the paper that could be switched to paragraphs.

I think a good starting point for addressing my feedback is: What is the key
insight of this paper? That is, if you had to summarize the contributions of
this paper in one sentence, what would you say?

**Questions:**

The following are my additional comments and questions on the manuscript.

1. The axis labels in all the figures tend to be quite small and hard to read.
   In particular, the plots in Figure 1 are illegible.
1. The main performance metric seems to be the success rates on the tasks; these
   success rates are treated as validating the changes introduced by SPEAR. Are
   there any measurements showing that SPEAR controls the entropy across a wide
   variety of tasks? Figure 3 seems to only show SPEAR's capabilities with
   respect to entropy in one task.

---

> ### Author Response · Authors · 2025-11-23
> **Rebuttal-Part 1**
>
> # About the Writing
>
> Thank you very much for pointing out the `overflow` problem of low-level details of the manuscript.
>
> First, we polish the abstract and the introduction to avoid being redundant and overlong. We also bring back related work and discussions on ablation study into the main text via properly using the space.
>
> Second, we rewrite the section 4 (method) to emphasize the purpose of each component. We remove those tedious, over-long descriptions that impede comprehension of the core ideas.
>
> Third, we would like to explain the key insight of the proposed paper: `we propose to achieve a exploration-exploitation balance of agentic RL by harmonizing scheduled self-imitation and intrinsic reward.` The progressive exploration is achieved across stage:
> * At the early stage:
> 1) facilitation of frequent tool calling by tool call reward (scheduled strength: high) -> mastering of tools -> *skill-level* exploration of environment with interactions;
> 2) restricted intervention from self-imitation replay buffer (scheduled strength: low) -> free-form rollout -> prevention of suboptimal policy convergence.
> * At the later stage,
> 1) downplaying of the tool call reward (scheduled strength: low) -> dominance by outcome reward -> prevention of reward hacking and competition against final accuracy;
> 2) guidance of the promising experience from the buffer (scheduled strength: high) -> familiarity about the environment and potential decition path -> *action-level* exploration for stable policy iteration.
>
> We have polished the entire manuscript to make it clearer and easy to follow for a broad range of readers.
>
> # About the Font Size
>
> Thank you for the suggestions. We have modified the figures to make them legible, and also enlarge the Figures 2,3,4 for readability.
>
> # About the Measurement of Exploration-Entropy
>
> Thank you for the suggestions. We modify the manuscript to provide more qualitative analysis on the entropy variance in the newly added Section A.14.3.
>
> Figure 12 illustrates the variance of entropy of the proposed *Dr.BoT* with and without SPEAR. We can observe that:
> * For most tasks and model scales, the policy entropy of the vanilla *Dr.BoT* does not converge. This is in line with our findings in Figure 3 where the consistent uncertainty about the environments and actions causes policy entropy divergence.
> * Due to the curriculum scheduling of self-imitation, the policy entropy maintains a steady trend across stages. The SPEAR allows sufficient exploration at the beginning and gradually strengthens imitation of self-generated promising experience. Therefore, the entropy varies smoothly during training.
> * Due to the curriculum scheduling of tool call reward, the interaction with the environments is encouraged and therefore the policy entropy of SPEAR can even surpass the baseline (e.g., ALFWorld 1.5B and WebShop 1.5B). However, such exploration about the environment does not necessarily correlate with entropy variation. We believe the distributional gap between task domains and the pretrained knowledge of LLMs plays a critical role. For larger models (7B), its internal parameterized knowledge is richer to handle the observation states properly.

---

> ### Comment · Reviewer_pTe3 · 2025-11-27
>
> I thank the authors for addressing my comments. I am quite impressed with how the presentation of the paper has been improved, and I can better understand it now. I am happy to recommend acceptance for this paper. I did have some minor comments remaining:
>
> 1. Where does the name Dr. BoT come from? I think the BoT stands for Bag of Tricks. What does the Dr. stand for?
> 2. The citations are often a bit hard to read since they don't have parentheses around them and thus are mixed into the text; I suggest using `\citep` instead of `\cite` so that the citations are placed within parentheses.

---

> > ### Author Response · Authors · 2025-11-27
> >
> > Thank you for the discussion. We are delighted that the response and modified manuscript resolve your concerns. With regard to the comments:
> >
> > - The *Dr.BoT* stands for the combination of *Dr.GRPO* [Ref1] and *Bag-of-Tricks* (BoT). Specifically, we adopt the *Dr.GRPO* techniques for removing the length and difficulty bias of the original GRPO algorithm. This is achieved by: 1) removal of intra-group normalization, and 2) removal of length normalization (Section A.8).
> >
> > - We have polished the paper and wrap all the in-line citations with parentheses for readability. Thank you for helping us improve the manuscript.
> >
> > [Ref1] Zichen Liu, Changyu Chen, Wenjun Li, Penghui Qi, Tianyu Pang, Chao Du, Wee Sun Lee, and Min Lin. Understanding r1-zero-like training: A critical perspective.

---

### Official Review · Reviewer_Zk4g · 2025-10-31

**Soundness:** 2
**Presentation:** 3
**Contribution:** 3
**Rating:** 6
**Confidence:** 3

**Summary:**

The paper tackles the exploration-exploitation trade-off in reinforcement learning (RL) for large language model (LLM) agents on long-horizon, sparsely-rewarded tasks. It proposes SPEAR, a curriculum-based self-imitation learning (SIL) method with intrinsic reward shaping and regularization, to progress policy entropy smoothly. SPEAR achieves performance improvements over baselines like GRPO, GiGPO, and Dr.BoT on ALFWorld, WebShop, and AIME24/25 benchmarks, with manageable computational costs.

**Strengths:**

**Innovative Method**: Proposes SPEAR, extending vanilla SIL with curriculum scheduling, advantage recalibration, and covariance-based clipping to address entropy collapse/divergence in LLM agents (Sec. 4; Eq. (5); Sec. 1). Integrates intrinsic reward shaping with a curriculum, distinguishing skill-level and action-level exploration tailored to LLM dynamics (Sec. 4.2; Fig. 4). Introduces Dr.BoT, a strong baseline combining RL tricks, enabling robust comparison of SPEAR’s performance (Sec. 4.3; Table 1, Table 2).

**Curriculum Design**: The integration of curriculum-based SIL with intrinsic reward shaping enables progressive transition from skill-level to action-level exploration, addressing entropy collapse (Sec. 4 and Fig. 2). Advantage recalibration using a FIFO buffer and 50th percentile baseline mitigates off-policy estimation drift and aligns with policy improvement (Eq. 2 and Sec. 4.1). Reward composition with curriculum scheduling (μ) for tool-call reward prevents competition with outcome reward (Eq. 6 and Fig. 4).

**Extensive Experimental Validation**: Evaluates on diverse benchmarks: ALFWorld (multi-turn), WebShop (e-commerce), AIME (math with code), covering different action spaces (Sec. 5.1; Table 1, Table 2). Compares against multiple baselines (GRPO, GiGPO, etc.), showing consistent gains (Table 1, Table 2). Uses ablation-like figures (Fig. 3, Fig. 4) to demonstrate self-imitation and intrinsic reward’s impact on entropy and performance.

**Weaknesses:**

**Mathematical Formulation Ambiguity**: Equation (2) for advantage recalibration uses the 50th percentile without theoretical justification; it is empirically described as "conservative but robust" but lacks derivation or comparison to other percentiles (Sec. 4.1). The SIL objective in Eq. (3) and (4) introduces multiple conditions, but the interaction between these advantages and their combined effect on policy convergence is not analytically explained.

**Limited Generalization**: Experiments focus on specific tasks (ALFWorld, WebShop, AIME) and models (Qwen series), with no main-text results on broader domains like vision-language agents, though mentioned in Appendix A.17 (Sec. 5.3). The method assumes that successful trajectories are clearly identifiable via positive advantages, but in stochastic environments with sparse rewards, this may lead to noisy experience selection, as acknowledged in Sec. 6.

**Vague Definition of Good Experiences**: The paper notes the vague definition of good experiences in highly complex, stochastic environments with unreliable tools, limiting generalizability (Sec. 6). No concrete examples or analyses in evaluated benchmarks illustrate this vagueness’s impact or occurrence frequency. The proposed stepwise supervision solution is mentioned but not elaborated or evaluated (Sec. 6).

**Questions:**

1.	How does SPEAR’s curriculum schedule adapt to different LLM sizes and task complexities? For instance, does a larger LLM need a different entropy progression rate, and does SPEAR adjust this dynamically (Sec. 4.1; Fig. 3)?
2.	In AIME24/25 experiments, how does SPEAR balance tool-call frequency and final accuracy under context length constraints (Table 2), and are there qualitative reasoning path examples illustrating this trade-off?
3.	How does the 10–25% extra theoretical complexity of SPEAR translate to actual training time and GPU memory usage across different benchmarks and LLM sizes (Sec. 1; Sec. 5.2)?

---

> ### Author Response · Authors · 2025-11-23
> **Rebuttal-Part 1**
>
> # About the Mathematical Rigor
>
> First, we would like to explain why the 50th percentile is preferred theoretically. For any estimator $b$, the loss minimization objectives of mean and median are respectively given as:
> * Mean: $\underset{b}{\operatorname{argmin}} \sum_{i=1}^n (R_i - b)^2 \implies b = \frac{1}{n}\sum_i R_i$
> * Median: $\underset{b}{\operatorname{argmin}} \sum_{i=1}^n |R_i - b| \implies b = P_{50}(\mathcal{R})$
> The influence functions $\text{IF}(\cdot)$ of the mean and median are respectively presented as:
> * Mean: $\text{IF}(R; \text{mean}) = R - \mathbb{E}[R]$ (unbounded), where a single outlier $R \to \infty$ drives $b \to \infty$.
> * Median: $\text{IF}(R; \text{median}) = \frac{\operatorname{sgn}(R - P_{50})}{2f(P_{50})}$ (bounded), where $f$ is the probability density function of returns; bounded even if $R \to \infty$.
> Therefore, the median is robust to outliers as its influence function is bounded.
> We also demonstrate that the median baseline provides a Lipschitz-continuous performance tracker under the assumption that the return distribution is modeled as non-stationary $R\sim p_{t_{\text{iter}}}(R)$ at the iteration $t_{\text{iter}}$. For two consecutive policy iterations $t_{\text{iter}}$ and $t_{\text{iter}}+1$ with return distributions $p_{t_{\text{iter}}}$ and $p_{t_{\text{iter}}+1}$, we have $|P_{50} (p_{t_{iter}+1}) - P_{50} (p_{t_{iter}})|$ $\leq$ $|p_{t_{iter}+1}$ - $p_{t_{iter}}|$. Therefore, the variation of $50$-th percentile $P_{50}$ is bounded for two consecutive iterations whereas the mean can change arbitrarily under the distribution shift.
>
>
> Second, we would like to explain that the 50-th percentile $P_{50}$ enjoys the following property while other percentiles do not. For any symmetric, unimodal return distribution with unknown variance, the median achieves minimax risk among all translation-equivariant estimators: $\inf_{\hat{b}} \sup_{p \in \mathcal{P}} \mathbb{E}[|\hat{b} - \mu(p)|] \propto P_{50}(\mathcal{R})$. Comparatively, the $P_{25}$ and $P_{75}$ percentiles are either overly conservative, pessimistic (e.g., slowing down self-imitation) or aggressive, optimistic (e.g., overfitting outliers) during advantage-based replay filtering.
>
>
> Third, the dual-gate replay-filtering is reflected in both Eq.3 and Eq.4. In Eq.3, we calibrate the advantage baseline of the experience in the replay buffer by the maintained $P_{50}$ percentile. The original $\hat{A}>0$ follows the definition of good experience in the original SIL, and the second $\tilde{A}>0$ further filters out-dated experience in the buffer given the improved baseline performance. In Eq.4, we use the calibrated $\tilde{A}$ to perform policy loss computation.
>
>
> Last but not least, we provide the convergence analysis on the combined effect of both $\hat{A}$ and $\tilde{A}$. We show that the combined effect of dual-gate filtering reduce the variance of the gradient updates, and ensures smoother policy improvement together with the clipping mechanism.
>
>
> We have added all the theoretical anaysis into the revised manuscript (see Section A.9).
>
> # About the Generalization
>
>
> First, the involved ALFWorld, WebShop, and AIME benchmarks respectively challenge the agent for world knowledge, planning, and complex reasoning capabilities. We use the Qwen series out of comparability, which ensures fair comparison with most of the studies in agent learning.
>
> Second, to further validate the generalizability,  we conduct experiments on reasoning-integrated information seeking tasks. We follow the settings of Search R1 [Ref1][Ref2]. Evaluation results show that the proposed SPEAR can greatly stimulate the tool-call based broad exploration in return for double-check of the information and thereafter specifically improves multi-hop QA (Section 5.5).
>
> Third, we provide the generalization studies on vision-language models in Section A.17 (initial submission). We add the experiments and analysis into the main text due to the extended page limit (10) at rebuttal/discussion period.
> Fourth, the response to the definitions of good experience are provided below. We will add the related discussions into the manuscript to make it clearer.
>
> [Ref1] Jin, B., Zeng, H., Yue, Z., Yoon, J., Arik, S., Wang, D., ... & Han, J. (2025). Search-r1: Training llms to reason and leverage search engines with reinforcement learning. arXiv preprint arXiv:2503.09516.
>
> [Ref2] Gao, J., Fu, W., Xie, M., Xu, S., He, C., Mei, Z., ... & Wu, Y. (2025). Beyond ten turns: Unlocking long-horizon agentic search with large-scale asynchronous rl. arXiv preprint arXiv:2508.07976.

---

> > ### Author Response · Authors · 2025-11-23
> > **Rebuttal-Part 2**
> >
> > # Definition of Good Experiences
> >
> >
> > First, to ensure fairness and reproducibility, all benchmarks involved are equipped with reliable tools and environments. It ensures that the existing RL algorithms are comparable.
> >
> > Second, we agree that the definition of good experience under complex, stochastic environments is an interesting topic as discussed in Section 6. The vanilla GRPO-like credit assignment (by the outcome reward) deteriorates experience learning from both positive and negative trajectories [Ref1][Ref2]. In this case, step-wise reward appears a promising solution. The process reward benefits the discrimination between good and bad actions at a fine-grained level. However, it remains quite unavailable to train a process reward model (PRM) for each agent task as manual evaluation and preference annotation is prohibitive at step-level. In this case, process reward modeling is impractical.
> > Motivated by very recent studies, we believe there exist a few potential directions:
> >
> > 1) The usage of meta-reward via LLM-as-a-Judge. Instead of  training a process reward from scratch, one could directly use an off-the-shelf LLM to assess each step not from the accuracy but from  the aspect of meta-reasoning behaviors (e.g., planning, exploration, and reflection) [Ref3].
> >
> > 2) The employment of implicit PRMs. One could derive an implicit PRM by reparameterization of the outcome reward as a sum of log-likelihood ratios of two LLMs [Ref4] over steps.  Therefore, the step-wise reward can be approximated as the differences between two adjacent steps (agent actions) [Ref5].
> >
> > 3) The introduction of world models. The noise from real-world tool ecosystem might be inevitable and therefore it is reasonable to perform a model-based sim2real RL [Ref6][Ref7]. One could prepare an internal world model to deliver reliable state transition for tool-based interaction, which help the agentic LLMs develop strategies via RL. Then, the trained LLM further adapts to real environment after a few more steps of training to acquire robustness against noise.
> >
> > We will provide more explanations on the potential solutions to noisy tool ecosystem in the newly added Section A.17.2.
> >
> > [Ref1] Deng, W., Ren, Y., Li, M., Sutherland, D. J., Li, X., & Thrampoulidis, C. (2025). On the Effect of Negative Gradient in Group Relative Deep Reinforcement Optimization. arXiv preprint arXiv:2505.18830.
> >
> > [Ref2] Wei, Q., Zeng, S., Li, C., Brown, W., Frunza, O., Deng, W., ... & Hong, M. Reinforcing Multi-Turn Reasoning in LLM Agents via Turn-Level Reward Design and Credit Assignment. In First Workshop on Multi-Turn Interactions in Large Language Models.
> >
> > [Ref3] Zhang, Z., Chen, Z., Li, M., Tu, Z., & Li, X. (2025). Rlvmr: Reinforcement learning with verifiable meta-reasoning rewards for robust long-horizon agents. arXiv preprint arXiv:2507.22844.
> >
> > [Ref4] Cui, G., Yuan, L., Wang, Z., Wang, H., Zhang, Y., Chen, J., ... & Ding, N. (2025). Process reinforcement through implicit rewards. arXiv preprint arXiv:2502.01456.
> >
> > [Ref5] Liu, X., Wang, K., Wu, Y., Huang, F., Li, Y., Zhang, J., & Jiao, J. (2025). Agentic Reinforcement Learning with Implicit Step Rewards. arXiv preprint arXiv:2509.19199.
> >
> > [Ref6] Moerland, T. M., Broekens, J., Plaat, A., & Jonker, C. M. (2023). Model-based reinforcement learning: A survey. Foundations and Trends in Machine Learning, 16(1), 1-118.
> >
> > [Ref7] Gu, Y., Zhang, K., Ning, Y., Zheng, B., Gou, B., Xue, T., ... & Su, Y. (2024). Is your llm secretly a world model of the internet? model-based planning for web agents. arXiv preprint arXiv:2411.06559.

---

> ### Author Response · Authors · 2025-11-23
> **Rebuttal-Part 3**
>
> # About the Curriculum-Schedule Sensitivity
>
> First, we would like to explain that most of the scheduling settings remain the same across benchmarks and LLM sizes (see Table 4). The performance gains are achieved on various tasks, domains, and LLMs of different sizes without meticulous fine-tuning.
>
> Second, the current scheduling design is simple yet efficient in: 1) stimulation of exploration about the environments via tool call as the medium; and 2) the prevention of overfitting premature, incomplete reasoning and acting patterns at an early stage. Extensive experiments across benchmarks of various domains and tasks confirm its effectiveness even with the same or similar hyper-parameter settings (see $T_{\text{warm-up}}$ and $T_{\text{decay}}$ from Table 5).
>
> Third, we would like to share guidelines on development of dynamic scheduling techniques for future studies.
>
> 1) Entropy as the medium. Following ARPO [Ref8], we could schedule the self-imitation and intrinsic reward with monitoring of the entropy itself.
> Pros: 1) It is direct and intuitive. 2) It allows flexible and frequent adjustments.
> Cons: 1) The modeling of the relationship between policy entropy and scheduling itself is often task-dependent and parameter-involved, introducing additional computation. 2) The entropy is prone to noise where outliers of certain tokens might interfere with the scheduling negatively.
>
> 2) Performance as the medium. One could also adjust the scheduling by the performance-related metrics [Ref10] such as the task completion rate and the number of tool-calls.
> Pros: 1) The association between exploration and success rate can be utilized. 2) The number of tool-calls often indicates the degree of exploration with the environment.
> Cons: The metrics might be deceptive as an early stop of exploration stimulation could lead to suboptimal convergence.
>
> 3) Curiosity or Self-confidence as the medium. One could intensify the exploration when the policy exhibits uncertainty [Ref4][Ref9] about its actions or confusion about the transition of environment states.
> Pros: The policy's familiarity of the environment and its action reflects the exploration status.
> Cons: It often requires parameterized learning of the curiosity or confidence via quantification of the inconsistency between the expected state transition and the real one.
>
> We have added the above discussions into the manuscript (see Section A.17.1).
>
> # About the Trade-off between Tool-Call Frequency and Final Accuracy
>
> Thank you for the suggestions. For 16K and 32K contexts, we respectively set the number of maximum turns of tool calls as 8 and 15. The average number of tool calls are reported below:
>
> | Model | # turns@16K | responses len@16K | # turns@32K | responses len@32K |
> |---|---|---|---|---|
> | Qwen2.5-32B | 7.18 | 4855.48 | 7.13 | 7502.59 |
> | Qwen3-32B | 3.23 | 10522.38 | 4.43 | 12371.95 |
>
> Interesting findings are observed:
>
> * For Qwen2.5 models, the number of tool call turns does not increase abruptly from 16K to 32K context. Instead, they stay similar. Two reasons are involved: 1) The tool call reward (Eq. 20) allows a maximum of 1 score which corresponds to 10 tool calls. More tool calls (>10) will not be specifically rewarded. 2) The intrinsic reward design is mainly targeted at stimulating exploration at the beginning and the dominance of outcome reward is guaranteed via scheduling. The mechanical increase of tool use for reward hacking will be penalized to promote reasoning itself.
> * For Qwen3-32B, the number of tool calls increase a bit but still falls behind that of Qwen2.5-32B. This is because the Qwen3 series are reasoning models and tend to develop sophisticated solution patterns via pure text. In this case, the agent mainly uses the tool to double-check its previous textual reasoning and computation. The context budget from 16K to 32K allows 2K more reasoning tokens and thereafter follows one or two more rounds of tool calls for verification.
>
> We also provide the qualitative examples to showcase the difference of tool-integrated reasoning under 16K/32K context in the newly added Section A.14.2. The reasoning and tool-calling patterns in both Qwen2.5 and Qwen3 examples are in line with the statistics mentioned above.

---

> > ### Author Response · Authors · 2025-11-23
> > **Rebuttal-Part 4**
> >
> > # About the Additional Cost
> >
> > First, we would like to explain that the proposed SPEAR does not increase GPU memory usage. The introduction of the experience replay buffer is equivalent to increasing the `train batch size`. Due to the current sequential implementation that uses gradient accumulation with a fixed `mini train batch size`, we can achieve policy optimization on batches of any size without OOM issues.
> >
> >
> > Second, we report the detailed actual training time per iteration step across tasks and model scales in Fig.12. In ALFWorld and WebShop, 1.5B models have higher training time variance than 7B models due to their smaller size and shallower reasoning. For DAPO-MATH-17K, Qwen2.5 and Qwen3 models witness 5% and 26% increase in training time (s), likely caused by longer reasoning traces and more tool call interactions. This effect is more obvious in Qwen3.
> >
> > To make it clearer, we have added explanations in Section A.15.

---

### Official Review · Reviewer_K8qo · 2025-10-31

**Soundness:** 3
**Presentation:** 3
**Contribution:** 3
**Rating:** 6
**Confidence:** 3

**Summary:**

This paper proposes SPEAR, a curriculum-based self-imitation learning recipe for training agentic LLMs that progressively balances exploration and exploitation by (i) shaping intrinsic rewards to encourage early, skill-level tool use, (ii) strengthening self-imitation later for action-level refinement, and (iii) stabilizing learning via advantage recalibration and covariance-based clipping to control entropy and policy drift. SPEAR stores only promising trajectories in a replay buffer (positive-advantage episodes), then: (1) schedules intrinsic "tool-call" rewards that decay over training so outcome rewards dominate; (2) recalibrates off-policy advantages using a rolling, median baseline to avoid costly recomputation; and (3) clips tokens whose log-probabilities covary strongly with advantage, preventing over-confidence and entropy collapse. Together, these yield a smooth transition from broad tool-skill exploration to focused action-selection exploitation.

**Strengths:**

- The paper targets a concrete failure mode in agentic RL for LLMs—entropy collapse from naive self-imitation—and designs a remedy that explicitly balances exploration and exploitation. This focus is well-motivated and illustrated (Figure 3) with clear problem framing.

- The method SPEAR is a cohesive recipe rather than a single trick: curriculum-style intrinsic tool-call rewards for early skill exploration, action-level self-imitation later, advantage recalibration for off-policy stability, and entropy/covariance-based clipping to prevent over-confidence and policy collapse. The components and their roles are specified precisely.

- The paper identifies and mitigates a practical reward-design pitfall: competition between outcome reward and tool-call reward causing oscillations. The scheduled intrinsic reward is a simple, effective fix.

- Empirical results are broad and positive across distinct action spaces (ALFWorld, WebShop, and DAPO-MATH-17K), demonstrating that the recipe transfers beyond a single domain. Metrics and evaluation protocols are clearly stated.

- The authors provide a strong, modern baseline (Dr.BoT) assembled from recent "bag-of-tricks," improving the fairness and relevance of comparisons; SPEAR then yields additional gains on top.

**Weaknesses:**

1/ The method hinges on a fuzzy notion of "good experiences" in stochastic, tool-heavy settings; sparse outcome rewards can make positive advantages look like noise rather than genuine policy improvement. This undermines the replay filter’s reliability. Would step-wise process rewards be mandatory in noisy tool ecosystems?

2/ Entropy control depends on hand-designed schedules and covariance-based clipping, whose optimal settings likely vary by task. This rigidity risks under-/over-regularization. Can the paper provide principled guidelines or adaptive rules beyond fixed schedules?

3/ The advantage recalibration and replay-filter condition $\hat{A} > 0, \tilde{A} > 0$ may bias learning toward "easier/high-return" slices of the data, potentially narrowing exploration and reducing robustness to hard cases. How sensitive are results to relaxing this double-positive gate?

4/ Parts of the qualitative analysis rely on an external LLM to label code intent, which could introduce evaluation bias. Is there an annotator-agreement check or a non-LLM rubric to validate these labels?

**Questions:**

Please refer to the questions in Weaknesses.

---

> ### Author Response · Authors · 2025-11-23
> **Rebuttal-Part 1**
>
> # About the PRM in noisy tool ecosystems
>
> Thanks for the enlightening comments. The current dual-gate replay buffer filtering enjoys two benefits: 1) It can capture the precious positive signals via intra-group relative advantage ($A_j>0$), which is regardless of difficulty bias (either $R_j>0$ or $R_j<0$); 2) It is robust to noise and off-policy update due to the recalibration of the baseline performance via the median.
>
> Indeed, the application of GRPO-like algorithms under the long-horizon agent tasks is rather challenging itself, especially with stochastic environment and sparse reward. The credit misassignment, caused by the outcome reward, deteriorates experience learning from both positive and negative trajectories [Ref1][Ref2]. In this case, step-wise reward appears a promising solution. The process reward benefits the discrimination between good and bad actions at a fine-grained level. However, it remains quite unavailable to train a process reward model (PRM) for each agent task as manual evaluation and preference annotation is prohibitive at step-level. In this case, process reward modeling is impractical.
> Motivated by very recent studies, we believe there exist a few potential directions:
>
> 1)The usage of meta-reward via LLM-as-a-Judge. Instead of  training a process reward from scratch, one could directly use an off-the-shelf LLM to assess each step not from the accuracy but from  the aspect of meta-reasoning behaviors (e.g., planning, exploration, and reflection) [Ref3].
>
> 2)The employment of implicit PRMs. One could derive an implicit PRM by reparameterization of the outcome reward as a sum of log-likelihood ratios of two LLMs [Ref4] over steps.  Therefore, the step-wise reward can be approximated as the differences between two adjacent steps (agent actions) [Ref5].
>
> 3)The introduction of world models. The noise from real-world tool ecosystem might be inevitable and therefore it is reasonable to perform a model-based sim2real RL [Ref6][Ref7]. One could prepare an internal world model to deliver reliable state transition for tool-based interaction, which help the agentic LLMs develop strategies via RL. Then, the trained LLM further adapts to real environment after a few more steps of training to acquire robustness against noise.
>
> We will provide more explanations on the potential solutions to noisy tool ecosystem in the newly added Section A.17.2.
>
> [Ref1] Deng, W., Ren, Y., Li, M., Sutherland, D. J., Li, X., & Thrampoulidis, C. (2025). On the Effect of Negative Gradient in Group Relative Deep Reinforcement Optimization. arXiv preprint arXiv:2505.18830.
>
> [Ref2] Wei, Q., Zeng, S., Li, C., Brown, W., Frunza, O., Deng, W., ... & Hong, M. Reinforcing Multi-Turn Reasoning in LLM Agents via Turn-Level Reward Design and Credit Assignment. In First Workshop on Multi-Turn Interactions in Large Language Models.
>
> [Ref3] Zhang, Z., Chen, Z., Li, M., Tu, Z., & Li, X. (2025). Rlvmr: Reinforcement learning with verifiable meta-reasoning rewards for robust long-horizon agents. arXiv preprint arXiv:2507.22844.
>
> [Ref4] Cui, G., Yuan, L., Wang, Z., Wang, H., Zhang, Y., Chen, J., ... & Ding, N. (2025). Process reinforcement through implicit rewards. arXiv preprint arXiv:2502.01456.
>
> [Ref5] Liu, X., Wang, K., Wu, Y., Huang, F., Li, Y., Zhang, J., & Jiao, J. (2025). Agentic Reinforcement Learning with Implicit Step Rewards. arXiv preprint arXiv:2509.19199.
>
> [Ref6] Moerland, T. M., Broekens, J., Plaat, A., & Jonker, C. M. (2023). Model-based reinforcement learning: A survey. Foundations and Trends in Machine Learning, 16(1), 1-118.
>
> [Ref7] Gu, Y., Zhang, K., Ning, Y., Zheng, B., Gou, B., Xue, T., ... & Su, Y. (2024). Is your llm secretly a world model of the internet? model-based planning for web agents. arXiv preprint arXiv:2411.06559.

---

> > ### Author Response · Authors · 2025-11-23
> > **Rebuttal-Part 2**
> >
> > # About the Hyperparameter Sensitivity
> >
> > Thank you for the suggestions. We would like to explain that:
> >
> > 1) Most of the current hyper-parameters share the same value settings across benchmarks of various domains and tasks. The performance gains can be achieved without specifically fine-tuning for each task. One could refer to our newly added Section A.13.2 for reference settings in practice.
> >
> > 2) The current scheduling settings are simple and intuitive without further modeling of their relationship with metrics such as entropy and performance. We agree with the reviewer that the adaptive design of such adjustment is a promising future direction. In this case, we would like to share three kinds of guidelines on adaptive rules:
> >
> > * Entropy as the medium. We suggest leveraging the entropy itself as a dynamic measure to guide the scheduling of self-imitation and intrinsic rewards. One could accelerate exploration if the entropy declines consistently, by encouraging more interactions via tool calls and reducing overfitting via covariance-based clipping and downplaying self-imitation. However, we believe the modeling between policy entropy and scheduling might still be parameterized and therefore the noise of entropy (e.g., spikes) should be considered during learning of such scheduling models.
> > * Performance as the medium. Another approach is to introduce performance-related metrics for scheduling, such as task completion rates and the number of tool calls. The lower completion rates or infrequent interactions indicate insufficient exploration and thereafter the scheduled decaying should be postponed. Nevertheless, metrics can be deceptive as the initial high success rates might cause inappropriate shortening of the exploration schedules and lead to pre-mature suboptimal convergence.
> > * Curiosity or Self-confidence as the medium. We suggest that the uncertainty of the policy about its actions and the environment state transitions can be used to measure exploration status. The low confident actions and unexpected transitions can be viewed as signals of `curiosity` and thereafter supervises broader exploration.
> >
> > We have added the above guidelines in the Section A.17.1 to help practitioners adjust scheduling according to their tasks.
> >
> >
> >
> >
> > # About the Advantage Recalibration
> >
> > First, we would like to explain that it is the usage of advantage instead of reward that helps alleiviate the bias of learning. The advantage computes the relative gains within each group where the $j$-th sample $A_j>0$ can still contribute to the policy even with $R_j<0$. That is to say, for both easy (e.g., positive rewards) and difficult (e.g., negative rewards) tasks, one can find sufficient replay experiences as long as there exists variance in a group. Extremely easy (with all responses correct) and hard (with all responses incorrect) cases will be ruled out ($A_j=0$).
> >
> > Second, the removal of standard deviation proposed by Dr.GRPO is vital in reducing the bias of difficulty. The division by std. of outcome reward in a group causes distortions to the advantage and imbalanced contributions from samples of different difficulty to the policy update. Therefore, our current replay filtering conditions are default with the removal of std. in advantage estimation.
> >
> > Third, our double-positive advantage gate for replay filtering is targeted at the off-policy problem where the policy is expected to improve over time and therefore the baseline gets elevated. Under such circumstance, we re-estimate the baseline using the 50-th percentile of a reward buffer and remove the outdated experience. To demonstrate the effectiveness of such double-gate filtering, we provide ablation in Fig.6 where we discard the advantage reestimation and filtering with $N_{\mathcal{D}_{R}}=0$. A decrease in both success rate and score is observed on the WebShop task (Dr.BoT 7B), showcasing that the direct imitation of experiences without proper calibration leads to severe off-policy update. Therefore, we believe the timely evaluation of replay buffer plays a critical role in shaping policy.
> >
> > To make it clearer, we have polished Section A.15.1 with more explanations.

---

> > > ### Author Response · Authors · 2025-11-23
> > > **Rebuttal-Part 3**
> > >
> > > # About the Details of LLM-based Code Intent Analysis
> > >
> > > For the qualitative analysis of the code intent, we follow ReTool [Ref1] to classify code snippets via LLMs. We first ask an external LLM to perform intent classification with open-ended categories in a free-form manner. Then, we manually inspect the mentioned categories for deduplication. We only keep the top 20 frequent intent categories: *calculation, computation, solution finding, problem-solving, geometric problem-solving, verification, geometric calculation, solution verification, modular calculation, probability calculation, geometric analysis, analysis, optimization, intersection finding, solution search, function analysis, configuration validation, data computation, game analysis, data processing, game strategy analysis, solution exploration, data analysis, list validation*. Then, we ask the LLM to further classify each code snippet into at most three categories. Finally, we randomly sample 50 classification results and manually check if the intent is correct or not. The agreement between the LLM and the manual inspection is above 90\% and therefore we believe the accuracy of an external LLM is sufficient for qualitative analysis.
> > >
> > > To make it clearer, more explanations have been added into Section A.14.1.
> > >
> > > [Ref1] Feng, J., Huang, S., Qu, X., Zhang, G., Qin, Y., Zhong, B., ... & Zhong, W. (2025). Retool: Reinforcement learning for strategic tool use in llms. arXiv preprint arXiv:2504.11536.

---

### Official Review · Reviewer_uAiT · 2025-11-01

**Soundness:** 3
**Presentation:** 3
**Contribution:** 2
**Rating:** 2
**Confidence:** 4

**Summary:**

The paper proposes SPEAR (Self-Imitation with Progressive Exploration for Agentic Reinforcement Learning), which addresses the exploration–exploitation trade-off for large language model (LLM) agents in long-horizon reinforcement learning. The method builds upon a curriculum-based self-imitation learning (SIL) framework to progressively control exploration during training.

**Strengths:**

1. The direction of agentic reinforcement learning for LLMs is highly relevant and important.
2. The paper demonstrates solid engineering effort, with comprehensive implementations and comparisons.
3. The writing is clear and easy to follow.

**Weaknesses:**

1. From my perspective, the paper seems to be more of an engineering combination than a foundational innovation. The paper reads as a well-crafted integration of existing techniques, e.g., self-imitation, curriculum learning, intrinsic reward shaping, and covariance-based regularization, etc., rather than a conceptually new algorithmic contribution. While the integration is effective, it feels closer to a sophisticated *recipe* or *best-practice collection* for training LLM-based agents than a novel RL framework with theoretical depth.
2. The paper defines exploration primarily as maintaining policy entropy to avoid collapse or divergence, and ties it closely to tool-use behavior. The notion of “skill-level exploration” is implemented through intrinsic rewards that encourage more tool calls. This design works well for tool-using environments but doesn't generalize to settings where exploration challenges arise from reasoning complexity, world modeling, creative planning, etc. As a result, SPEAR seems to address a narrow sub-problem of exploration–exploitation, rather than offering a broadly applicable framework.
3. SPEAR’s success depends on several manually set hyperparameters, like warm-up steps, decay schedules, clipping rates, covariance thresholds, and so on. The paper states that these are empirically chosen, and its own ablation results show significant sensitivity to them. This dependence undermines reproducibility and may limit generalization across tasks or models. Without principled guidance or adaptive mechanisms, achieving the reported results could require substantial tuning effort, reducing its appeal as a “plug-and-play” solution.
4. The paper states that the progressive entropy scheduling is one of main innovations, but all scheduling parameters (stages, thresholds, ratios) are fixed manually, with no criterion for determining when the policy is “ready to exploit.” This makes the method appear heuristic and task-specific, rather than a self-consistent or theoretically grounded algorithm.

**Questions:**

See the "Weakness".

---

> ### Author Response · Authors · 2025-11-23
> **Rebuttal-Part 1**
>
> # About the Novelty
>
> We would like to emphasize that the naive application of the GRPO-like RL algorithms for the agent learning of LLMs is facing various challenges such as: 1) the collapsing entropy where the agent fails to effectively build up the tool-use patterns and sufficiently interact with the environment; 2) the instability and runaway divergence where the agent exhibits consistent uncertainty about the low-probable, uncommon environment feedback; and 3) the short-cut reward hacking where the agent overfit the action tokens whose covariance with the outcome reward are overly high.
>
> In this case, we resort to the existing techniques but go one step further by presenting a recipe of harmonizing them into one framework that is validated across a wide range of tasks. We do acknowledge that the constituting components are motivated by previous studies (e.g., traditional RL methods), however, their integration practice under the agent context takes quite an amount of compute for verification and analysis of training dynamics for modifications. To test the generalizability, we conduct experiments across tasks and models. We confirm that the proposed recipe is indeed effective, which could benefit the research community for developing LLM agents via RL.
>
> # About the Limited Scope
> We would like to explain that:
> 1) The agent learning of LLMs is naturally tightly associated with tool learning, where the LLMs must perform tool calls to interact with the environments. This should not be interpreted as the limitation of research scope because tool usage is currently the bridge between the text generation and the acting agency for LLMs.
> 2) In our present study, we percieve tool learning as the **medium** instead of the ultimate goal. The stimulation of tool-usage behaviors is to encourage exploration of the environments rather than to simply master the tool itself. More frequent tool-calling brings about more varied, diverse feedback observations, which benefits the LLMs in self-reflection of previous actions and thereafter leads to the improved task-decomposiiton and planning.
> 3) We downplay the tool-call reward with a curriculum schedule to prevent reward hacking where the agent must focus on the final task completion. During such transition, the agent has to develop reasoning and planning skills.
> 4) Our generalization is exactly validated via the diverse benchmarks involved for various exploration challenges:
> - **Reasoning Complexity**->**AIME24/25**: This experimental settings requires high-level mathematic reasoning capabilities for problem decomposition and abstractive thinking. The code interpreter is only an assistant tool for executing codes, where the agentic reasoning itself cannot be ignored.
> - **World Modeling**->**ALFWorld**: This embodied setings challenges the LLMs world knowledge to complete household tasks. It requires the agent to reason and develop high-level policies in the textual world before solving embodied tasks via low-level actuation.
> - **Planning**->**WebShop**: This task needs sophisticated planning and requires the agent to adjust its strategy according to the webpage consistently.
> - **Compound**->**Sokoban**: This game challenges VLM to explore both the environments and the tactics to win.
> 5)To further validate the generalizability of our SPEAR, we introduce the new experimental settings of information seeking with local wiki database [Ref1][Ref2]. We follow [Ref1] to train agents to develop **knowledge-intensive reasoning** capabilities. Our results on Qwen2.5-7B-Instruct and Qwen2.5-14B-Instruct demonstrate the effectiveness of our SPEAR (Section 5.5).
>
> In a word, our extensive experiments across benchmarks of various domains and tasks verify that the proposed SPEAR, via the medium of tool learning, boosts LLM agents under exploration challenges with reasoning complexity, world modeling, and creative planning.
>
>
> [Ref1]. Gao, J., Fu, W., Xie, M., Xu, S., He, C., Mei, Z., ... & Wu, Y. (2025). Beyond ten turns: Unlocking long-horizon agentic search with large-scale asynchronous rl. arXiv preprint arXiv:2508.07976.
>
> [Ref2]. Jin, B., Zeng, H., Yue, Z., Yoon, J., Arik, S., Wang, D., ... & Han, J. (2025). Search-r1: Training llms to reason and leverage search engines with reinforcement learning. arXiv preprint arXiv:2503.09516.

---

> ### Author Response · Authors · 2025-11-23
> **Rebuttal-Part 2**
>
> # About Hyper-Parameters
>
>
> First, from Table 4, most of our hyper-parameters (e.g., $N_{\mathcal{D}}$, $N_{\mathcal{D_R}}$, $\lambda$, $T_{\text{decay}}$) remain the **same** across domains, tasks, and models (LLM/VLM). Such settings with consistent performance gains confirm the generalization of our SPEAR and its plug-and-play potentials.
> Second, the Fig. 6 should be interpreted properly as we vary hyper-parameters over a **very large scale** to investigate their impact at extreme. We increase $N_D$, $T_{\text{warm-up}}$, $T_{\text{decay}}$ by orders of magnitude (doubled each time), and we increased $N_{\mathcal{D_{R}}}$ and $\lambda$ by 4 and 10, respectively. Therefore, **variations of performance are expected** and informative in understanding the mechanism of each component.
> Third, Fig.6 shows that except extreme values (e.g., $N_{\mathcal{D}}=512/1024$, $N_{\mathcal{D_{R}}}=0/512/40960$, $\lambda=0.8$, $T_{\text{warm-up}}=0$, and $T_{\text{decay}}=400$), all other hyper-parameter settings of SPEAR achieve **superior performance** over the baseline (*Dr.BoT* on WebShop). It suggests that the choice of hyper-parameters within a reasonable range consistently brings in benefits and therefore enables plug-and-play property.
> Fourth, we would like to provide guidelines on how to set the hyper-parameters to help practitioners quickly apply SPEAR:
> * Replay buffer size $N_{\mathcal{D}}$: It should not be set too large to avoid severe off-policy deviation. A modest size of 2K~4K proportional to the training batchsize of 128 and group size of 16 (128x16=1024) is expected to work well for frequent refilling and policy update. In other word, $N_{\mathcal{D}}$ cam be set as 2x/4x of train_batch_size}$\times$n_samples_per_prompt.
> * Baseline buffer size $N_{\mathcal{D_{R}}}$: An appropriate setting between 2K and 10K prevents outdated and untimely estimation of current policy baseline performance. In other word, $N_{\mathcal{D_{R}}}$ can be set as 1x/4x of $N_{\mathcal{D}}$.
> * Covariance-based Clipping Ratio $\lambda$: The percentage of clipped tokens should be controlled between 0.02\% and 2\%. A smaller percentage would reduce the effect of anti-overfitting while a larger percentage slows down the policy exploitation of experiences.
> * Warm-up Step $T_{\text{warm-up}}$: The self-imitation should be scheduled to reach its maximum after 200 steps. For difficult tasks, it should be increased to allow exploration of diverse trajectories without convergence to local sub-optimum. One could first try $T_{\text{warm-up}}$=num_steps.
> * Decay Step $T_{\text{decay}}$: A decay step between 100 and 200 would be sufficient. If the tool is hard to master (e.g., complex slot-filling), the decay step should be increased to allow more stimulation of tool-calling behaviors. One could first try $T_{\text{decay}}$=num_steps.
>
> To help the readers better interpret the hyper-parameters and easily adapt SPEAR to different practical settings, we add explanations and clearer guidelines in the new Section A.13.2.

---

> > ### Author Response · Authors · 2025-11-23
> > **Rebuttal-Part 3**
> >
> > # About the Exploration-Exploitation Balance
> >
> > First, the exploitation-exploration trade-off is indeed a clich\'e topic in RL researches. Existing studies show that there exists no clear-cut line between exploitation and exploration during training [Ref1][Ref2]. In fact, they are intertwined and optimized together, which is often context-dependent [Ref3] or guided by the policy itself over time [Ref4].
> >
> > Second, the current scheduling design is simple yet efficient in: 1) stimulation of exploration about the environments via tool call as the medium; and 2) the prevention of overfitting premature, incomplete reasoning and acting patterns at an early stage. Extensive experiments across benchmarks of various domains and tasks confirm its effectiveness even with the same or similar hyper-parameter settings (see $T_{\text{warm-up}}$ and $T_{\text{decay}}$ from Table 5).
> >
> > Third, we follow the vanilla SIL [Ref5], PPO [Ref6], and GRPO [Ref7] to provide the theoretical analysis on the performance improvement bound in the newly added Section A.9. We demonstrate that *The self-imitation, with a warm-up schedule coefficient $\gamma(t_{\text{iter}})$ (Eq. 5), implements a constrained porjection onto the distribution of good responses, ensuring monotonic improvement on the high-reward subset.*
> >
> > Fourth, we would like to provide few promising techniques that can dynamically adjust the scheduling parameters, which could be studied in the future:
> > 1) Entropy as the medium. Following ARPO [Ref8], we could schedule the self-imitation and intrinsic reward with monitoring of the entropy itself.
> >
> > Pros: 1) It is direct and intuitive. 2) It allows flexible and frequent adjustments.
> >
> > Cons: 1) The modeling of the relationship between policy entropy and scheduling itself is often task-dependent and parameter-involved, introducing additional computation. 2) The entropy is prone to noise where outliers of certain tokens might interfere with the scheduling negatively.
> >
> > 2) Performance as the medium. One could also adjust the scheduling by the performance-related metrics [Ref10] such as the task completion rate and the number of tool-calls.
> >
> > Pros: 1) The association between exploration and success rate can be utilized. 2) The number of tool-calls often indicates the degree of exploration with the environment.
> >
> > Cons: The metrics might be deceptive as an early stop of exploration stimulation could lead to suboptimal convergence.
> >
> > 3) Curiosity or Self-confidence as the medium. One could intensify the exploration when the policy exhibits uncertainty [Ref4][Ref9] about its actions or confusion about the transition of environment states.
> >
> > Pros: The policy's familiarity of the environment and its action reflects the exploration status.
> >
> > Cons: It often requires parameterized learning of the curiosity or confidence via quantification of the inconsistency between the expected state transition and the real one.
> >
> > To make it clearer, we have added explanations on the scheduling for future research in the newly added Section A.17.1.
> >
> > [Ref1] Wang, H., Zariphopoulou, T., & Zhou, X. (2018). Exploration versus exploitation in reinforcement learning: A stochastic control approach. arXiv preprint arXiv:1812.01552.
> >
> > [Ref2] Snoek, J., Larochelle, H., & Adams, R. P. (2012). Practical bayesian optimization of machine learning algorithms. Advances in neural information processing systems, 25.
> >
> > [Ref3] Bellemare, M., Srinivasan, S., Ostrovski, G., Schaul, T., Saxton, D., & Munos, R. (2016). Unifying count-based exploration and intrinsic motivation. Advances in neural information processing systems, 29.
> >
> > [Ref4] Pathak, D., Agrawal, P., Efros, A. A., & Darrell, T. (2017, July). Curiosity-driven exploration by self-supervised prediction. In International conference on machine learning (pp. 2778-2787). PMLR.
> >
> > [Ref5] Oh, J., Guo, Y., Singh, S., & Lee, H. (2018, July). Self-imitation learning. In International conference on machine learning (pp. 3878-3887). PMLR.
> >
> > [Ref6] Schulman, J., Wolski, F., Dhariwal, P., Radford, A., & Klimov, O. (2017). Proximal policy optimization algorithms. arXiv preprint arXiv:1707.06347.
> >
> > [Ref7] Shao, Z., Wang, P., Zhu, Q., Xu, R., Song, J., Bi, X., ... & Guo, D. (2024). Deepseekmath: Pushing the limits of mathematical reasoning in open language models. arXiv preprint arXiv:2402.03300.
> >
> > [Ref8] Dong, G., Mao, H., Ma, K., Bao, L., Chen, Y., Wang, Z., ... & Dou, Z. (2025). Agentic reinforced policy optimization. arXiv preprint arXiv:2507.19849.
> >
> > [Ref9] Ladosz, P., Weng, L., Kim, M., & Oh, H. (2022). Exploration in deep reinforcement learning: A survey. Information Fusion, 85, 1-22.
> >
> > [Ref10] Agrawal, S., & Goyal, N. (2012, June). Analysis of thompson sampling for the multi-armed bandit problem. In Conference on learning theory (pp. 39-1). JMLR Workshop and Conference Proceedings.

---

### Author Response · Authors · 2025-11-23
**Summary**

Dear Reviewers, Area Chair, and Senior Area Chair,

We sincerely thank you for all the constructive comments that help benefit the polishing of our manuscript. Most of the concerns are around three aspects: 1) the robustness to hyper-parameters; 2) the adjustment of the curriculum schedule; and 3) the definition of the good experience. In addition, the writing of the paper raises the concerns of readability. We address the above concerns by:
* For the hyper-parameters, we show that most of our current settings remain the same across model scales and benchmarks of various tasks and domains, which confirms the generalization of our SPEAR. We also provide guidelines on the adjustment of the hyper-parameters. More discussions on the interpretation of the effect of the hyper-parameters are added into the manuscript.
* For the adjustment of the curriculum schedule, we show that the current design is simple-yet-effective. For the advanced development of dynamic scheduling, we provide three more potential techniques that are inspired by entropy, performance, and curiosity.
* For the definition of good experience and agent learning under noisy environment, we provide more explanations on the dual-gate advantage filtering mechanism, more theoretical analysis, and the promising future directions with step-wise, process supervision.
* For the writing and organization of the paper, we improve the usage of space and properly consider the broad aspect of the audience by emphasizing the motivations rather than the low-level details.

All the comments are addressed in the rebuttal and we perform modifications on the manuscript accordingly (marked in blue for differences). If there exist any further suggestions and questions, we are glad for discussions.

Authors of the Submission 288

---

### Author Response · Authors · 2025-12-01

Dear Reviewers, Area Chair, and Senior Area Chair,

We thank you for all the constructive comments and discussions. We appreciate reviewers for highlighting the stregnths of the present study:
- `relevant and important problem direction`, `solid engineering effort`, `clear writing` (Reviewer uAiT)
- `well-motivation and illustration`, `cohesive recipe`, `effective remedy for reward-design pitfall`, `broad empirical validation across domains`, `strong baseline Dr.BoT` (Reviewer K8qo)
- `innovative curriculum SIL design`, `progressive exploration-exploitation balance`, `extensive validation` (Reviewer Zk4g)
- `strong benchmark performance`, `achieve entropy control`, `good low-level details` (Reviewer pTe3)

A major concern is around the **hyper-parameter sensitivity** and **manual scheduling**. In response, 1) We demonstrate that (Table 4) most of the schedule-related hyper-parameters *remain fixed across all tasks, domains, and model scales*. It indeed confirms plug-and-play generalization. 2) We provide explicit practical guidelines (Section A.13.2) on the settings of these values relative to the batch size. 3) We clarify the ablation studies to showcase that an extensive magnitude of variation is set to study the mechanism where only those extreme values (Fig.7) bring significant performance drop. 4) We newly add the theoretical analysis to help analyze the benefits of SPEAR (Section A.9). 5) Detailed solutions to adaptive scheduling design are presented in the new Section A.16.1.

Another concern is around the **definition of good experience** in noisy environment. We address this by: 1) More explanations on the dual-gate filtering are provided with convergency analysis (Section A.9). 2) The detailed descriptions of the solutions agains noise (e.g., PRM, meta-rewards, world modeling) are presented in Section A.16.2.

We are glad that reviewer pTe3 explicitly confirms that our rebuttal and the revised manuscript resolve the concerns and recommend acceptance. `Thanks to the reviewer pTe3, we notice the overflow problem of low-level details that impede comprehension.` With such constructive suggestions, we could improve our manuscript consistently.

Despite that other reviewers did not post further comments to our response, we tried our best to polish the manuscript with point-by-point explanations, additional experiments, and analyses.

Finally, we would like to express gratitude that reviewers recognize our efforts to provide practical recipes for agentic RL via progressive exploration-exploitation. Our pioneering investigation on **harmonizing self-imitation and intrinsic reward design** would benefit the research community by highlighting that the naive, mechnical misappropriation of traditional RL techniques would cause unexpected training failure (e.g., entropy collapsing or runaway divergence).

Best regards,

Authors of Submission 288

---

### Meta-Review · Area_Chair_ZeH1 · 2026-01-06

**Summary:**

Reviewer concerns stated in the original reviews that informed my suggested decision:

Reviewer uAiT
- "more of an engineering combination than a foundational innovation"
- "design works well for tool-using environments but doesn't generalize to settings where exploration challenges arise from reasoning complexity"
- "ablation results show significant sensitivity to [hyperparameters]"
- "no criterion for determining when the policy is “ready to exploit.” This makes the method appear heuristic and task-specific, rather than a self-consistent or theoretically grounded algorithm."

Reviewer K8qo
- "The method hinges on a fuzzy notion of "good experiences" ... Would step-wise process rewards be mandatory in noisy tool ecosystems?"
- "Can the paper provide principled guidelines or adaptive rules beyond fixed schedules?"
- "How sensitive are results to relaxing [advantage recalibration and replay-filter condition]?"
- "Is there an annotator-agreement check or a non-LLM rubric to validate [external LLM code intent labels]?"

Reviewer Zk4g
- Mathematical Formulation Ambiguity
- Limited Generalization
- Vague Definition of Good Experiences

Reviewer pTe3
- too many low-level details and lapses in clarity
- missing explanation of high-level insights
- important sections deferred to appendix

**Reviewer Concerns:**

I believe the main outstanding concern is that there is no criterion for determining when the policy is "ready to exploit", this concern was shared by multiple reviewers. The authors acknowledged this as an issue with the approach and mentioned some "guidelines" (unvalidated) on directions for addressing this in the future.

**Reviewer Scores:**

- Reviewer uAiT: increase from 2->4, unsatisfied by the outstanding concern mentioned above
- Reviewer K8qo: maintain 6
- Reviewer Zk4g: maintain 6
- Reviewer pTe3: increase from 2->6 (they said they are happy to recommend acceptance but did not update their score https://openreview.net/forum?id=Kssko33Ekq&noteId=3QYTyxUWxu)

---

### Decision · Program_Chairs · 2026-01-26

Accept (Poster)